NORDITA 2024-028,
CQUeST-2024-0748

# Entanglement Rényi entropies in celestial holography

Federico Capone[1□], Andy O'Bannon[2○], Ronnie Rodgers[3⋆] and Somyadip Thakur[4Δ]

**1** Theoretisch-Physikalisches Institut, Friedrich-Schiller-Universität Jena,
Max-Wien-Platz 1, D-07743 Jena, Germany
**2** Department of Chemistry and Physics, SUNY Old Westbury, Old Westbury, NY
11568, United States
**3** Nordita, KTH Royal Institute of Technology and Stockholm University, Hannes
Alfvéns väg 12, SE-106 91 Stockholm, Sweden
**4** Center for Quantum Spacetime, Sogang University, Seoul 121-742, Korea
□ federico.capone@uni-jena.de, ○ obannona@oldwestbury.edu, ⋆ ronnie.rodgers@su.se,
Δ somyadip@sogang.ac.kr

January 23, 2025

## Abstract

Celestial holography is the conjecture that scattering amplitudes in $(d + 2)$-dimensional asymptotically Minkowski spacetimes are dual to correlators of a $d$-dimensional conformal field theory (CFT) on the celestial sphere, called the celestial CFT (CCFT). In a CFT, we can calculate sub-region entanglement Rényi entropies (EREs), including entanglement entropy (EE), from correlators of twist operators, via the replica trick. We argue that CCFT twist operators are holographically dual to cosmic branes in the $(d + 2)$-dimensional spacetime, and that their correlators are holographically dual to the $(d + 2)$-dimensional partition function (the vacuum-to-vacuum scattering amplitude) in the presence of these cosmic branes. We hence compute the EREs of a spherical sub-region of the CCFT's conformal vacuum, finding the form dictated by conformal symmetry, including a universal contribution determined by the CCFT's sphere partition function (odd $d$) or Weyl anomaly (even $d$). We find that this universal contribution vanishes when $d = 4$ mod 4, and otherwise is proportional to $i$ times the $d^{\text{th}}$ power of the $(d+2)$-dimensional long-distance cutoff in Planck units.

# 1 Introduction and Summary

## 1.1 Motivation: celestial holography

Celestial holography is the conjecture that scattering amplitudes of quantum field theories (QFTs) and (quantum) gravity in $(d+2)$-dimensional asymptotically Minkowski spacetime are dual to correlation functions of a $d$-dimensional CFT on the celestial sphere, that is, the sphere at null infinity, denoted $\mathcal{CS}^d$ [1–6]. We will call the $(d+2)$-dimensional theory the "bulk theory," and the putative CFT on $\mathcal{CS}^d$ the "celestial CFT" (CCFT).

To date, the main evidence for celestial holography comes from symmetry, and specifically from identifying the asymptotic symmetry algebra (ASA) of the bulk theory with the symmetry algebra of the CCFT. The ASA of $d$-dimensional asymptotically Minkowski spacetime includes the Poincaré algebra, which is the semi-direct sum of translations and the Lorentz algebra. Translations act as a constant shift of the advanced or retarded time coordinate, and have no effect on $\mathcal{CS}^d$. Lorentz transformations acts as globally smooth conformal transformations of $\mathcal{CS}^d$. The latter provide compelling evidence that a CFT might "live" on $\mathcal{CS}^d$, much the way the ASAs of asymptotically Anti-de Sitter (AdS) and de Sitter (dS) spacetimes provided compelling evidence of holographically dual CFTs at their boundaries, fully realized in the AdS/CFT [7–9] and dS/CFT [10–12] correspondences.

Any duality is ultimately an isomorphism between gauge-invariant states of the Hilbert spaces of two theories. In holography these are the bulk and boundary theories, and the isomorphism is called the "holographic dictionary." The AdS/CFT holographic dictionary is the most direct: we identify the asymptotic AdS time coordinate with the CFT time coordinate, which allows us to identify the Hamiltonians of the bulk and boundary theories, and hence to identify the Hilbert spaces. Equivalently, we equate the bulk and boundary theories' generating functionals of correlators. Indeed, in each theory these correlators are

the only observables, since an S-matrix is forbidden by the bulk theory's asymptotic AdS boundary conditions, or by the CFT's conformal symmetry.

The dS/CFT holographic dictionary is less direct, primarily because the bulk theory is Lorentzian while the boundary theory is Euclidean. The CFT thus encodes bulk physics in a fashion similar to a long-exposure photograph. More precisely, in dS/CFT we equate the bulk theory's wavefunction of the universe with the CFT generating functional [12].

The celestial holographic dictionary is even less direct, not only because the bulk theory is Lorentzian while the CCFT is Euclidean, but also because $\mathcal{CS}^d$ is codimension *two* relative to the bulk, in contrast to the codimension one boundary of AdS or dS. Moreover, the bulk theory's asymptotically Minkowski boundary conditions imply that, although a generating functional can be defined formally [13, 14], only the S-matrix is actually observable [15, 16].

The bulk theory's S-matrix encodes a transition matrix between states of the in-going and out-going Hilbert spaces. The celestial holographic dictionary maps this to a transition matrix between the two CCFT Hilbert spaces of the hemispheres of $\mathcal{CS}^d$. To understand how, in the bulk theory consider massless particles. An element of their S-matrix is determined by each in-coming and out-going particle's energy, and its position on $\mathcal{CS}^d$. If we Mellin transform the bulk S-matrix element from the momentum basis to the Lorentz boost basis, and also shadow transform either the in-coming or out-going wave functions, then the result manifestly has the form of a $d$-dimensional CFT correlator [17–22]. In particular, each bulk particles' energy and position on $\mathcal{CS}^d$ maps to the dimension and position of a conformal primary operator, respectively.[1] In the CCFT, we split $\mathcal{CS}^d$ into hemispheres, such that the operator insertions in a hemisphere define a state at the equator. A CCFT correlator thus encodes a transition matrix element between two states of the two hemispheres' Hilbert spaces. For more details, see for instance ref. [19].

Celestial holography has tremendous potential for studying QFTs and (quantum) gravity in $(d+2)$-dimensional asymptotically Minkowski spacetime, using the highly-developed techniques of $d$-dimensional CFT. Indeed, celestial holography already has tremendous successes. Chief among these are: the description of bulk soft theorems and memory effects in terms of CCFT Ward identities beyond tree level, the identification of the associated conserved operators in CCFT, the identification of novel soft theorems and memory effects, and the identification of higher-spin symmetry algebras' role in organizing scattering amplitudes. An essential, but partial, list of references is [21, 24–35]. Other CFT techniques, such as the conformal bootstrap, $c$-theorems, and so on, hold the promise for many more discoveries in the bulk theory. In particular, unitary and local CFTs are "solvable," in the sense that, via the operator product expansion (OPE), all local correlators are completely determined by the two- and three-point functions of conformal primaries. Celestial holography thus raises a tantalizing question: can we "solve" a CCFT in this sense, and if so, can we then "solve" the holographically dual QFT and/or (quantum) gravity theory?

However, celestial holography still faces many challenges. Most prominently, at present CCFT has a definition independent of the bulk theory only in special cases [13, 36–40]. In other words, almost everything we know about CCFT comes from the bulk theory: we use the bulk theory's S-matrix as input, and we extract CCFT information as output. In general, then, we have no independent way to compute anything in CCFT, so we have no way to check, prove, or derive the celestial holography conjecture.[2] Other challenges are more practical than conceptual. For example, the CCFT operator spectrum appears to

---

[1]For massive particles similar statements apply, with $\mathcal{CS}^d$ replaced by the sphere at past and future infinity, and the Mellin transform replaced by integration over a hyperbolic slice of Minkowski spacetime [23].

[2]The analogous situation in AdS/CFT would be knowing type IIB supergravity on $AdS_5 \times S^5$ but not knowing $\mathcal{N} = 4$ supersymmetric Yang-Mills theory, so that the only way to learn about $\mathcal{N} = 4$ super Yang-Mills would be through calculations in $AdS_5 \times S^5$.

violate reflection positivity, or in Lorentzian signature unitarity, casting doubt on whether the full strength of CFT can be brought to bear. Given such challenges, many basic questions about the CCFT remain unanswered, for instance about the spectrum of CCFT operators, their OPE, and so on.

One approach to these challenges is to adapt lessons learned from AdS/CFT. For examples of this strategy, see refs. [41–45]. One of AdS/CFT's biggest lessons about holography is the fundamental importance of the entanglement Rényi entropies (EREs), including the Entanglement Entropy (EE). Indeed, EREs have revealed properties characteristic of, if not unique to, the CFTs that appear in AdS/CFT, such as monogamy of mutual information (a linear combination of EEs of different subregions) [46], a maximal speed of entanglement spreading after a quench [47, 48], and others [49]. Most importantly, in AdS/CFT the bulk theory itself may emerge from the CFT's entanglement. For example, in AdS/CFT the EE of fluctuations about the CFT vacuum obeys the linearized Einstein equation [50]. More generally, AdS/CFT itself may be a "quantum error-correcting code" built from a tensor network, where the AdS geometry emerges from entanglement between the tensors: see for example refs. [51–55], and references therein.

AdS/CFT strongly suggests that EREs may be of similarly fundamental importance for celestial holography. Our goal is therefore to compute EREs in CCFT. Given that currently our only definition of CCFT is via the bulk theory, our main question is thus: how do we compute EREs of CCFT using the bulk theory? We will answer this question by translating CFT methods to the bulk theory, using the celestial holographic dictionary, and by drawing analogies to AdS/CFT and dS/CFT.

## 1.2 The replica trick in CFT

Let us quickly review key facts about EREs that we will use. For a quantum theory with Hilbert space $\mathcal{H}$, in a state specified by density matrix $\rho$, we choose a subspace $\mathcal{A} \subset \mathcal{H}$ and perform a partial trace of $\rho$ over states in $\mathcal{A}$'s complement, $\bar{\mathcal{A}}$. The result is a reduced density matrix, $\rho_{\mathcal{A}} \equiv \mathrm{tr}_{\bar{\mathcal{A}}} \, \rho$, which encodes all physical information available to an observer with access only to $\mathcal{A}$. For integers $n \geq 2$, the Rényi entropies are defined as

$$S_{\mathcal{A},n} \equiv \frac{1}{1-n} \log \mathrm{tr}_{\mathcal{A}} \, \rho_{\mathcal{A}}^n, \tag{1.1}$$

and the EE is defined by analytically continuing $S_{\mathcal{A},n}$ to non-integer $n$ and then taking the $n \to 1$ limit, which gives $\rho_{\mathcal{A}}$'s von Neumann entropy,

$$S_{\mathcal{A}} \equiv \lim_{n \to 1} S_{\mathcal{A},n} = -\mathrm{tr}_{\mathcal{A}} \left( \rho_{\mathcal{A}} \log \rho_{\mathcal{A}} \right). \tag{1.2}$$

For a QFT on a Lorentzian manifold, the Hilbert space $\mathcal{H}$ is the set of states on a Cauchy surface, and we will choose $\mathcal{A} \subset \mathcal{H}$ to be the states in a spatial subregion of this surface.[3] We will call the boundary of this subregion, $\partial\mathcal{A}$, the "entangling surface." In this context, the resulting Rényi entropies $S_{\mathcal{A},n}$ are called EREs, and encode information about the spatial distribution of correlations in the state specified by $\rho$.

Our main tool for computing EREs will be the "replica trick," developed for any QFT state in which $\rho$ can be represented as a path integral on a Euclidean manifold $\mathcal{M}$. (For a QFT in $(d+1)$-dimensional Minkowski spacetime, $\mathcal{M} = \mathbb{R}^{d+1}$.) The replica trick starts from the observation that $\mathrm{tr}_{\mathcal{A}} \, \rho_{\mathcal{A}}^n$ in equation (1.1) is proportional to the QFT's partition

---

[3]In what follows, in a mild abuse of notation, we will use $\mathcal{A}$ to denote both a subspace of $\mathcal{H}$ and the corresponding subregion of the Cauchy surface.

function $Z(\mathcal{M}_n)$ on the "replica manifold," $\mathcal{M}_n$, consisting of $n$ copies of $\mathcal{M}$ sewn together cyclically along $\partial\mathcal{A}$ by boundary conditions on QFT fields [56,57].[4]

In general, $Z(\mathcal{M}_n)$ is prohibitively difficult to calculate. However, for the conformal vacuum of a CFT on $\mathbb{R}^{1,1}$, such that $\mathcal{M} = \mathbb{R}^2$, and for $\mathcal{A}$ a single interval of length $\ell$, such that $\partial\mathcal{A}$ is two points, Cardy and Calabrese determined $Z(\mathcal{M}_n)$ using symmetries special to $d = 2$ CFT [56,57]. Specifically, when $\mathcal{M} = \mathbb{R}^2$, the conformal algebra is the Virasoro algebra, which includes global conformal transformations, namely $SL(2,\mathbb{C})$ transformations of $\mathbb{R}^2$'s complex coordinates $z$ and $\bar{z}$, plus an infinite number of *singular* conformal transformations. When $\mathcal{A}$ is a single interval, one such singular transformation maps $\mathcal{M} = \mathbb{R}^2$ to $\mathcal{M}_n$, namely the inverse of the "uniformisation map" that sends $\mathcal{M}_n$ to $\mathcal{M} = \mathbb{R}^2$. Cardy and Calabrese started from $Z(\mathcal{M} = \mathbb{R}^2)$, which is trivial, and then performed the inverse uniformisation map to determine $Z(\mathcal{M}_n)$. The resulting EREs are

$$S_{\ell,n} = \left(1 + \frac{1}{n}\right)\frac{c}{6}\log\left(\frac{\ell}{\epsilon}\right) + k_n + \dots, \tag{1.3}$$

where $c$ is the Virasoro central charge, $\epsilon$ is a short-distance/ultra-violet (UV) cutoff, $k_n$ is a constant that changes under re-definitions of $\epsilon$, and hence is unphysical, and $\dots$ represents terms that vanish as $\epsilon \to 0$. EREs in QFT typically diverge due to short-distance correlations across $\partial\mathcal{A}$, as the $S_{\ell,n}$ in eq. (1.3) does. Nevertheless, EREs can contain physical information. For example, the logarithmic derivative of $S_{\mathcal{A},n}$ in eq. (1.3) with respect to $\ell$ is $\left(1 + \frac{1}{n}\right)\frac{c}{6}$, which is $\epsilon$-independent, and hence is physical, and depends only on $c$, and hence is "universal." Crucially for us, eq. (1.3) depends only on the replica trick and Virasoro symmetry, and is valid even if the CFT is not reflection positive.[5]

In principle, the replica trick is applicable in any $d$, but when $d > 2$ conformal transformations are globally smooth, so no (inverse) uniformisation map exists. However, for any $d$ and any entangling surface, $Z(\mathcal{M}_n)$ is equivalent to that of $n$ decoupled copies of the CFT on $\mathcal{M}$, with a $\mathbb{Z}_n$ "replica symmetry" and with "twist fields" inserted at $\partial\mathcal{A}$ in each copy. These twist fields are defined by replica symmetry permutations around a circle of radius $\epsilon$ centered at $\partial\mathcal{A}$. They are thus codimension two operators, and are neutral under all global symmetries. When $d = 2$ they are local operators, and are Virasoro scalar primaries. Indeed, for $d = 2$ and $\mathcal{A}$ a single interval, Cardy and Calabrese identified $Z(\mathcal{M}_n)$ as the two-point function of these twist fields, to the $n^{\text{th}}$ power. This two-point function is completely determined by conformal symmetry up to the dimension of the twist fields, $h_n$, which the inverse uniformisation map determines to be

$$h_n = \left(1 - \frac{1}{n^2}\right)\frac{c}{24}, \tag{1.4}$$

leading to the EREs in eq. (1.3). Cardy and Calabrese's case was special due to the high degree of symmetry of a single interval in a $d = 2$ conformal vacuum. In most other cases, calculating twist field correlators is prohibitively difficult. Even in a $d = 2$ conformal vacuum, EREs of *more than one* interval require calculation of a *higher-point* correlator

---

[4]In a gauge theory the Hilbert space does not completely factorise, roughly speaking because closed loops of electric and magnetic flux can cross the entangling surface. The definition of EREs must therefore be modified, for example by expressing $\mathcal{H}$ as a subspace of the tensor product of Hilbert spaces associated to $\mathcal{A}$ and $\bar{\mathcal{A}}$, enlarged with edge modes [58–61]. We will compute EREs using the replica trick, which agrees with the edge mode definition of EREs [61]. As a result, we will not encounter any signs of Hilbert space non-factorisation, even if the CCFT is a gauge theory.

[5]However, in a non-reflection positive CFT the conformal vacuum may not be the ground state. For example, if an operator has dimension below the unitarity bound, then the state-operator correspondence can imply a state with energy below that of the conformal vacuum. Generically, in a non-reflection positive $d = 2$ CFT's actual ground state, single-interval EREs differ from eq. (1.3): for details, see ref. [62].

of twist fields, which is not completely determined by a conformal transformation, and so depends on more CFT data than just $c$. When $d > 2$, the twist fields are *non-local*, adding another layer of complexity to the calculation of their correlators.

## 1.3 The replica trick in celestial holography

To compute CCFT EREs holographically, we choose our bulk theory to be Einstein–Hilbert gravity with Newton's constant $G_N$, with any kind of minimally-coupled matter fields. Our strategy is to start with $d = 2$ and with $\mathcal{A}$ a single interval, $\mathcal{A} = \ell$, defined by two points on $\mathcal{CS}^2$, which define a great circle that splits $\mathcal{CS}^d$ into hemispheres. We will consider only the $d = 2$ CCFT conformal vacuum, meaning no operator insertions in either hemisphere. The celestial holographic dual is then 4-dimensional empty Minkowski spacetime, with no in-coming or out-going particles. In both the bulk theory and the CCFT, then, the transition matrix is in fact just the density matrix of the vacuum state. We then use two important entries from the celestial holographic dictionary, as follows.

First, we identify the bulk transformation that is holographically dual to the inverse uniformisation map from $\mathcal{M} = \mathcal{CS}^2$ to the replica manifold $\mathcal{M}_n$. As mentioned above, the bulk theory's Lorentz algebra is dual to the *global* conformal algebra, $SL(2,\mathbb{C})$, so to obtain the *singular* conformal transformation that maps $\mathcal{M} \to \mathcal{M}_n$, we must extend the Lorentz algebra. The extension that we need is well-known, and requires two steps. First, we extend the ASA to include "supertranslations," which shift the advanced or retarded time coordinate by an arbitrary function of $z$ and $\bar{z}$. The resulting ASA is a semi-direct sum of *super*translation and Lorentz generators, called the "Bondi, van Der Burg, Metzner, and Sachs (BMS) algebra." Second, we extend the BMS algebra to include "superrotations," defined as asymptotic conformal Killing vectors that act as *meromorphic* transformations of $z$ and $\bar{z}$, thus providing singular conformal transformations of $\mathcal{CS}^2$. The resulting ASA is a semi-direct sum of supertranslations and a Virasoro algebra with zero central charge (equivalently, a Witt algebra), called the "extended BMS algebra."[6] We cannot add superrotations without supertranslations: that algebra does not close.

Crucially for us, the superrotation that maps $\mathcal{M} = \mathcal{CS}^2 \to \mathcal{M}_n$ is well-known [69]: it generates a cosmic string in the bulk theory, with tension $(n-1)/n$ times $1/4G_N$, endpoints on $\mathcal{CS}^2$ at $\partial \mathcal{A}$, and zero charge under all bulk gauge symmetries. This cosmic string produces a conical singularity in the bulk spacetime, with no singularities on $\mathcal{CS}^2$. Invoking Cardy and Calabrese's arguments, we identify this cosmic string as the holographic dual of the CCFT twist fields, analogous to Dong's "cosmic brane" used to compute EREs in AdS/CFT [70]. We can make the bulk spacetime smooth by extending the angular coordinate around the cosmic string, but doing so produces conical singularities on the celestial sphere [71], thus giving us $\mathcal{M}_n$.

The second entry from the holographic dictionary that we use is: the CCFT's partition function, $Z(\mathcal{CS}^2)$, is equivalent to the bulk theory's partition function [15,16,72], which we call $Z_{\text{grav}}$. For practical purposes, we additionally assume a leading saddle-point approximation in the bulk theory, $Z_{\text{grav}} \approx e^{i S_{\text{grav}}^\star}$, where $S_{\text{grav}}^\star$ is the on-shell bulk action. To be clear, we do *not* identify *generating functionals* of the CCFT and the bulk theory (as done in AdS/CFT). Indeed, in the bulk theory the generating functional is not observable, as mentioned above. Instead, we simply follow the existing celestial holography dictionary, and equate an element of the bulk theory's S-matrix, namely the 0-to-0 S-matrix element, $Z_{\text{grav}}$, to a CCFT correlator, $Z(\mathcal{CS}^2)$, interpreted as the expectation value of the iden-

---

[6] We can extend the ASA beyond the extended BMS algebra, to include *diffeomorphisms* of $\mathcal{CS}^2$, producing the "generalized BMS algebra" [63–66]. In fact, we can extend the ASA even further, to include *local Weyl rescalings* of $\mathcal{CS}^2$, producing the "Weyl BMS algebra" [67,68]. These larger ASAs give rise to putative celestial theories more exotic than CFTs, so we will not consider them here.

tity on $\mathcal{CS}^2$, or equivalently as the inner product of the conformal vacuum with itself. In the bulk theory we then perform the superotation that enacts the singular conformal transformation $\mathcal{M} = \mathcal{CS}^2 \to \mathcal{M}_n$, as described above.

Crucially, $S_{\text{grav}}^{\star}$ diverges both at short distance, that is, in the UV, and at long distance, that is, in the infrared (IR). The UV divergence occurs near the cosmic string, which we interpret as the holographic dual of the UV divergence near the twist fields. We thus regulate the UV divergence with a cutoff, $\epsilon$. The IR divergence occurs near null infinity. Following for example refs. [42,73,74], we regulate this IR divergence with a cutoff, $L$. In principle, we should perform holographic renormalization [75] adding covariant counterterms on the regulator surfaces to cancel divergences as we remove the regulators by taking $\epsilon \to 0$ and $L \to \infty$. Holographic renormalization should produce results that are cutoff-independent, finite, and unambiguous, and hence in principle physically observable. However, how to perform holographic renormalization remains a (perhaps the) major open question in celestial holography [76,77], which we will not attempt to answer. Instead, we will simply match the $\epsilon$ dependence of our results to those expected in EREs, and match the $L$ dependence to existing results in celestial holography.

We can thus calculate the CCFT twist field two-point function on $\mathcal{CS}^2$, using the bulk theory. However, an essential question is how to interpret the result as EREs, given that the CCFT is defined in Euclidean signature, on $\mathcal{CS}^2$, and Wick rotation to Lorentzian signature is not unique. In general, different choices of Wick rotation lead to different Hilbert spaces, and thus to different interpretation of the result as EREs. For example, we could Wick rotate the direction normal to the great circle defined by the two twist fields, thus producing a compact time direction. Alternatively, we could Wick rotate the bulk theory to $(2,2)$ signature, thus changing the celestial sphere to a celestial torus [78,79], and again producing a compact time direction.[7] We could also perform a conformal transformation from $\mathcal{CS}^2$ to $\mathbb{R}^2$, which is simply stereographic projection to the complex plane, and then Wick rotate the direction normal to the line defined by the two twist fields. We prefer this last choice, which is conceptually simple, and gives the single-interval EREs of the CCFT in $\mathbb{R}^{1,1}$ in a fashion similar to Cardy and Calabrese.

## 1.4 Summary of results

With our preferred choice of Wick rotation, we find EREs of the form in eq. (1.3), with

$$c = i \, \frac{3 \, L^2}{G_N}. \tag{1.5}$$

Two features of the $c$ in eq. (1.5) are particularly prominent. First, the $c$ in eq. (1.5) is purely imaginary, providing additional evidence that the $d = 2$ CCFT violates reflection positivity/unitarity. Though unpleasant, this feature does not necessarily undermine holography, for instance, purely imaginary $c$ appears also in dS/CFT [11,12].

Second, the $c$ in eq. (1.5) depends on the bulk IR cutoff, $L$. Such $L$-dependence is actually required by dimensional analysis [42, 74, 82]. In the EREs of eq. (1.3), the coefficient of $\log{(\ell/\epsilon)}$ must be dimensionless, and the only parameters available in the bulk theory are $G_N$ and $L$. (In contrast, asymptotically AdS or dS spacetimes also have a radius of curvature/cosmological constant.) Moreover, $c$ comes from the on-shell bulk action, $S_{\text{grav}}^{\star}$, which is $\propto 1/G_N$, so we expect $c \propto L^2/G_N$.

On one hand, such $L$ dependence may indicate that $c$ is unphysical. For example, $L$ dependence could be a hint that $c$ comes from unphysical degrees of freedom that exist only

---

[7]For a bulk theory with $(2,2)$ signature, refs. [80, 81] develop a model of the CCFT as a quantum error-correcting code.

on the IR cutoff surface [83], and which might disappear after holographic renormalization. On the other hand, such $L$ dependence could be hinting that the number of degrees of freedom in the $d = 2$ CCFT diverges, since we remove the IR cutoff by sending $L \to \infty$. Such behavior is consistent with our experience/bias from AdS/CFT and dS/CFT, where the CFTs dual to classical gravity are usually in an 't Hooft large-$N$ limit, which in $d = 2$ CFT means $|c| \to \infty$.

However, in $d = 2$ CCFT the value of $c$ is in fact ambiguous. The reason is that the extended BMS algebra is a semi-direct sum of supertranslations with a center-less Virasoro algebra, and linear combinations of supertranslation and Virasoro generators produce new Virasoro algebras with different values of $c$. Equivalently, the $d = 2$ CCFT appears to have more than one symmetric, conserved spin-two operator [17,84,85], and as a result appears to be a logarithmic CFT [82]. To our knowledge, two proposals for $c$ currently appear in the literature. The first proposal is precisely eq. (1.5) [42,73,74], while the second proposal is $c = 0$ [86–88]. Our results actually do not answer the question of which $c$ the $d = 2$ CCFT has—we believe that only holographic renormalization can answer that. Instead, our main result is the *method* for computing twist field correlators from the bulk theory, based on the identification of the holographic dual to a twist field, namely a certain type of cosmic string.

To extend our method to cases that are *not* completely determined by symmetry, we also consider $d \geq 2$. Our bulk theory is again Einstein–Hilbert gravity with any kind of minimally-coupled matter fields, we work in the CCFT's conformal vacuum, dual to empty Minkowski spacetime, and we choose $\mathcal{A}$ to be a spherical subregion of radius $\ell$. For $d > 2$, codimension-two cosmic brane metric are known [89], but have conical singularities both in the bulk and on $\mathcal{CS}^d$, and hence are not dual to twist fields. Furthermore, in contrast to $d = 2$, we cannot obtain the appropriate cosmic brane solutions simply via a superrotation, consistent with the fact that when $d > 2$, conformal transformations are non-singular.[8] To obtain the appropriate cosmic brane solutions, we borrow techniques from Casini, Huerta, and Myers (CHM) [92], splitting the bulk spacetime into AdS and dS-sliced patches and writing the metric on each slice as a topological black hole. Doing so allows us to obtain solutions with a bulk conical singularity and smooth $\mathcal{CS}^d$, which we transform to a smooth bulk spacetime with twist field singularities on $\mathcal{CS}^d$, as in $d = 2$.

With our preferred choice of Wick rotation, we find the expected form for the EREs of a spherical sub-region of the conformal vacuum of a $d \geq 2$ CFT: our results appear in eqs. (4.63) and (4.67), which are the main results of this paper. These EREs include physical contributions. In particular, for $n = 1$ and odd $d$ they contain an $\epsilon$-independent term, and for $n = 1$ and even $d$ they contain a $\log(\ell/\epsilon)$ term. In general, for CFTs the former term is $\propto (-1)^{\frac{d}{2}-2} Z(\mathcal{CS}^d)$ and the coefficient of the latter is $\propto (-1)^{\frac{d}{2}-1} a_d$, where $a_d$ is the coefficient of the Euler density term in the CFT's trace anomaly. For reflection-positive/unitary RG flows between CFTs in $d = 2, 3, 4$, these physical quantities obey $c$-theorems [93–97], suggesting that they count dynamical degrees of freedom. Our two main results for $d \geq 2$ are the following. First, the physical contributions to ERE are all $\propto i L^d / G_N$, where again dimensional analysis dictates the $L$ and $G_N$ dependence, and the factor of $i$ suggests that $d \geq 2$ CCFTs violate reflection positivity/unitarity. Second, we find $a_d = 0$ for $d = 4 \mod 4$, raising the question of whether these CCFTs have *zero* dynamical degrees of freedom, in the sense mentioned above. As in $d = 2$, we believe that only holographic renormalization can answer this question.

To our knowledge, the only previous attempt to study CCFT entanglement appears in ref. [73], which applied AdS/CFT and dS/CFT in different "wedges" of pure Minkowski

---

[8]The $d > 2$ ASA can be extended to include *diffeomorphisms* of $\mathcal{CS}^d$ [90,91], which are also dubbed "superrotations." As mentioned in footnote 6, we will not consider such cases.

spacetime to compute EE (not all EREs) for a spherical sub-region of the CCFT conformal vacuum with $2 \leq d \leq 4$. Our results agree with those of ref. [73] where they overlap, including in particular the $c$ in eq. (1.5) and $a_4 = 0$. However, our methods are more general than those of ref. [73]. In particular, we identify the bulk dual of a twist field, which should allow for the calculation of more general EREs—for states besides the conformal vacuum, for sub-regions of various shapes, for any $n$, for any $d$, and so on. Indeed, for large classes of cosmic string solutions of the bulk theory, our method should provide holographic interpretations as correlators of twist fields in CCFT, although the interpretation as EREs will depend on the choice of Wick rotation, as mentioned above. We hope that our results will be useful for exploring the entanglement structure of CCFTs.

This paper is organized as follows. In section 2 we review Cardy and Calabrese's replica trick/twist field approach to calculating EREs in $d = 2$ CFTs. In sec. 3, we use the superrotation dual to the inverse uniformisation map to identify the cosmic string dual to the twist fields in $d = 2$ CCFT, and we subsequently compute the single-interval EREs. The main result of that section is eq. (3.46). In sec. 4 we adapt CHM's methods to Minkowski spacetime to compute spherical EREs of CCFTs in $d \geq 2$. The main results of this paper are in eqs. (4.63) and (4.67). In sec. 5 we conclude with a summary of our results and a discussion of questions for follow-up research. In the appendix, we reproduce the $d = 2$ result of eq. (3.46) by extending the methods of ref. [73] to $n \neq 1$.

## 2 Review: uniformisation and twist fields in CFT

In this section we review Cardy and Calabrese's replica trick approach to the computation of single-interval EREs in the conformal vacuum of a $d = 2$ CFT [56,57]. Readers familiar with this technique may safely skip to sec. 3.

As reviewed in sec. 1.2, for a quantum system with Hilbert space $\mathcal{H}$ in a state specified by density matrix $\rho$, the reduced density matrix $\rho_{\mathcal{A}}$ of a subspace $\mathcal{A} \subset \mathcal{H}$ is defined by tracing over states in the subsystem's complement, $\bar{\mathcal{A}}$: $\rho_{\mathcal{A}} = \mathrm{tr}_{\bar{\mathcal{A}}} \rho$. For integer $n \geq 2$, the EREs are then defined as

$$S_{\mathcal{A},n} \equiv \frac{1}{1-n} \log \mathrm{tr}_{\mathcal{A}} \rho_{\mathcal{A}}^n, \tag{2.1}$$

and the EE is the $n \to 1$ limit,

$$S_{\mathcal{A}} \equiv \lim_{n \to 1} S_{\mathcal{A},n} = -\mathrm{tr}_{\mathcal{A}} \left( \rho_{\mathcal{A}} \log \rho_{\mathcal{A}} \right). \tag{2.2}$$

In a Lorentzian QFT, when $\rho$ can be written as a path integral on a Euclidean manifold $\mathcal{M}$, when $\mathcal{A}$ is a spatial subregion, and when $n$ is an integer, $\mathrm{tr}_{\mathcal{A}} \rho_{\mathcal{A}}^n$ is proportional to the QFT's partition function $Z(\mathcal{M}_n)$ on a *replica manifold* $\mathcal{M}_n$, consisting of $n$ copies of $\mathcal{M}$ sewn together cyclically along $\mathcal{A}$ by boundary conditions on the QFT's fields. Writing eq. (2.1) in terms of $Z(\mathcal{M}_n)$ gives

$$S_{\mathcal{A},n} = \frac{1}{1-n} \left[ \log Z(\mathcal{M}_n) - n \log Z(\mathcal{M}) \right]. \tag{2.3}$$

In general, computing $Z(\mathcal{M}_n)$ is prohibitively difficult. However, for the conformal vacuum of a CFT in $\mathbb{R}^{1,1}$, such that $\mathcal{M} = \mathbb{R}^2$ or equivalently $\mathcal{M} = \mathbb{C}$, and when $\mathcal{A}$ is a single interval of length $\ell$ ($\mathcal{A} = \ell$), such that $\partial \mathcal{A}$ is two points, conformal symmetry determines $Z(\mathcal{M}_n)$ completely, as follows.

Let $\mathcal{M} = \mathbb{C}$ have coordinates $(z, \bar{z})$. The conformal vacuum's density matrix $\rho$ can then be written as a path integral over half of $\mathcal{M}$, say $\mathrm{Im}(z) \leq 0$, and $Z(\mathcal{M})$ can be written as a path integral over all of $\mathcal{M}$. If we choose $\mathcal{A}$ to be an interval with endpoints at $z = z_1$

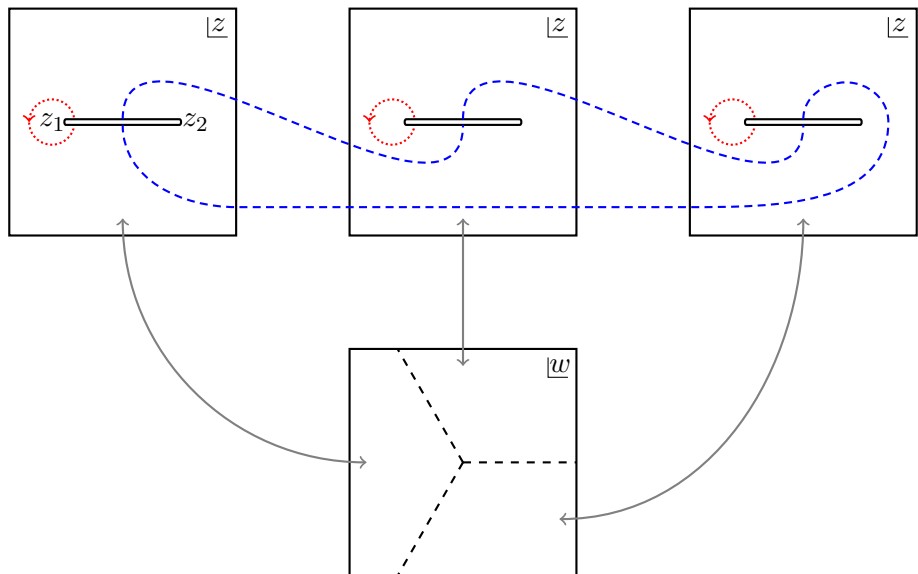

Figure 1: A depiction of the replica manifold $\mathcal{M}_n$ for a single interval in two dimensions, drawn here for replica index $n = 3$. Each square represents a copy of the complex plane $\mathbb{C}$. These are sewn together cyclically along the interval from $z = z_1$ to $z = z_2$, as indicated by the dashed blue curves. The dotted red curves show a closed loop encircling one of the endpoints of the interval. The total angle subtended by each loop is $2\pi n$, indicating a conical singularity at each endpoint. We also depict how the uniformisation map in eq. (2.4) sends each sheet of the replica manifold to part of a single complex plane with coordinate $w$, which then has conical singularities at the origin and infinity.

and $z = z_2$, then the replica manifold $\mathcal{M}_n$ consists of $n$ copies of $\mathcal{M} = \mathbb{C}$ sewn together cyclically along this interval, as depicted in figure 1 for $n = 3$. Conical singularities appear on $\mathcal{M}_n$ at $z = z_1$ and $z = z_2$, each of total angle $2\pi n$.

We can compute $Z(\mathcal{M}_n)$ using a singular conformal transformation, namely the so-called *uniformisation map* from $Z(\mathcal{M}_n)$ to the complex plane $\mathbb{C}$, where the latter has coordinates $w$ and $\bar{w}$:

$$w : \ \mathcal{M}_n \to \mathbb{C}, \qquad z \mapsto w(z) = \left( \frac{z - z_1}{z - z_2} \right)^{1/n} . \tag{2.4}$$

The uniformisation map sends the conical singularity at $z = z_1$ to $w = 0$, and sends the conical singularity at $z = z_2$ to $w = \infty$, as depicted in fig. 1 for $n = 3$. These conical singularities appear in the metric of the complex $w$-plane: if $\mathcal{M} = \mathbb{C}$ has metric $\mathrm{d}z \, \mathrm{d}\bar{z}$, then the uniformisation map gives

$$\mathrm{d}z \, \mathrm{d}\bar{z} = n^2 |z_2 - z_1|^2 \frac{|w|^{2(n-1)}}{|w^n - 1|^4} \, \mathrm{d}w \, \mathrm{d}\bar{w}, \tag{2.5}$$

where the conformal factor vanishes at $w = 0$ and $w = \infty$, reflecting the conical singularities at these points [98]. (The conformal factor in eq. (2.5) also diverges at $w^n = 1$ because these are the images of the points at infinity of the $n$ copies of the complex $z$-plane.)

Under a conformal transformation $z \to w(z)$ that produces $\mathrm{d}z \, \mathrm{d}\bar{z} = e^{2\chi(w,\bar{w})} \, \mathrm{d}w \, \mathrm{d}\bar{w}$, by definition the Weyl anomaly means the CFT partition function transforms as $Z \to Z'$

where $Z'$ is given by

$$\log \left( Z'/Z \right) = -\frac{c}{12\pi} \int dw \, d\bar{w} \, \left( \partial_w \chi \, \partial_{\bar{w}} \chi - 2 \, \partial_w \partial_{\bar{w}} \chi \right), \tag{2.6}$$

where $c$ is the Virasoro central charge.[9] Crucially, $Z(\mathcal{M} = \mathbb{C})$ is constant, or at most depends on marginal couplings. The conformal transformation in eq. (2.4) and the Weyl anomaly in eq. (2.6) thus completely determine $Z(\mathcal{M}_n)$ in terms of $Z(\mathcal{M})$, and in particular fix $Z(\mathcal{M}_n)$'s dependence on $z_1$, $z_2$, $c$, and $n$ to be

$$Z(\mathcal{M}_n) \propto |z_1 - z_2|^{-\left(n - \frac{1}{n}\right)\frac{c}{6}}. \tag{2.7}$$

Plugging eq. (2.7) for $Z(\mathcal{M}_n)$ into eq. (2.3) for the EREs gives the result in eq. (1.3): writing $\ell = |z_1 - z_2|$ and introducing a short-distance cutoff $\epsilon$, we find

$$S_{\ell,n} = \left( 1 + \frac{1}{n} \right) \frac{c}{6} \log \left( \frac{\ell}{\epsilon} \right) + k_n + \dots, \tag{2.8}$$

where $k_n$ is a cutoff-dependent, unphysical constant and ... represents terms that vanish as we remove the cutoff by sending $\epsilon \to 0$. The $n \to 1$ limit of eq. (2.8) gives the well-known result for a $d = 2$ CFT's single-interval EE [99],

$$S_\ell = \frac{c}{3} \log \left( \frac{\ell}{\epsilon} \right) + k_1 + \dots. \tag{2.9}$$

Cardy and Calabrese actually derived the $Z(\mathcal{M}_n)$ in eq. (2.7) in a conceptually distinct but completely equivalent way, using *twist fields*, as follows. Suppose we have $n$ decoupled copies of the QFT, that is, $n$ replicas of the QFT, on a single copy of $\mathcal{M}$. Such a theory has a $\mathbb{Z}_n$ *replica symmetry* that acts by permuting the replicas. A replica symmetry twist field is then defined by the property that QFT fields are permuted from one copy to the next as we encircle the twist field at radial distance $\epsilon$. We will denote such a twist field as $\Phi_n(z)$. These twist fields are scalar Virasoro primaries neutral under all global symmetries. For more details about twist fields see for example ref. [100], and references therein. The result for $Z(\mathcal{M}_n)$ in eq. (2.7) has the form of a two-point function of twist fields on $\mathcal{M}$, to the $n^{\text{th}}$ power, one power for each replica,

$$Z(\mathcal{M}_n) \propto \langle \Phi_n(z_1) \Phi_{-n}(z_2) \rangle^n, \tag{2.10}$$

where the subscripts $n$ and $-n$ indicate opposite permutations, and conformal symmetry requires, for twist field conformal weights $h_n = \bar{h}_n$,

$$\langle \Phi_n(z_1) \Phi_{-n}(z_2) \rangle \propto |z_1 - z_2|^{-2(h_n + \bar{h}_n)}. \tag{2.11}$$

Cardy and Calabrese determined $h_n = \bar{h}_n$ from the transformation of the stress tensor under the uniformisation map of eq. (2.4), with the result

$$h_n = \bar{h}_n = \left( 1 - \frac{1}{n^2} \right) \frac{c}{24}. \tag{2.12}$$

The $Z(\mathcal{M}_n)$ in eq. (2.10) is completely equivalent to that in eq. (2.7): plugging eqs. (2.11) and. (2.12) into eq. (2.10) reproduces eq. (2.7). We thus have two equivalent methods

---

[9]In eq. (2.6) the $\partial_w \partial_{\bar{w}} \chi$ term is a total derivative that contributes only in the presence of a boundary. We include this term for completeness, and because in the appendix we reproduce it for the CCFT via a bulk calculation: see eq. (A.16a).

to compute $Z(\mathcal{M}_n)$, namely by computing $Z(\mathcal{M}_n)$ itself (the "direct approach"), or by computing a twist field correlation function on $\mathcal{M}$.

We can also obtain $Z(\mathcal{M}_n)$ for any $\mathcal{M}$ conformal to $\mathbb{C}$, via Weyl transformation. We can use either method above, that is, we can Weyl transform either $Z(\mathcal{M}_n)$ itself, as determined by the Weyl anomaly, or the twist field two-point function, which has homogeneous Weyl transformation. For the $d = 2$ CCFT, we want $Z(\mathcal{M}_n)$ for the (celestial) sphere $\mathcal{M} = \mathcal{S}^2$, which is conformal to $\mathbb{C}$: the Weyl transformation $\mathrm{d}z\,\mathrm{d}\bar{z} \to 4(1 + z\bar{z}/R^2)^{-2}\mathrm{d}z\,\mathrm{d}\bar{z}$ maps $\mathbb{C}$ to a round sphere with radius $R$. In the first method, we Weyl transform from $\mathbb{C}$ to $\mathcal{S}^2$ and then perform the uniformisation map in eq. (2.4), again producing singularities at $w = 0$ and $w = \infty$ that are visible in the metric,

$$\frac{4}{(1 + z\bar{z}/R^2)^2}\mathrm{d}z\,\mathrm{d}\bar{z} = n^2|z_2 - z_1|^2 \frac{4|w|^{2(n-1)}}{(|w^n - 1|^2 + |z_2 w^n - z_1|^2/R^2)^2}\mathrm{d}w\,\mathrm{d}\bar{w}. \qquad (2.13)$$

We then find

$$Z(\mathcal{M}_n) \propto \left[2R\sin\left(\frac{\ell}{2R}\right)\right]^{-\frac{c}{6}(n-1/n)}, \qquad (2.14)$$

where $\ell$ is the great circle distance between $z = z_1$ and $z = z_2$, given by

$$R\sin\left(\frac{\ell}{2R}\right) = \frac{|z_1 - z_2|}{\sqrt{(1 + z_1\bar{z}_1/R^2)(1 + z_2\bar{z}_2/R^2)}}. \qquad (2.15)$$

In the second method, we use the fact that the twist field two-point function has homogeneous Weyl transformation, so that

$$\langle \Phi_n(z_1)\Phi_{-n}(z_2)\rangle = \left(\frac{2}{1 + z_1\bar{z}_1/R^2}\right)^{-2h_n}\left(\frac{2}{1 + z_2\bar{z}_2/R^2}\right)^{-2h_n}|z_1 - z_2|^{-4h_n} \qquad (2.16\text{a})$$

$$= \left[2R\sin\left(\frac{\ell}{2R}\right)\right]^{-\left(1 - 1/n^2\right)\frac{c}{6}}. \qquad (2.16\text{b})$$

Both methods give for the EREs,

$$S_{\ell,n} = \left(1 + \frac{1}{n}\right)\frac{c}{6}\log\left[\frac{2R}{\epsilon}\sin\left(\frac{\ell}{2R}\right)\right] + k_n + \ldots, \qquad (2.17)$$

where $\epsilon$ and the $k_n$ are the same cutoff and constants as in eq. (2.8). In the next section, our main goal will be to reproduce eq. (2.17) in the $d = 2$ CCFT's conformal vacuum from a calculation in the bulk theory, and specifically by identifying the holographic duals of the uniformisation map and twist fields.

The case of a single interval in the $d = 2$ conformal vacuum is highly symmetric, such that the EREs are completely determined by the conformal transformation in eq. (2.4). As a result, the only cutoff-independent information in these ERE is the central charge $c$, as we see in eqs. (2.8) and (2.17). However, practically any other case will be less symmetric and will not be determined just by a conformal transformation, and thus will likely depend on more detailed information than just $c$. For example, the case of two disjoint intervals in the $d = 2$ conformal vacuum can be calculated from a four-point function of twist fields, which in general will depend on details of the CFT's operator spectrum [101].

When $d > 2$ both methods for computing EREs remain available, but in general each is more difficult to use than in $d = 2$. Computing $Z(\mathcal{M}_n)$ directly is more difficult, even for a highly symmetric entangling surface (such as a plane or sphere) in a highly symmetric state (such as the conformal vacuum): when $d > 2$ the conformal algebra is finite-dimensional, in contrast to the Virasoro algebra, and conformal transformations are

non-singular, so nothing like the uniformisation map is available, in general. Computing $Z(\mathcal{M}_n)$ using twist fields is more difficult because twist fields are always codimension two, so when $d > 2$ they are no longer local operators. As a result, their correlation functions are more difficult to calculate, and are sensitive to more details of the CFT, such as higher-point functions of the stress-energy tensor [102]. In sec. 4, we discuss difficulties involved with cosmic branes dual to CCFT twist fields when $d > 2$, which is why we instead adapt CHM's methods to Minkowski spacetime to compute EREs when $d > 2$.

# 3 Holographic calculation of CCFT EREs: $d = 2$

## 3.1 Holographic duals of uniformisation and twist fields

In this section we focus on $d = 2$ and identify the superrotation dual to the uniformisation map in eq. (2.4), and subsequently identify the cosmic string dual to twist fields.

Throughout the paper we will use many different coordinate systems. To establish our conventions, we begin with a brief review of the 4-dimensional Minkowski metric written in three different coordinate systems.

First is spherical coordinates, $(t_s, r_s, \theta_s, \phi_s)$ (the subscript "s" stands for "spherical," as in ref. [103]), where $t_s \in (-\infty, \infty)$, $r_s \in [0, \infty)$, $\theta_s \in [0, \pi/2]$, and $\phi_s \in [0, 2\pi]$, and where the 4-dimensional Minkowski metric is

$$\mathrm{d}s^2 = -\mathrm{d}t_s^2 + \mathrm{d}r_s^2 + r_s^2(\mathrm{d}\theta_s^2 + \sin^2\theta_s\,\mathrm{d}\phi_s^2). \tag{3.1}$$

In spherical coordinates, we approach a generic point at future null infinity, $\mathcal{I}^+$, by sending $r_s \to \infty$ with $(t_s - r_s, \theta_s, \phi_s)$ fixed, and we approach a generic point at past null infinity, $\mathcal{I}^-$, by sending $r_s \to \infty$ with $(t_s + r_s, \theta_s, \phi_s)$ fixed.

Second is retarded Eddington–Finkelstein coordinates, $(u_s, r_s, \theta_s, \phi_s)$, where the retarded and advanced times are defined as, respectively,

$$u_s \equiv t_s - r_s, \qquad v_s \equiv t_s + r_s, \tag{3.2}$$

and hence $u_s \in (-\infty, \infty)$ and $v_s \in (-\infty, \infty)$. In retarded Eddington–Finkelstein coordinates, the 4-dimensional Minkowski metric is

$$\mathrm{d}s^2 = -\mathrm{d}u_s^2 - 2\,\mathrm{d}u_s\,\mathrm{d}r_s + r_s^2(\mathrm{d}\theta_s^2 + \sin^2\theta_s\,\mathrm{d}\phi_s^2). \tag{3.3}$$

We will also use stereographic coordinates on the sphere: upon switching to $z_s \equiv e^{-i\phi_s}\tan\theta_s/2$ where $z_s \in \mathbb{C}$, the metric in eq. (3.3) becomes

$$\mathrm{d}s^2 = -\mathrm{d}t_s^2 + \mathrm{d}r_s^2 + r_s^2\frac{4\,\mathrm{d}z_s\,\mathrm{d}\bar{z}_s}{(1 + z_s\bar{z}_s)^2}. \tag{3.4}$$

In retarded Eddington–Finkelstein coordinates, we approach a generic point at $\mathcal{I}^+$ by sending $v_s \to \infty$ with $(u_s, z_s, \bar{z}_s)$ fixed, and we approach a generic point at $\mathcal{I}^-$ by sending $u_s \to -\infty$ with $(v_s, z_s, \bar{z}_s)$ fixed.

Third is "flat slicing" coordinates, $(U, V, w, \bar{w})$, defined from spherical coordinates as

$$U \equiv \frac{t_s^2 - r_s^2}{t_s + r_s\cos\theta_s}, \qquad V \equiv t_s + r_s\cos\theta_s, \qquad w \equiv \frac{r_s\sin\theta_s}{t_s + r_s\cos\theta_s}e^{i\phi_s}, \tag{3.5}$$

so that $U \in (\infty, \infty)$, $V \in (-\infty, \infty)$, and $w \in \mathbb{C}$. In flat slicing coordinates, the 4-dimensional Minkowski metric is

$$\mathrm{d}s^2 = -\mathrm{d}U\mathrm{d}V + V^2\mathrm{d}w\,\mathrm{d}\overline{w}, \tag{3.6}$$

which is a flat slicing of Minkowski spacetime in $w$ and $\bar{w}$, hence the name. In flat slicing coordinates, we approach a generic point at $\mathcal{I}^{\pm}$ by sending $V \to \pm\infty$ with $(U, w, \overline{w})$ fixed.[10] We will also need the metric of the complex $w$-plane in polar coordinates: writing $w \equiv W e^{i\phi_s}$ with $W \in \mathbb{R}$, the metric is given by

$$\mathrm{d}s^2 = -\mathrm{d}U\mathrm{d}V + V^2 \left(\mathrm{d}W^2 + W^2 d\phi_s^2\right). \tag{3.7}$$

In what follows, two properties of flat slicing coordinates will be particularly useful to us. First, flat slicing coordinates preserve the product of retarded and advanced times:

$$UV = t_s^2 - r_s^2 = (t_s - r_s)(t_s + r_s) = u_s v_s. \tag{3.8}$$

Second, flat slicing coordinates automatically incorporate an antipodal matching between $\mathcal{I}^+$ and $\mathcal{I}^-$ at the point where they meet, namely at spatial infinity, $r_s \to \infty$ with $(t_s, \theta_s, \phi_s)$ fixed. This antipodal matching is necessary to define a unique ASA that acts on scattering ampltiudes [1]. At leading order near $\mathcal{I}^+$,

$$w = \tan\left(\frac{\theta_s}{2}\right) e^{i\phi_s}, \tag{3.9}$$

while at leading order near $\mathcal{I}^-$,

$$w = \tan\left(\frac{\pi - \theta_s}{2}\right) e^{i(\phi_s + \pi)}. \tag{3.10}$$

Comparing eqs. (3.9) and (3.10), we see that a point $(\theta_s, \phi_s)$ on the celestial sphere at $\mathcal{I}^+$ and the antipodal point $(\pi - \theta_s, \phi_s + \pi)$ on the celestial sphere at $\mathcal{I}^-$ give the same value of $w$. The flat slicing coordinates thus automatically incorporate the antipodal matching between $\mathcal{I}^+$ and $\mathcal{I}^-$ at spatial infinity, as advertised. We will therefore not explicitly mention the antipodal matching in what follows.

For a 4-dimensional spacetime describing an uncharged, static, straight cosmic string, the metric has the same form as eq. (3.1), (3.3), or (3.7), but with the replacement $\mathrm{d}\phi_s^2 \to K^2 \mathrm{d}\phi_s^2$, where $K$ is determined by the string's tension, $\mu$, as $K = 1 - 4\mu G_N$, where $K \neq 0$ or 1, or equivalently $\mu \neq 0$ or $1/4G_N$. If we maintain $\phi_s$'s periodicity as $\phi_s \sim \phi_s + 2\pi$, then introducing $K \neq 0$ or 1 introduces an angular deficit ($K < 1$) or excess ($K > 1$). Replacing $\mathrm{d}\phi_s^2 \to K^2 \mathrm{d}\phi_s^2$ in eq. (3.3) gives

$$\mathrm{d}s^2 = -\mathrm{d}u_s^2 - 2\,\mathrm{d}u_s\,\mathrm{d}r_s + r_s^2(\mathrm{d}\theta_s^2 + K^2 \sin^2\theta_s\,\mathrm{d}\phi_s^2). \tag{3.11}$$

The metric in eq. (3.11) is not asymptotically flat, but is asymptotically *locally* flat, in the sense of Bondi, because an asymptotic coordinate transformation from $(u_s, r_s, \theta_s, \phi_s)$ to Bondi coordinates $(\hat{u}, \hat{r}, \hat{\theta}, \hat{\phi})$ exists such that the asymptotic metric as $\hat{r} \to \infty$ is locally Minkowski, and the subleading terms have a non-vanishing shear tensor that is singular at the string's location [104]. This coordinate transformation is not a globally well defined BMS transformation, and so was named a *finite* superrotation in refs. [69,105].

Equivalently, we can perform a superrotation of the Minkowski metric, characterised by a local meromorphic function, that produces an exact metric (not just asymptotic) with a bulk conical deficit or excess, that is, a metric describing a cosmic string [105]. We will now review such superrotations, which will allow us to identify the superrotation holographically dual to the uniformisation map in eq. (2.4).

---

[10]We can also approach the south pole of the celestial sphere at $\mathcal{I}^+$ by sending $r_s \to \infty$ and $\theta_s \to \pi$ with $(t_s - r_s, r_s \sin\theta_s)$ fixed, or in flat slicing coordinates, $U \to \infty$ with $(V, w, \overline{w})$ fixed. Similarly, we can approach the north pole of the celestial sphere at $\mathcal{I}^-$ by sending $U \to -\infty$ with $(V, w, \overline{w})$ fixed. We mention these limits only for completeness. We will not need them in what follows.

In principle, we can write a finite superrotation using any coordinates. We will use the closed-form expression in ref. [103] (see also ref. [106]), who start with retarded Eddington–Finkelstein coordinates $(u_s, r_s, z_s, \bar{z}_s)$, transform to flat slicing coordinates $(U, V, w, \bar{w})$, and then write a finite superrotation as a coordinate transformation $(U, V, w, \bar{w}) \to (u, r, z, \bar{z})$ characterized by a meromorphic function $Z(z)$ as follows [103]:

$$
V \equiv \frac{\partial_z \partial_{\bar{z}} X}{Z' \bar{Z}'} + \sqrt{\frac{4r^2 - Y}{(\partial_u X)^2} + \frac{(Z'' \partial_z X - Z' \partial_z^2 X)(\bar{Z}'' \partial_{\bar{z}} X - \bar{Z}' \partial_{\bar{z}}^2 X)}{Z'^3 \bar{Z}'^3}}, \tag{3.12a}
$$

$$
U \equiv X - \frac{\partial_z X \partial_{\bar{z}} X}{Z' \bar{Z}' V}, \tag{3.12b}
$$

$$
w \equiv Z - \frac{\partial_{\bar{z}} X}{\bar{Z}' V}, \tag{3.12c}
$$

where $Z' \equiv \partial_z Z(z)$, $\bar{Z}' \equiv \partial_{\bar{z}} \bar{Z}(\bar{z})$, and

$$
X \equiv u(1 + z\bar{z})\sqrt{Z'(z)\bar{Z}'(\bar{z})}, \qquad Y \equiv \frac{u^2}{4}(1 + z\bar{z})^4 \{Z, z\}\{\bar{Z}, \bar{z}\}, \tag{3.13}
$$

and where $\{Z, z\} \equiv \frac{Z'''(z)}{Z'(z)} - \frac{3}{2}\left(\frac{Z''(z)}{Z'(z)}\right)^2$ is the Schwarzian derivative. The finite superrotation transforms the flat slicing metric in eq. (3.6) to a retarded Newman–Unti form,

$$
ds^2 = -du^2 - 2du\,dr + \left[\frac{4r^2}{(1 + z\bar{z})^2} + \frac{1}{4}u^2(1 + z\bar{z})^2\{Z, z\}\{\bar{Z}, \bar{z}\}\right]dz\,d\bar{z}
$$
$$
- ur\left(\{Z, z\}dz^2 + \{\bar{Z}, \bar{z}\}d\bar{z}^2\right). \tag{3.14}
$$

In the coordinates $(u, r, z, \bar{z})$, we approach a generic point on $\mathcal{I}^+$ by taking $r \to \infty$ with $(u, z, \bar{z})$ fixed. At leading order in that limit, eq. (3.12) approaches

$$
V = \frac{2r}{(1 + z\bar{z})\sqrt{Z'\bar{Z}'}} + \mathcal{O}(r^0), \tag{3.15a}
$$

$$
U = u(1 + z\bar{z})\sqrt{Z'\bar{Z}'} + \mathcal{O}(r^{-1}), \tag{3.15b}
$$

$$
w = Z + \mathcal{O}(r^{-1}). \tag{3.15c}
$$

Eq. (3.15c) shows that, asymptotically, a finite superrotation acts as a meromorphic map from the complex $z$-plane to the complex $w$-plane. Indeed, finite superrotations are defined to act asymptotically as a combination of a conformal transformation of the celestial sphere $\mathcal{CS}^2$ and a compensating Weyl transformation that restores the round sphere metric on $\mathcal{CS}^2$. We will define the $\mathcal{CS}^2$ metric as $\lim_{r\to\infty} ds^2/r^2$, in which case eq. (3.14) shows that any superrotation preserves $\mathcal{CS}^2$'s round sphere metric, $4dz\,d\bar{z}/(1 + z\bar{z})^2$. Finite superrotations also preserve the product of retarded and advanced times: eq. (3.12) preserves $UV = uv$.

To be clear, the coordinates $(u, r, z, \bar{z})$ in the superrotated metric in eq. (3.14) are in general different from the coordinates $(u_s, r_s, z_s, \bar{z}_s)$ in the Minkowski metric in eq. (3.4). To see why, compare for example $U$, $V$, and $w$ in eq. (3.5) to those in eq. (3.15). Only for the identity map, $Z(z) = z$, do we find $(u, r, z, \bar{z}) = (u_s, r_s, z_s, \bar{z}_s)$. For any other $Z(z)$, these two coordinate systems will be different. However, a subset of superrotations act as isometries of the Minkowski metric, namely the Lorentz transformations, given by $SL(2, \mathbb{C})$ transformations,

$$
Z(z) = \frac{az + b}{cz + d}, \tag{3.16}
$$

with $a, b, c, d \in \mathbb{C}$ and $ad - bc = 1$. By definition, the Schwarzian derivative vanishes for $SL(2, \mathbb{C})$ transformations, in which case the superrotated metric in eq. (3.14) clearly has the form of the Minkowski metric in eq. (3.4).

A superrotation with $Z(z) = z^K$, where $K \neq 0$ or 1, converts Minkowski spacetime to a spacetime with an uncharged, straight, static cosmic string [69,107]. Following ref. [108], Strominger and Zhiboedov showed this in ref. [69] by converting the cosmic string metric in eq. (3.11) to Newman–Unti form with stereographic coordinates, and compared to the superrotated metric in eq. (3.14) with $Z(z) = z^K$, finding that the two metrics had identical shear tensors. More intuitively, plugging $Z(z) = z^K$ into eq. (3.15c) gives $w = z^K$ asymptotically, and writing this as $w = |z|^K e^{iK\phi_s}$ shows that $K \neq 0$ or 1 produces an angular deficit or excess in $\phi_s$, as expected for a cosmic string. In particular, in Cartesian coordinates $(t_s, x_1, x_2, x_3)$, the superrotation with $Z(z) = z^K$ produces a spacetime with a static cosmic string along $x_3$, whose endpoints reach $\mathcal{CS}^2$ at $\theta_s = 0$ and $\theta_s = \pi$, or equivalently $z = 0$ and $z = \infty$, corresponding to $\mathcal{CS}^2$'s north and south poles, respectively.

Now consider the finite superrotation with the same meromorphic function as the uniformisation map in eq. (2.4),

$$Z(z) = \left( \frac{z - z_1}{z - z_2} \right)^{1/n}. \tag{3.17}$$

Like the superrotation with $Z(z) = z^K$, a superrotation with the $Z(z)$ in eq. (3.17) converts Minkowski spacetime to a spacetime with an uncharged, straight cosmic string, but now $K = 1/n$ and the cosmic string's endpoints reach $\mathcal{CS}^2$ at $z = z_1$ and $z = z_2$. To see why, observe that we can go from $Z(z) = z^K$ to the $Z(z)$ in eq. (3.17) by choosing $K = 1/n$ and sending

$$z \to \frac{z - z_1}{z - z_2}, \tag{3.18}$$

which is an $SL(2, \mathbb{C})$ transformation of the form in eq. (3.16), with $a = c \neq 0$, $b/a = -z_1$, and $d/a = -z_2$, such that $ad - bc = 1$ implies $z_1 - z_2 = 1/(ac)$. In other words, we can start with a cosmic string with $K = 1/n$ extended along $x_3$, which intersects $\mathcal{CS}^2$ at $z = 0$ and $z = \infty$, and then perform a rotation and/or Lorentz boost to obtain a cosmic string which intersects $\mathcal{CS}^2$ at $z = z_1$ and $z = z_2$, with $z_1 - z_2 = 1/(ac)$. Equating $K = 1/n$ and $K = 1 - 4\mu G_N$, we find that such a cosmic string has tension

$$\mu = \frac{n-1}{n} \frac{1}{4 G_N}. \tag{3.19}$$

Plugging the $Z(z)$ in eq. (3.17) into eq. (3.14) gives the metric of the spacetime with the cosmic string in retarded Newman–Unti gauge,

$$ds^2 = -du^2 - 2du\,dr + \left[ \frac{4r^2}{(1+z\bar{z})^2} + u^2 \frac{(n^2-1)^2}{16n^4} \frac{|z_2 - z_1|^4}{|z-z_1|^4 |z-z_2|^4} (1+z\bar{z})^2 \right] dz\,d\bar{z}$$
$$- ur \frac{(n^2-1)}{2n^2} \left[ \frac{(z_2 - z_1)^2}{(z-z_1)^2(z-z_2)^2} dz^2 + \frac{(\bar{z}_2 - \bar{z}_1)^2}{(\bar{z}-\bar{z}_1)^2(\bar{z}-\bar{z}_2)^2} d\bar{z}^2 \right]. \tag{3.20}$$

The metric in eq. (3.20) has a conical singularity in the bulk, of deficit/excess angle $2\pi(1 - n^{-1})$, while $\mathcal{CS}^2$ still has the round metric, $4 dz\,d\bar{z}/(1 + z\bar{z})^2$, as with any finite superrotation. We thus identify the bulk theory with the metric in eq. (3.20) as the holographic dual of the CCFT on a *single sheet* of the replica manifold, or in other words on the complex $z$-plane of sec. 2, and the cosmic string as the holographic dual of the twist fields. By implication, we identify the bulk theory with the flat slicing metric in eq. (3.6) as the holographic dual of the CCFT on the uniformised complex $w$-plane. What, then,

is the holographic dual of the CCFT on the replica manifold, $\mathcal{M}_n = \mathcal{CS}_n^2$? As explained in detail for example in ref. [71], in the metric in eq. (3.20) we can extend the angular coordinate wrapping around the cosmic string to remove the conical singularity in the bulk, transforming the bulk metric to Minkowski. However, doing so produces conical singularities on the celestial sphere at $z = z_1$ and $z = z_2$ with total angle $2\pi n$, matching the expected conical singularities of the replica manifold. We therefore identify the bulk theory with that metric as the holographic dual of the CCFT on $\mathcal{M}_n = \mathcal{CS}_n^2$.

Our story so far parallels that of Cardy and Calabrese, as reviewed in sec. 2. Our story also closely parallels that in AdS/CFT with $d = 2$. In that case, we can compute the bulk theory's partition function either using an asymptotically-AdS$_3$ spacetime with a cosmic string, with precisely the same tension as in eq. (3.19), dual to the CFT on a single sheet of the replica manifold [70], or in AdS$_3$ in Poincaré slicing coordinates with a non-trivial cutoff surface, dual to the CFT on the uniformised plane [98]. In the latter case, the non-trivial cutoff surface implements the conformal factor in eq. (2.5). However, despite these similarities, some subtleties appear to be specific to asymptotically flat spacetimes, as we will see in our explicit calculation of EREs in the next subsection.

## 3.2   Setup of holographic calculation

In this section we use the results of sec. 3.1 to compute $S_{\ell,n}$, the EREs of a single interval of length $\ell$, in the $d = 2$ CCFT. We will find the form expected for a $d = 2$ CFT on a round sphere, eq. (2.17), with the central charge in eq. (1.5).

As mentioned in sec. 1.3, we follow refs. [15, 16, 72], and assume that the CCFT and bulk partition functions are equal,

$$Z_{\text{CCFT}}(\mathcal{M}_n) \stackrel{!}{=} Z_{\text{grav}}(n), \tag{3.21}$$

where $Z_{\text{CCFT}}(\mathcal{M}_n)$ denotes the CCFT partition function on the replica manifold $\mathcal{M}_n$ and $Z_{\text{grav}}(n)$ denotes the bulk theory's partition function on a spacetime with $\mathcal{M}_n$ as its "celestial sphere." As also mentioned in sec. 1.3, for practical purposes, to compute $Z_{\text{grav}}(n)$ we take a semiclassical limit, as follows. We assume that the bulk theory has a classical action, $S_{\text{grav}}$, consisting of the Einstein–Hilbert action, possibly plus minimally-coupled matter fields, and boundary terms that guarantee a well-posed variational principle. Denoting by $S_{\text{grav}}^{\star}(n)$ the bulk theory's action evaluated on the solution with replica index $n$, the semiclassical limit is the leading saddle point approximation,

$$Z_{\text{grav}}(n) \approx e^{iS_{\text{grav}}^{\star}(n)}. \tag{3.22}$$

Plugging eq. (3.22) into the holographic relation eq. (3.21) gives $Z_{\text{CCFT}}(\mathcal{M}_n) \approx e^{iS_{\text{grav}}^{\star}(n)}$, and plugging that into eq. (2.3) gives for the EREs

$$S_{\mathcal{A},n} = \frac{i}{1-n} \left[ S_{\text{grav}}^{\star}(n) - n S_{\text{grav}}^{\star}(1) \right]. \tag{3.23}$$

To compute $S_{\mathcal{A},n}$ holographically, using the bulk theory, we thus need to compute $S_{\text{grav}}^{\star}(n)$. We will do so using a method that closely follows that in AdS/CFT [70, 109]. We start by assuming that the replica symmetry is unbroken, which allows us to write $S_{\text{grav}}^{\star}(n) = n\hat{S}_{\text{grav}}^{\star}(n)$, where we think of $\hat{S}_{\text{grav}}^{\star}(n)$ as the on-shell action of the bulk dual to a single sheet of the replica manifold. Eq. (3.23) then becomes

$$S_{\mathcal{A},n} = \frac{in}{1-n} \left[ \hat{S}_{\text{grav}}^{\star}(n) - \hat{S}_{\text{grav}}^{\star}(1) \right]. \tag{3.24}$$

To compute $S_{\mathcal{A},n}$ holographically, we now need to compute $\hat{S}^{\star}_{\mathrm{grav}}(n)$. Actually, because in the replica trick we analytically continue in $n$, eq. (3.24) implies the following differential equation for the EREs [70],

$$\partial_n \left( \frac{n-1}{n} S_{\mathcal{A},n} \right) = -i\partial_n \hat{S}^{\star}_{\mathrm{grav}}(n). \tag{3.25}$$

To compute $S_{\mathcal{A},n}$ holographically, we now need to compute $\partial_n \hat{S}^{\star}_{\mathrm{grav}}(n)$ and then integrate in $n$, with the boundary condition that the $n \to 1$ limit is finite in $n$.

To compute $\partial_n \hat{S}^{\star}_{\mathrm{grav}}(n)$, we need a bulk spacetime metric describing a conical singularity of opening angle $2\pi/n$ for each twist field, that is, we need a bulk spacetime metric describing a cosmic string. In principle, we can write the cosmic string metric using any of the coordinates reviewed in sec. 3.1, however we will start with coordinates simpler than any in sec. 3.1, namely cylindrical coordinates $(t, a, b, \phi)$, where $t$ is a time coordinate, $a$ is the coordinate along the cosmic string's axis, and $b$ and $\phi$ are the radial and angular coordinates, respectively, in the plane perpendicular to the cosmic string. The cosmic string metric then has the form

$$\mathrm{d}s^2 = -\mathrm{d}t^2 + \mathrm{d}a^2 + \mathrm{d}b^2 + \frac{b^2}{n^2}\mathrm{d}\phi^2. \tag{3.26}$$

If we chose $\phi \sim \phi + 2\pi n$, then eq. (3.26) would be the metric of Minkowski spacetime. We choose $\phi \sim \phi + 2\pi$, so that our desired conical singularity appears at $b = 0$. We need to regularise the conical singularity. We do so simply by cutting out a small tube of radius $\delta$ around the conical singularity, that is, we restrict to $b \geq \delta$, such that the near-cosmic string limit is $\delta \to 0$. Treating an infinitesimal change of $n$ as a small variation of the metric, $\hat{S}^{\star}_{\mathrm{grav}}(n)$ will change only by boundary terms. Imposing boundary conditions that the bulk metric is fixed at the holographic boundary, the only boundary contributions come from the tube regularising the conical singularity, $b = \delta$, of the form [70, 109]

$$\partial_n \hat{S}^{\star}_{\mathrm{grav}}(n) = -\frac{1}{16\pi G_N} \int_{b=\delta} \mathrm{d}^3\vec{x}\sqrt{-\gamma}\, N^\mu \left( \nabla^\nu \partial_n g_{\mu\nu} - g^{\nu\rho}\nabla_\mu \partial_n g_{\nu\rho} \right), \tag{3.27}$$

where $\vec{x}$ denotes the coordinates along the regulator tube, $\gamma$ denotes the determinant of the regulator tube's induced metric, $N^\mu$ with $\mu = 1, 2, \ldots, d$ denotes the unit vector normal to the regulator tube, pointing out from the tube into the bulk spacetime, $g_{\mu\nu}$ denotes the bulk spacetime metric, and $\nabla^\mu$ denotes $g_{\mu\nu}$'s Levi-Civita connection.[11] Plugging the metric in eq. (3.26) into eq. (3.27) gives

$$\partial_n \hat{S}^{\star}_{\mathrm{grav}}(n) = -\frac{1}{n^2}\frac{1}{8\pi G_{\mathrm{N}}}\frac{A_\delta}{\delta}, \tag{3.28}$$

where $A_\delta$ denotes the surface area of the regulator tube. For a straight cylinder of radius $\delta$ in flat space, $A_\delta \propto \delta$. If such a cylinder has finite length, then $A_\delta/\delta$ is finite in the limit $\delta \to 0$. We may thus be tempted to take the $\delta \to 0$ limit of eq. (3.28), which, after performing the integral over $\phi$, would give

$$\partial_n \hat{S}^{\star}_{\mathrm{grav}}(n) \overset{?}{=} -\frac{1}{n^2}\frac{A_{\mathrm{string}}}{4G_{\mathrm{N}}}, \tag{3.29}$$

---

[11]We assume that boundary terms on any cutoff surfaces do not contribute to the right-hand side of eq. (3.27). This should be guaranteed by the fact that the metrics with and without conical singularities obey the same boundary conditions [109], although that statement is difficult to make precise without a well-developed holographic renormalisation procedure for null boundaries.

where $A_{\text{string}}$ is the area of the cosmic string's worldsheet. Indeed, that is precisely the procedure in the analogous calculation in AdS/CFT [70]. However, we must be more careful. Our regulator tube is infinite in extent, and in particular reaches $\mathcal{I}^{\pm}$, so we need another cutoff surface somewhere near $\mathcal{I}^{\pm}$. Crucially, the near-string and near-$\mathcal{I}^{\pm}$ limits do not commute [71], so we should maintain non-zero $\delta$ until we have properly implemented this latter cutoff. Indeed, if we naïvely use eq. (3.29) then we find that $S_{\ell,n}$ is independent of $n$. Such $n$-independence can occur in principle, an example being "fixed-area" states [110, 111]. However, in what follows we will use three different holographic methods to obtain the EREs, and in each case we find $n$ dependence of the usual form in eq. (2.17).[12] In the next subsection we present the first of these three methods, which uses the results of sec. 3.1 for the superrotation dual to the uniformisation map.

All of our statements in this section generalise straightforwardly to other $\mathcal{A}$, or equivalently, to cases with more than two twist fields. What changes in these more general cases? The essential change is that the bulk spacetime will have more than one cosmic string. As a result, formally eqs. (3.21) to (3.25) will remain the same, but eq. (3.26) will only be the form of the metric near each cosmic string, and eqs. (3.27) to (3.29) will involve sums over all of the cosmic strings in the bulk spacetime.

## 3.3 Holographic calculation

In this subsection we evaluate $\partial_n \hat{S}^{\star}_{\text{grav}}(n)$ in eq. (3.27). As mentioned above, we will need multiple cutoffs in addition to the near-string cutoff $\delta$. To implement these cutoffs, we change from the cylindrical coordinates in eq. (3.26) to coordinates $(U, V, W, \overline{W})$ similar to the flat slicing coordinates of eq. (3.5), namely we define

$$U \equiv \frac{t^2 - a^2 - b^2}{t+a}, \qquad V \equiv t+a, \qquad W \equiv \frac{b}{|t+a|}. \tag{3.30}$$

The metric in eq. (3.26) becomes in these coordinates

$$\mathrm{d}s^2 = -\mathrm{d}U\,\mathrm{d}V + V^2 \left(\mathrm{d}W^2 + \frac{W^2}{n^2}\mathrm{d}\phi^2\right), \tag{3.31}$$

which is in fact equivalent to the metric in flat slicing coordinates, eq. (3.6), but with $w = We^{i\phi/n}$. In other words, we can obtain the metric in eq. (3.31) by a $\mathbb{Z}_n$ orbifold of the flat slicing metric in eq. (3.6).

We have introduced many different coordinate systems and metrics, so in fig. 2 we summarise the relationships between three of the most important. First, at the top of fig. 2, is the generic superrotated metric in eq. (3.14), in $(u, r, z, \bar{z})$ coordinates, and in which we imagine using the superrotation dual to the uniformisation map, obtaining the metric in eq. (3.20). As mentioned below eq. (3.20), we can remove the bulk conical singularity at the cost of introducing conical singularities on the celestial sphere. We identify the latter spacetime as the holographic dual to the $d = 2$ CCFT on the replica manifold, as shown in the top row of fig. 2. The middle row of fig. 2 shows the flat slicing metric of eq. (3.6), in $(U, V, w, \bar{w})$ coordinates, which in this context we imagine obtaining from the superrotated metric via the *inverse* uniformisation map. The conical

---

[12]Moreover, an alternative prescription to evaluate $\partial_n \hat{S}^{\star}_{\text{grav}}$ in eq. (3.25) appears in ref. [112], which does not require the cutoff $\delta$. The prescription of ref. [112] involves cutting open the bulk spacetime and replacing the conical singularity with a corner where the two boundaries created by the cut meet. The only non-zero contribution to $\hat{S}^{\star}_{\text{grav}}(n)$, for integer $n$, then comes from the variation of a Hayward corner term, with no need for the cutoff $\delta$. We believe the prescription of ref. [112] would also produce the EREs in eq. (2.17), with the usual $n$ dependence, although we have not checked this explicitly.

singularities on the celestial sphere map to singularities at $w = 0$ and $w = \infty$, encoded in the singular conformal factor of the sphere's metric, eq. (2.13). We will reproduce this singular conformal factor holographically below: see eq. (3.39). The bottom row of fig. 2 shows the metric in eq. (3.31), obtained from the flat slicing metric by a $\mathbb{Z}_n$ orbifold. The celestial sphere is mapped to a single segment with conical singularities at $W = 0$ and $W = \infty$, again encoded in a singular conformal factor of the celestial sphere's metric.

Plugging the metric in eq. (3.31) into eq. (3.27) and performing the trivial integration over $\phi$ gives

$$\partial_n \hat{S}^{\star}_{\text{grav}}(n) = -\frac{1}{n^2} \frac{1}{8G_{\text{N}}} \int dU \, dV. \tag{3.32}$$

The integral in eq. (3.32) diverges because $U \in (-\infty, \infty)$ and $V \in (-\infty, \infty)$. More intuitively, the integral in eq. (3.32) diverges because the cosmic string reaches $\mathcal{I}^{\pm}$, and also because it is static, and so extends over all time. Each of these limits requires a cutoff. Such behavior is not unique to a cosmic string spacetime. Indeed, generically celestial holography requires two cutoffs for practically any calculation because the celestial sphere is codimension two with respect to the bulk—in contrast to AdS/CFT, which requires only one cutoff because the AdS boundary is codimension one.

We cutoff $U$ by imposing

$$|UV| \leq L^2, \tag{3.33}$$

with some large value of $L$. In other words, our $U$ cutoff depends on $V$ as $|U| \leq L^2/|V|$. We choose this primarily for simplicity, especially when comparing to other cases. For instance, the coordinate transformations in sec. (3.1) and in this section preserve $UV = uv$, hence this cutoff is easy to translate between coordinate systems. In fact, our $U$ cutoff is identical to that of ref. [42], which was a cutoff on Milne time, and is a special case of the cutoffs in ref. [73]: in terms of their cutoffs $\eta_{\infty}$ and $r_{\infty}$, we choose $\eta_{\infty} = L$ and $r_{\infty} = L$.

Our cutoff on $V$ is more complicated, and requires two ingredients. The first ingredient is a near-string cutoff. In the coordinates of eq. (3.30), the cosmic string is at $W = 0$. We thus cut out a tube around the cosmic string, that is, we impose $W > W_0$ for some small function $W_0$. We can in principle choose any $W_0$. We will see below that the physical information contained in $S_{\ell,n}$ depends only on $W_0$'s behaviour near $\mathcal{I}^{\pm}$. To be concrete, we will use our cylindrical cutoff $b \geq \delta$, which via eq. (3.30) gives

$$W_0(V) = \delta |V|^{-1}. \tag{3.34}$$

Our choice of $W_0$ has several advantages, as we will see. For example, the $|V|^{-1}$ in eq. (3.34) will cancel against the $V^2$ in the metric of eq. (3.31). Without that, our near-string cutoff would shrink to zero radius at $V = 0$ and expand to infinite radius at $\mathcal{I}^{\pm}$.

The second ingredient for our $V$ cutoff is a near-$\mathcal{I}^{\pm}$ cutoff. To define this, we return to the coordinates $(u, r, z, \bar{z})$ of the superrotated metric in eq. (3.14), and impose

$$r \leq r_c, \tag{3.35}$$

for some large value of $r_c$. Using eq. (3.15a) to translate to the flat slicing coordinates of eq. (3.31), the cutoff in eq. (3.35) becomes

$$V \leq \frac{2r_c}{(1 + z\bar{z})\sqrt{Z'\bar{Z}'}}. \tag{3.36}$$

For the superrotation dual to the uniformisation map, eq. (3.17), eq. (3.36) becomes

$$V \leq \frac{2n|z_2 - z_1||w|^{n-1}r_c}{|w^n - 1|^2 + |z_2 w^n - z_1|^2}, \tag{3.37}$$

| **Bulk spacetime** | **Celestial sphere** |
|---|---|

**Superrotated spacetime:**

$$\mathrm{d}s^2 = -\mathrm{d}u^2 - 2\mathrm{d}u\,\mathrm{d}r - ru\big(\{Z,z\}\mathrm{d}z^2 + \text{c.c.}\big)$$
$$+ \Big[\frac{4r^2}{(1+z\bar z)^2} + \frac{u^2(1+z\bar z)^2}{4}|\{Z,z\}|^2\Big]\mathrm{d}z\,\mathrm{d}\bar z$$

No bulk conical singularity

**Replica manifold:**

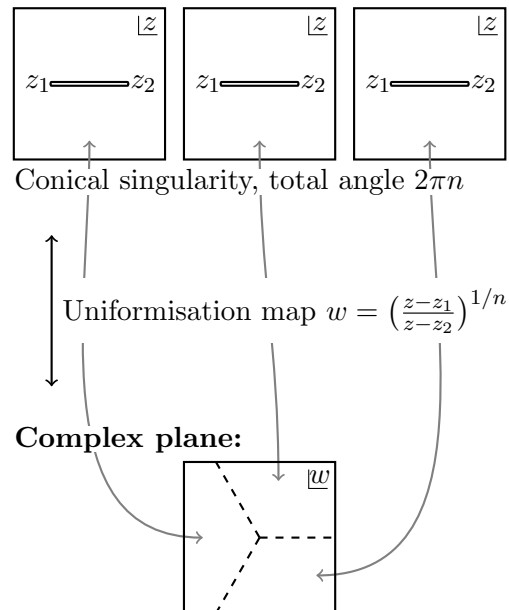

Conical singularity, total angle $2\pi n$

Superrotation $Z(z) = \left(\frac{z-z_1}{z-z_2}\right)^{1/n}$

Uniformisation map $w = \left(\frac{z-z_1}{z-z_2}\right)^{1/n}$

**Minkowski space in flat slicing:**

$$\mathrm{d}s^2 = -\mathrm{d}U\,\mathrm{d}V + V^2\mathrm{d}w\,\mathrm{d}\bar w$$

No bulk conical singularity

Singular cutoff surface

**Complex plane:**

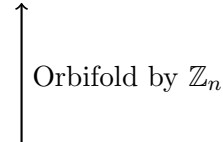

Singular conformal factor

Orbifold by $\mathbb{Z}_n$

Orbifold by replica symmetry

**Cosmic string:**

$$\mathrm{d}s^2 = -\mathrm{d}U\,\mathrm{d}V + V^2\left(\mathrm{d}W^2 + \frac{W^2}{n^2}\mathrm{d}\phi^2\right)$$

Bulk conical singularity, total angle $2\pi/n$

Singular cutoff surface

**Segment of complex plane:**

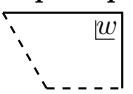

Conical singularity, total angle $2\pi/n$

Singular conformal factor

Figure 2: The relationships among the three most important geometries in our holographic calculation of single interval ERE's in $d = 2$ CCFT. The top row on the left shows the superrotated metric of eq. (3.14), in $(u, r, z, \bar z)$ coordinates. The top row on the right shows that, for the superrotation dual to the uniformisation map in eq. (3.17), we can remove the bulk conical singularity at the cost of producing singularities on the celestial sphere, which we then identify as the replica manifold, as mentioned below eq. (3.20). The middle row on the left shows the Minkowski metric in flat slicing, eq. (3.6), in $(U, V, w, \bar w)$ coordinates. The middle row on the right shows how, if we obtain this from the first row via the inverse uniformisation map, then the singularities on the celestial sphere map to $w = 0$ and $w = \infty$, as encoded in the sphere's singular conformal factor in eq. (2.13). The bottom row on the left shows the metric in eq. (3.31), obtained from that in the middle row by a $\mathbb{Z}_n$ orbifold. The bottom row on the right shows how this orbifold maps the conical singularities to $W = 0$ and $W = \infty$, as again encoded in the celestial sphere's singular conformal factor.

where again $w = We^{i\phi/n}$.

The near-$\mathcal{I}^\pm$ cutoff in eq. (3.37) is the only place in this section that we use a super-rotation, and specifically the superrotated metric of eq. (3.14). As added value, we can also use this near-$\mathcal{I}^\pm$ cutoff to recover the metric of the uniformised sphere, eq. (2.13), as follows. The induced metric on the near-$\mathcal{I}^\pm$ cutoff in eq. (3.37) is

$$\mathrm{d}s^2_{r_c} = r_c^2 \frac{4n^2|z_2 - z_1|^2|w|^{2(n-1)}}{(|w^n - 1|^2 + |z_2 w^n - z_1|^2)^2} \mathrm{d}w\, \mathrm{d}\bar{w} + \mathcal{O}(r_c). \tag{3.38}$$

Extracting the metric on the celestial sphere as $\mathrm{d}s^2_{\mathcal{CS}^2} \equiv \lim_{r_c \to \infty}(\mathrm{d}s^2_{r_c}/r_c^2)$ then gives

$$\mathrm{d}s^2_{\mathcal{CS}^2} = \frac{4n^2|z_2 - z_1|^2|w|^{2(n-1)}}{(|w^n - 1|^2 + |z_2 w^n - z_1|^2)^2} \mathrm{d}w\, \mathrm{d}\bar{w}, \tag{3.39}$$

which is precisely the metric of a uniformised sphere, eq. (2.13), with unit radius, $R = 1$. Of course this had to be the case: in the coordinate system of eq. (3.20), the induced metric on the cutoff surface is $r_c^2$ times that of a unit-radius round sphere, and from the asymptotics of the superrotation in eq. (3.15), at large $r$ the transformation from the $z$ to the $w$ coordinate system is the uniformisation map.

Fig. 3 depicts the near-string cutoff in eq. (3.34) and the near-$\mathcal{I}^\pm$ cutoff in eq. (3.37), in both the $(U, V, W, \overline{W})$ coordinates (fig. 3a) and in retarded Newman–Unti coordinates (fig. 3b). The latter is obtained via the coordinate change in eq. (3.5), and hopefully provides a more intuitive picture of these cutoffs.

By combining the near-string and near-$\mathcal{I}^\pm$ cutoffs, we obtain cutoffs on $V$, as follows. Fig. 3a shows that the near-$\mathcal{I}^\pm$ cutoff in eq. (3.37) intersects the near-string cutoff $W_0(V) = \delta|V|^{-1}$ twice, at small and large $V$. We denote these intersections as $V_{\min}$ and $V_{\max}$, respectively. These two intersections occur close to the two endpoints of the string on $\mathcal{I}^\pm$, at $W = 0$ and $W = \infty$. Plugging $W = \delta|V|^{-1}$ into eq. (3.37) then defines our cutoff on $V$: we integrate over the range $V_{\min} \leq |V| \leq V_{\max}$, where

$$V_{\min} \equiv \left(\frac{1 + |z_2|^2}{2n|z_2 - z_1|} \frac{\delta^{n+1}}{r_c}\right)^{1/n}, \qquad V_{\max} \equiv \left(\frac{2n|z_2 - z_1|}{1 + |z_1|^2} \delta^{n-1} r_c\right)^{1/n}. \tag{3.40}$$

Eq. (3.40) allow us to relate the near-string cutoff $\delta$ to the CCFT near-twist field cutoff $\epsilon$, reviewed in sec. 2, as follows. The $V_{\min}$ and $V_{\max}$ in eq. (3.40) correspond to

$$W_0(V_{\min}) = \left(\frac{1 + |z_2|^2}{2|z_2 - z_1|} \frac{\delta}{nr_c}\right)^{-1/n}, \qquad W_0(V_{\max}) = \left(\frac{1 + |z_1|^2}{2|z_2 - z_1|} \frac{\delta}{nr_c}\right)^{1/n}, \tag{3.41}$$

which provide large- and small-$W$ cutoffs, respectively. To determine $\epsilon$, we calculate the distance between a twist field insertion and the corresponding cutoff from eq. (3.41). Under the uniformisation map the twist field insertion at $z_1$ is mapped to $W = 0$. Using the celestial sphere metric in eq. (3.39), the distance between the twist field insertion at $W = 0$ and the small-$W$ cutoff at $W = W_0(V_{\max}) \ll 1$ is

$$\frac{2n|z_2 - z_1|}{1 + |z_1|^2} \int_0^{W_0(V_{\max})} \mathrm{d}W\, W^{n-1} = \frac{2|z_2 - z_1|}{1 + |z_1|^2} W_0(V_{\max})^n = \frac{\delta}{nr_c}. \tag{3.42a}$$

Analogously, under the uniformisation map the twist field insertion at $z_2$ is mapped to $W = \infty$, and the distance between the large-$W$ cutoff at $W = W_0(V_{\min}) \gg 1$ and the twist field insertion at $W = \infty$ is

$$\frac{2n|z_2 - z_1|}{1 + |z_2|^2} \int_{W_0(V_{\min})}^\infty \mathrm{d}W\, W^{-(n+1)} = \frac{2|z_2 - z_1|}{1 + |z_2|^2} W_0(V_{\min})^{-n} = \frac{\delta}{nr_c}. \tag{3.42b}$$

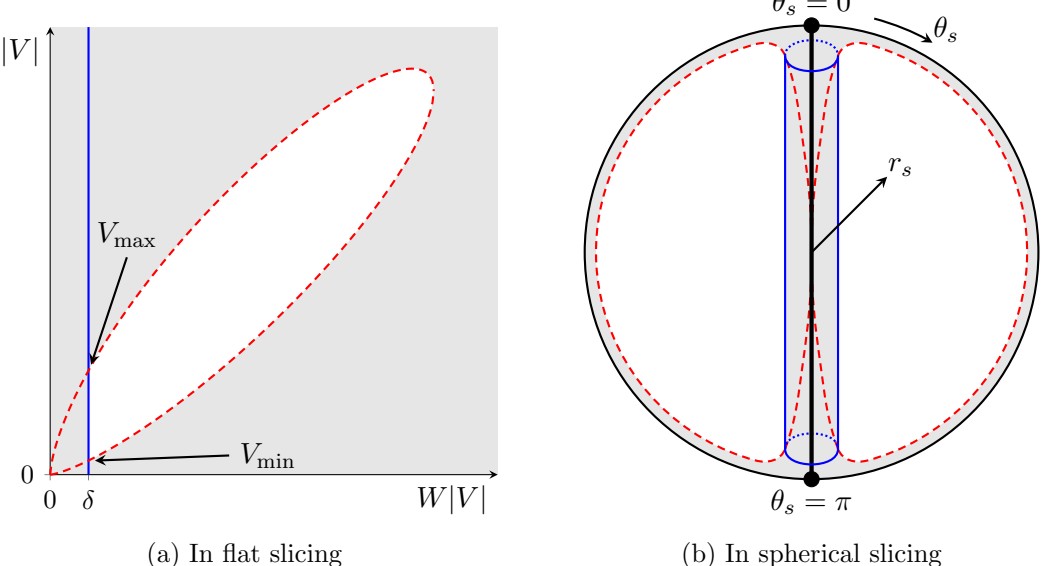

(a) In flat slicing | (b) In spherical slicing

Figure 3: **(a)** Cartoon of the cutoff surfaces in the $(U, V, W, \overline{W})$ coordinates at constant $U$, in the plane of $W|V|$ and $|V|$. The cosmic string/conical singularity is the vertical line at $W = 0$, or equivalently $W|V| = 0$. The vertical blue line is our near-string cutoff in eq. (3.34), $W|V| = \delta$. The dashed red curve is our near-$\mathcal{I}^{\pm}$ cutoff in eq. (3.36). We have shaded grey the regions of spacetime removed by the cutoffs. The two cutoff surfaces meet at two different values of $V$, labeled $V_{\min}$ and $V_{\max}$, which set the limits on the $V$ integration in eq. (3.32). **(b)** Cartoon of the cutoff surfaces in the retarded Eddington–Finkelstein coordinates $(u_s, r_s, \theta_s, \phi_s)$, obtained from $(U, V, W, \overline{W})$ coordinates via eq. (3.5). The black circle represents the celestial sphere at $r_s \to \infty$ at fixed $u$. The thick, vertical, solid black line represents the cosmic string/conical singularity. The blue cylinder surrounding the cosmic string represents the near-string cutoff, located at $r_s \sin \theta_s = \delta$ in these coordinates. The dashed red curve represents the near-$\mathcal{I}^{\pm}$ cutoff.

We thus identify the holographic dual of the CCFT's near-twist field cutoff as

$$\epsilon \equiv \frac{\delta}{n r_c}, \tag{3.43}$$

which is small both because $\delta$ is small and because $r_c$ is large. We will choose $\delta$ and $r_c$ such that $\epsilon$ is independent of $n$, as in Cardy and Calabrese's calculation of sec. 2. However, we will argue below that choosing $\delta$ and $r_c$ to be independent of $n$, so that eq. (3.43) then implies $\epsilon \propto 1/n$, does not affect the physical information contained in $S_{\ell,n}$.

Having fixed the cutoffs on $U$ and $V$, we now evaluate the integrals in eq. (3.32). The

regions with positive and negative $V$ contribute equally, so that

$$\partial_n \hat{S}_{\text{grav}}^{\star}(n) = -\frac{1}{n^2}\frac{1}{4G_N}\int_{V_{\min}}^{V_{\max}} dV \int_{-L^2/V}^{L^2/V} dU \tag{3.44a}$$

$$= -\frac{1}{n^2}\frac{1}{4G_N}\int_{V_{\min}}^{V_{\max}} dV \frac{2L^2}{V} \tag{3.44b}$$

$$= -\frac{1}{n^2}\frac{L^2}{2G_N}\log\left(V_{\max}/V_{\min}\right) \tag{3.44c}$$

$$= -\frac{1}{n^3}\frac{L^2}{G_N}\log\left(\frac{2}{\epsilon}\frac{|z_2 - z_1|}{\sqrt{(1+|z_1|^2)(1+|z_2|^2)}}\right). \tag{3.44d}$$

Writing eq. (3.44d) in terms of $\ell$, the great circle distance on the unit-radius sphere in eq. (2.15), then gives

$$\partial_n \hat{S}_{\text{grav}}^{\star}(n) = -\frac{1}{n^3}\frac{L^2}{G_N}\log\left[\frac{2}{\epsilon}\sin\left(\frac{\ell}{2}\right)\right]. \tag{3.45}$$

We now plug eq. (3.45) into eq. (3.25) and solve the resulting differential equation with the initial condition that the ERE has a finite $n \to 1$ limit. Choosing the cutoffs $\delta$ and $r_c$ such that $\epsilon$ is independent of $n$, as mentioned above, the solution is

$$S_{\ell,n} = \left(1 + \frac{1}{n}\right)\frac{iL^2}{2G_N}\log\left[\frac{2}{\epsilon}\sin\left(\frac{\ell}{2}\right)\right], \tag{3.46}$$

which is the main result of this section. Eq. (3.46) has the form of the single-interval EREs of a CFT on a sphere of unit radius in eq. (2.17), where our regularisation scheme has set the scheme-dependent constants to $k_n = 0$, and where we identify the central charge as

$$c = i\frac{3L^2}{G_N}, \tag{3.47}$$

whose interpretation we discussed in sec. 1.4.

As a check, the $n \to 1$ limit of eq. (3.46) gives the EE,

$$S_\ell = \frac{iL^2}{G_N}\log\left[\frac{2}{\epsilon}\sin\left(\frac{\ell}{2}\right)\right], \tag{3.48}$$

which agrees with that of ref. [73], computed using the wedge holography formalism [113–115] and the Ryu–Takayanagi prescription [116, 117]. To be specific, our result for the EE in eq. (3.48) is equal to the sum of contributions to EE from the different wedges of Minkowski spacetime computed in ref. [73], with appropriate identification of cutoffs (including $\eta_\infty = L$ and $r_\infty = L$, as mentioned below eq. (3.33)).

We end this section with three crucial questions about our cutoffs. First, how does our result, eq. (3.46), depend on our choice of the near-string cutoff $W_0$ in eq. (3.34)? Second, how does our result depend on the possible $n$ dependence of $\epsilon$ in eq. (3.43)? Third, if we used coordinates different from $(U, V, W, \overline{W})$ in eq. (3.30), then what cutoffs would we need? We will address each of these in turn.

How does our result for $S_{\ell,n}$ in eq. (3.46) depend on our choice of $W_0$? The physical information contained in our $S_{\ell,n}$, namely the logarithm, depends only on the fact that $W_0 \propto |V|^{-1}$ near $\mathcal{I}^\pm$. We see so explicitly in the integral over $V$ in eq. (3.44b), which depends only on $V_{\min}$ and $V_{\min}$, defined by the intersection of $W_0$ with the near-$\mathcal{I}^\pm$ cutoff. Moreover, eqs. (3.42a) and (3.42b) are the same only because $W_0 \propto |V|^{-1}$ near $\mathcal{I}^\pm$, thus

allowing us to identify the same cutoff, $\epsilon$ near each twist field. In other words, any other behaviour of $W_0$ near $\mathcal{I}^\pm$ would produce two different cutoffs near the two twist fields. In short, $W_0$ can have any behaviour away from $\mathcal{I}^\pm$, for example $W_0$ could depend on both $U$ and $V$, or not be $\propto |V|^{-1}$ elsewhere, but as long as $W_0 \propto |V|^{-1}$ near $\mathcal{I}^\pm$, the physical information in eq. (3.46) will be unchanged.

How does our result depend on the possible $n$ dependence of $\epsilon$ in eq. (3.43)? A constant rescaling of the near-string cutoff $\delta$ or the near-$\mathcal{I}^\pm$ cutoff $r_c$ will not change the physical information contained in $S_{\ell,n}$ in eq. (3.46), but may change the non-universal constants, $k_n$. This includes rescalings that depend on $n$. For instance, if we change our regularisation procedure so that $\delta$ and $r_c$ are independent of $n$, and hence eq. (3.43) implies $\epsilon \propto 1/n$, then the logarithmic term in eq. (3.46) is unchanged, but now

$$k_n = \frac{iL^2}{4G_N}\left(1 + \frac{1}{n} - \frac{\log n}{n(n-1)}\right).\tag{3.49}$$

If we used coordinates different from $(U, V, W, \overline{W})$ in eq. (3.30), then what cutoffs would we need? As mentioned below eq. (3.32), practically any calculation in celestial holography will require two cutoffs, essentially because the celestial sphere is codimension two with respect to the bulk. For example, if we had used the superrotated metric in eq. (3.20), with Newman–Unti coordinates $(u, r, z, \bar{z})$, then we would have needed cutoffs on $u$ and $r$. Crucially, however, we would *not* have needed a near-string cutoff, $\delta$: we could have sent $\delta \to 0$ from the start, in which case $A_\delta/\delta$ in eq. (3.27) would have become the cosmic string worldsheet's area. The cost of using the coordinates in eq. (3.20) would have been a more complicated integral for that area. Instead, we chose the coordinates $(U, V, W, \overline{W})$ to obtain the very simple integral in eq. (3.32). The cost of doing so was a near-string cutoff $\delta$: from fig. 3 we see that if we removed the near-string cutoff, either by moving the blue line all the way to the left in fig. 3a or shrinking the cylinder to zero radius in fig. 3b, then the near-$\mathcal{I}^\pm$ cutoff excises almost all of the string except the points at $V = 0$ in flat slicing, or equivalently $r_s = 0$ in spherical slicing. The need to maintain finite $\delta$ during the calculation arises because the coordinate transformation between the Newman–Unti gauge metric in eq. (3.20) and the flat slicing metric in eq. (3.31) is singular at the cosmic string. The intersection of the near-string cutoff and the near-$\mathcal{I}^\pm$ cutoff provided the cutoff on $V$, which was essential to produce the dependence on $z_1$, $z_2$, $n$, and $\epsilon$ that we expect for single-interval EREs in the $d = 2$ conformal vacuum. The general lesson is: any coordinate system in which the near-$\mathcal{I}^\pm$ cutoff is singular will require a finite near-string cutoff throughout the calculation of the EREs.

In the next section we will adopt a coordinate system inspired by, though not identical to, the CHM approach to the calculation of EREs in AdS/CFT [92]. This coordinate system, which we view as a coordinate system well-adapted to the replica symmetry, is simpler than Newman–Unti gauge while also allowing us to remove the near-string cutoff by taking $\delta \to 0$ early in the calculation. For $d = 2$ we will obtain eq. (3.46), providing a check of that result. However, we adopt this coordinate system primarily to generalise our results to CCFTs with $d > 2$.

In the appendix we also derive the result in eq. (3.46) by extending the methods of ref. [73] to $n \neq 1$, which provides another check of eq. (3.46).

# 4 Holographic calculation of CCFT EREs: $d \geq 2$

In this section we use holography to calculate the EREs, $S_{\mathcal{A},n}$, of a ball shaped entangling region $\mathcal{A}$ of radius $\ell$ in the conformal vacuum of a CCFT with dimension $d \geq 2$. (For

$d = 2$, $\ell$ denotes the interval's length, not its radius.) To do so, we will make all the same assumptions as in $d = 2$, described in sec. 3.2. Most importantly, we assume the bulk theory is Einstein–Hilbert gravity, possibly plus minimally-coupled matter fields, we identify the bulk theory and CCFT partition functions, and we employ a leading saddle point approximation for the bulk theory's partition function. These assumptions lead to eq. (3.25), which we reproduce here for convenience,

$$\partial_n \left( \frac{n-1}{n} S_{\mathcal{A},n} \right) = -i \partial_n \hat{S}^\star_{\text{grav}}(n),$$

where we remind the reader that $\hat{S}^\star_{\text{grav}}(n)$ is the on-shell action of the bulk theory, evaluated on a spacetime with conical singularity of total angle $2\pi/n$, where in the near $\mathcal{I}^\pm$ limit this conical singularity approaches the boundary of $\mathcal{A}$.

We thus need a metric that locally solves the $(d + 2)$-dimensional Einstein equation and approaches the metric of a round $d$-dimensional sphere near $\mathcal{I}^\pm$, but possesses a bulk conical singularity that asymptotes to the boundary of $\mathcal{A}$ on the celestial sphere. For $d = 2$, we wrote such a metric, namely the metric of a single straight, static cosmic string, in several different coordinate systems in secs. 3.1 and 3.3. For $d > 2$, the analogous metric describes a single, flat, static cosmic *brane* of codimension two [89]. We can again write such a metric in several different coordinate systems. For example, suppose we want to follow sec. 3.3, where we used the flat-slicing-like coordinates $(U, V, W, \overline{W})$ in eqs. (3.30) and (3.31). For $d > 2$ we simply need to add flat transverse coordinates, $y^i$ with $i = 1, 2, \ldots, d - 4$, so that the relevant cosmic brane metric is

$$\mathrm{d}s^2 = -\mathrm{d}U\,\mathrm{d}V + V^2 \left( \mathrm{d}W^2 + \frac{W^2}{n^2}\mathrm{d}\phi^2 + \delta_{ij}\,\mathrm{d}y^i\mathrm{d}y^j \right), \tag{4.1}$$

with the cosmic brane at $W = 0$.

However, a key lesson from the $d = 2$ case in sec. 3.3 was that, in the coordinates of eq. (4.1), we must define $V$'s cutoff as the intersection of a near-brane cutoff with a near-$\mathcal{I}^\pm$ cutoff. In $d = 2$, we obtained the near-$\mathcal{I}^\pm$ cutoff from the superrotation dual to the uniformisation map. Indeed, in sec. 3.3 the near-$\mathcal{I}^\pm$ cutoff was the only place we used this superrotation: see eqs. (3.35) through (3.37).

For $d > 2$ finding the near-$\mathcal{I}^\pm$ cutoff will be more difficult. We would need at least the asymptotic form of a diffeomorphism that maps eq. (4.1) to a metric with a radial coordinate $r$ whose large-$r$ behaviour gives $r^2\mathrm{d}s^2_{\mathrm{S}^{d-2}}$, and such that the cosmic brane asymptotes to $\partial\mathcal{A}$ on the celestial sphere. However, a straightforward analysis shows that we cannot reach $r^2\mathrm{d}s^2_{\mathrm{S}^{d-2}}$ by a transformation that first uniformises the $(W, \phi)$ plane without affecting the periodicity of $\phi$ and then proceed to "uniformize" the higher dimensional celestial manifold [89]. These statements are consistent with CFT expectations: in $d = 2$ the conformal algebra is infinite-dimensional, and includes singular transformations that can remove isolated singularities (like the points where a cosmic string pierces the celestial sphere), while in $d > 2$ the conformal algebra is finite-dimensional, and includes only globally smooth transformations (so we have little hope to remove extended loci of singularities, like those in eq. (4.1)). We could try more general asymptotic diffeomorphisms, in the spirit of refs. [90, 91], and possibly even singular diffeomorphisms, however that would no longer be uniformisation in the sense of Cardy and Calabrese.

Fortunately, we found a different approach, using coordinates derived from the Casini–Huerta–Myers (CHM) map, first proposed for AdS/CFT [92]. The CHM map will not only provide a near-$\mathcal{I}^\pm$ cutoff, but will also provide an alternative perspective on the EREs, in terms of the CCFT in a thermal state on a hyperbolic space. We will review the CHM map for AdS/CFT in sec. 4.1. Readers familiar with CHM in AdS/CFT may *not* want to

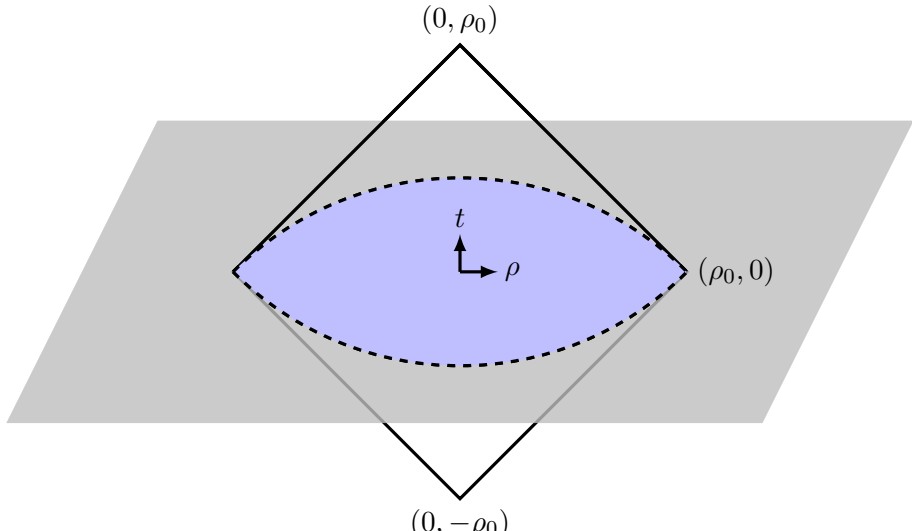

Figure 4: A depiction of the causal development, $\mathcal{D}$, of the spherical subregion $\mathcal{A}$ of Minkowski spacetime, in the coordinates of eq. (4.2), where $t$ is time, $\rho$ is the radial coordinate, and $\mathcal{A}$ has radius $\rho_0$. The grey area represents the $t = 0$ slice of Minkowski spacetime, the purple area represents $\mathcal{A}$, and the solid black lines represent $\mathcal{D}$'s outer boundary, defined by eq. (4.3).

skip sec. 4.1, since our emphasis will be different from AdS/CFT. Our strategy will then be to split $d \geq 2$ Minkowski spacetime into AdS and dS patches and employ CHM in each patch, thus obtaining the metric and cutoffs we need, and then to sum the contributions from all the patches. The details of these calculations are in sec. 4.2. Our main results are the EREs in eqs. (4.63) and (4.67).

## 4.1   Review: Casini–Huerta–Myers in AdS/CFT

In this section we review the CHM map and its application in AdS/CFT. Readers familiar with the CHM map may *not* wish to skip this section, since we will emphasise certain aspects in order to generalise more easily to celestial holography. We first review CHM's results for any CFT, and then review their results for a CFT with an AdS dual.

### 4.1.1   Casini–Huerta–Myers in CFT

CHM considered a spherical entangling region $\mathcal{A}$ in the conformal vacuum of a $d$-dimensional CFT in Minkowski spacetime. If we write the Minkowski metric in spherical polar coordinates as

$$\mathrm{d}s^2 = -\mathrm{d}t^2 + \mathrm{d}\rho^2 + \rho^2 \mathrm{d}s^2_{\mathrm{S}^{d-2}}, \tag{4.2}$$

with $t$ the time coordinate, $\rho$ the spherical radial coordinate, and $\mathrm{d}s^2_{\mathrm{S}^{d-2}}$ the metric on a round, unit $(d-2)$-dimensional sphere, $S^{d-2}$, then $\mathcal{A}$ is $\rho \leq \rho_0$ at $t = 0$. Fig. 4 depicts $\mathcal{A}$ and its causal development, $\mathcal{D}$, defined by

$$\rho_\pm \equiv \rho \pm t \leq \rho_0. \tag{4.3}$$

The modular Hamiltonian, $H$, defined from the reduced density matrix via $\rho_\mathcal{A} \equiv e^{-H}$, generates a symmetry of $\mathcal{D}$: acting with the unitary operator $e^{iHs}$, where $s$ is a dimen-

sionless real parameter, implements the modular flow

$$\rho_\pm \to \rho_0 \frac{(\rho_0 + \rho_\pm) - e^{\mp 2\pi s}(\rho_0 - \rho_\pm)}{(\rho_0 + \rho_\pm) + e^{\mp 2\pi s}(\rho_0 - \rho_\pm)}. \tag{4.4}$$

CHM found a conformal transformation that maps $\mathcal{D}$ to $\mathbb{R} \times \mathbb{H}^{d-1}$, where $\mathbb{R}$ is time and $\mathbb{H}^{d-1}$ is a $(d-1)$-dimensional hyperbolic plane. Explicitly, the CHM map is

$$t \equiv \frac{\rho_0 \sinh \tau}{\cosh \sigma + \cosh \tau}, \qquad \rho \equiv \frac{\rho_0 \sinh \sigma}{\cosh \sigma + \cosh \tau}, \tag{4.5}$$

which transforms the Minkowski metric from eq. (4.2) to

$$ds^2 = \frac{\rho_0^2}{(\cosh \sigma + \cosh \tau)^2} \left( -d\tau^2 + d\sigma^2 + \sinh^2 \sigma \, ds_{\mathbb{S}^{d-2}}^2 \right), \tag{4.6}$$

where $\tau \in (-\infty, \infty)$ is the time coordinate and $\sigma \in [0, \infty)$ is the radial coordinate of $\mathbb{H}^{d-1}$. We used a factor of $\rho_0$ to make both $\tau$ and $\sigma$ dimensionless. Using a Weyl transformation to remove the conformal factor leaves the metric in parentheses, which is indeed the metric of $\mathbb{R} \times \mathbb{H}^{d-1}$. After the CHM map, the modular flow in eq. (4.4) becomes simply time translation, $\tau \to \tau + 2\pi s$. CHM also showed that correlation functions of local operators obey a Kubo–Martin–Schwinger condition, that is, they are periodic in imaginary time, $\tau \sim \tau + 2\pi i$, showing that the CFT is in a thermal state with inverse temperature $\beta = 2\pi$.

The CHM map allows us to interpret the EE and EREs of $\mathcal{A}$ in terms of the thermal state on $\mathbb{R} \times \mathbb{H}^{d-1}$, as follows. The EE is defined in terms of $H$ as

$$S_{\rho_0} = \text{tr}(e^{-H} H). \tag{4.7}$$

The modular Hamiltonian $H$ is related to the the Hamiltonian $H_\tau$ that generates time translations on $\mathbb{R} \times \mathbb{H}^{d-1}$ by a unitary operator $U$: if $Z = \text{tr} \, e^{-\beta H_\tau}$ is the CFT's thermal partition function on $\mathbb{R} \times \mathbb{H}^{d-1}$, then

$$e^{-H} = \frac{1}{Z} U e^{-\beta H_\tau} U^\dagger, \qquad \beta = 2\pi. \tag{4.8}$$

Plugging eq. (4.8) into eq. (4.7) gives

$$S_{\rho_0} = \frac{2\pi}{Z} \text{tr} \left( e^{-2\pi H_\tau} H_\tau \right) + \log Z. \tag{4.9}$$

On the right-hand side of eq. (4.9), the first term is $\beta$ times the expectation value of the energy, while the second term is $\log Z = 2\pi F(2\pi)$, where $F(\beta)$ is the Helmholtz free energy at inverse temperature $\beta$. We thus identify the right-hand side of eq. (4.9) as the thermal entropy on $\mathbb{R} \times \mathbb{H}^{d-1}$ at inverse temperature $\beta = 2\pi$, which we denote as $S_{\text{th}}(2\pi)$. In short, the CHM map equates the CFT's conformal vacuum EE of spherical $\mathcal{A}$ with the CFT's thermal entropy on $\mathbb{R} \times \mathbb{H}^{d-1}$,

$$S_{\rho_0} = S_{\text{th}}(2\pi). \tag{4.10}$$

A similar calculation shows that the EREs may be expressed in terms of $F(\beta)$ as

$$S_{\rho_0,n} = 2\pi \frac{n}{n-1} \left[ F(2\pi n) - F(2\pi) \right]. \tag{4.11}$$

Multiplying both sides of eq. (4.11) by $n$ gives an expression for the derivative of the $n$th ERE in terms of the thermal entropy at inverse temperature $\beta = 2\pi n$,

$$n^2 \partial_n \left( \frac{n-1}{n} S_{\rho_0,n} \right) = S_{\text{th}}(2\pi n). \tag{4.12}$$

The CHM map thus allows us to express all the EREs in terms of a thermodynamic quantity on $\mathbb{R} \times \mathbb{H}^{d-1}$, namely an integral over $n$ of the thermal entropy $S_{\text{th}}(2\pi n)$.

Crucially for celestial holography, CHM's results rely only on conformal symmetry, regardless of whether the CFT is reflection positive/unitary.

### 4.1.2   Casini–Huerta–Myers in AdS

If the CFT is holographically dual to a gravity theory in AdS, then the CHM conformal transformation in eq. (4.5) may be implemented by a bulk coordinate transformation, as follows. Consider the Poincaré patch of $\text{AdS}_{d+1}$, whose metric we write as

$$\mathrm{d}s^2 = \frac{1}{y^2}\left(\mathrm{d}y^2 - \mathrm{d}t^2 + \mathrm{d}\rho^2 + \rho^2 \mathrm{d}s^2_{\mathrm{S}^{d-2}}\right), \tag{4.13}$$

where $y$ is the holographic coordinate, and we use units where the AdS curvature radius is unity. By definition, an AdS metric has a second-order pole at the conformal boundary. To extract a metric at the conformal boundary we must multiply the AdS metric by a defining function, which has a second order zero, and then take the limit towards the conformal boundary. In the coordinates of eq. (4.13), the conformal boundary is at $y = 0$. Multiplying by the defining function $y^2$ and taking $y \to 0$ then gives a metric at the boundary that is the Minkowski spacetime in spherical polar coordinates, eq. (4.2).

    In AdS/CFT, the CFT EE, $S_{\mathcal{A}}$, equals the area of the Ryu–Takayanagi minimal surface divided by $4G_{\mathrm{N}}$ [116, 117]. When $\mathcal{A}$ is a spherical sub-region $\rho \leq \rho_0$ at $t = 0$, the Ryu–Takayanagi minimal surface is simply the hemisphere $y^2 + \rho^2 = \rho_0^2$, which at the AdS boundary $y \to 0$ approaches the sphere $\rho = \rho_0$. More generally, to compute the CFT EREs in AdS/CFT we need Dong's cosmic brane [70], that is, the asymptotically locally AdS spacetime with a conical singularity in the bulk, produced by a brane with tension $\frac{n-1}{n}\frac{1}{4G_N}$ (the same as eq. (3.19)), but no singularity at the conformal boundary. We interpret such a spacetime as the bulk dual to the CFT on a single sheet of the replica manifold, with the cosmic brane as the bulk dual to the twist fields. If $A_{\mathrm{brane}}$ is the cosmic brane's area, then the EREs are given by

$$n^2 \partial_n \left(\frac{n-1}{n} S_{\mathcal{A},n}\right) = \frac{A_{\mathrm{brane}}}{4G_{\mathrm{N}}}, \tag{4.14}$$

similar to eq. (3.29). When $\mathcal{A}$ is a spherical subregion $\rho \leq \rho_0$ at $t = 0$, the CHM map gives the relevant cosmic brane metric, as we will review. In celestial holography, we will need only the asymptotic form of this cosmic brane metric in each AdS patch, to obtain the near-$\mathcal{I}^{\pm}$ cutoff in each AdS patch.

    We implement the CHM map of eq. (4.5) in AdS by defining three new bulk coordinates, $(\zeta, \tau, \sigma)$, through the coordinate transformation

$$y \equiv \frac{\rho_0}{\zeta \cosh \sigma + \sqrt{\zeta^2 - 1}\cosh \tau}, \qquad t \equiv y\sqrt{\zeta^2 - 1}\sinh \tau, \qquad \rho \equiv y\zeta \sinh \sigma, \tag{4.15}$$

where $\zeta \in [1, \infty)$ is now the holographic coordinate, and the metric becomes

$$\mathrm{d}s^2 = \frac{\mathrm{d}\zeta^2}{f(\zeta)} - f(\zeta)\,\mathrm{d}\tau^2 + \zeta^2 \mathrm{d}\sigma^2 + \zeta^2 \sinh^2 \sigma\, \mathrm{d}s^2_{\mathrm{S}^{d-2}}, \tag{4.16}$$

$$f(\zeta) \equiv \zeta^2 - 1. \tag{4.17}$$

This coordinate system does not cover the whole Poincaré patch, but only the entanglement wedge, defined as

$$\sqrt{y^2 + \rho^2} \pm t \leq \rho_0. \tag{4.18}$$

The boundary of the entanglement wedge is a horizon, $\zeta = 1$. In the original Poincaré coordinates, this horizon corresponds to the hemisphere $y^2 + \rho^2 = \rho_0^2$, which is precisely the Ryu–Takayanagi surface for a spherical subregion, as mentioned above. More precisely, the metric in eqs. (4.16) and (4.17) is still that of pure AdS, but in the coordinates of an

accelerated observer who sees a horizon at the location of the Ryu–Takayanagi minimal surface. We identify the horizon's Bekenstein–Hawking entropy as the Ryu–Takayanagi formula for the CFT's EE. The horizon has a Hawking temperature $T = 1/(2\pi)$, so the bulk theory is clearly in a thermal state.

After the coordinate change in eq. (4.15), we also expect the CFT to be on $\mathbb{R} \times \mathbb{H}^{d-1}$. To see that, we need to explain how to reach the AdS boundary in the coordinates of eq. (4.15), and we need to choose the appropriate defining function.

In the coordinates of eq. (4.15), we can approach the AdS boundary in multiple ways. If we fix $(\tau, \sigma)$ and send $\zeta \to \infty$, then $y \to 0$ and $(t, r)$ approach values given by the CHM transformation of eq. (4.5), so we arrive at an arbitrary point inside $\mathcal{D}$. Alternatively, we can fix $\zeta$ and take limits of other coordinates to reach the AdS boundary at points on the boundary of $\mathcal{D}$. For example, if we fix $(\zeta, \sigma)$ and send $\tau \to \pm\infty$, which sends $(y, t, r) \to (0, \pm\rho_0, 0)$, then we reach the upper and lower corners of $\mathcal{D}$ in fig. 4, whereas if we fix $(\zeta, \tau)$ and send $\sigma \to \infty$, which sends $(y, t, r) \to (0, 0, \rho_0)$, then we reach the sphere $\rho = \rho_0$ at $t = 0$ in fig. 4. In particular, fixing $\zeta = 1$ and sending $\sigma \to \infty$ sends $(y, t, r) \to (0, 0, \rho_0)$, indicating that the bulk horizon intersects the AdS boundary at the spherical entangling surface.

If we obtain the boundary metric using the defining function $y^2$, which after the coordinate change in eq. (4.15) is a function of $(\zeta, \tau, \sigma)$, then we obtain the metric of Minkowski spacetime in eq. (4.6). If instead we use the defining function $\zeta^2$, which is more natural in the new coordinates, then the boundary metric is precisely that of $\mathbb{R} \times \mathbb{H}^{d-1}$. In other words, changing defining function from $y^2$ to $\zeta^2$ implements the Weyl transformation that removes the conformal factor in eq. (4.6).

The change of coordinates in eq. (4.15) thus implements the CHM map. Suppose we start with the Poincaré-sliced metric in eq. (4.13), so the CFT is in Minkowski spacetime and in the conformal vacuum, and the EE of a spherical subregion is dual to the area of the Ryu–Takayanagi hemisphere, divided by $4G_N$. Eq. (4.15) switches us to an accelerated observer who sees that hemisphere as a horizon with Hawking temperature $T = 1/(2\pi)$ and Bekenstein–Hawking entropy given by the horizon's area divided by $4G_N$. In particular, this observer sees a thermal density matrix. Using the defining function $\zeta^2$, the boundary metric is that of $\mathbb{R} \times \mathbb{H}^{d-1}$. Translating to the CFT, we find a thermal state with $T = 1/(2\pi)$ and non-zero entropy on $\mathbb{R} \times \mathbb{H}^{d-1}$ equal to the EE of the original spherical subregion, precisely as expected from the CHM map in the CFT.

The change of coordinates in eq. (4.15) also allows us to compute the EREs holographically, using eq. (4.12), that is, using the Helmholtz free energy $F(\beta)$ with $\beta = 1/T = 2\pi n$. To do so, we need an asymptotically AdS metric that still has boundary metric $\mathbb{R} \times \mathbb{H}^{d-1}$, but has a horizon with Hawking temperature $T = 1/(2\pi n)$. Fortunately, this metric is known: eq. (4.16) remains a solution to Einstein's equations with negative cosmological constant if we replace $f(\zeta) = \zeta^2 - 1$ with

$$f(\zeta) = \zeta^2 - 1 + \frac{\zeta_0^{d-2} - \zeta_0^d}{\zeta^{d-2}}, \tag{4.19}$$

with arbitrary constant $\zeta_0$. The horizon is now at $\zeta = \zeta_0$, and the corresponding Hawking temperature is now

$$T = \frac{f'(\zeta_0)}{4\pi} = \frac{2 + d(\zeta_0^2 - 1)}{4\pi\zeta_0}. \tag{4.20}$$

If we choose

$$\zeta_0 = \frac{1 + \sqrt{1 + n^2 d(d-2)}}{nd}, \tag{4.21}$$

then $T = 1/(2\pi n)$. The Bekenstein–Hawking entropy of this horizon is then precisely $S_{\text{th}}(2\pi n)$ on the right-hand side of eq. (4.12), so that integrating in $n$ gives the EREs.

The Bekenstein–Hawking entropy of the horizon in eq. (4.21) is in fact $A_{\mathrm{brane}}/4G_N$, from eq. (4.14). In other words, for a spherical entangling region the CHM map, generalised to $n \neq 1$ as in eq. (4.19), gives the bulk metric with conical singularity produced by Dong's cosmic brane. This is the key result that we will use: in celestial holography we want to find the bulk spacetime produced by a cosmic brane, and we will do so using the CHM map, generalised to $n \neq 1$, in each AdS (and dS) patch.

Crucially, the interpretation of the CHM map differs between AdS/CFT and celestial holography, because their holographic dictionaries are fundamentally different, as we emphasised in sec. 1.1. In the AdS/CFT dictionary, the AdS conformal boundary is codimension one, and we identify the asymptotic AdS time coordinate with the CFT time coordinate. As a result, after the CHM map we can identify the bulk theory's thermal equilibrium state as the dual to the CFT's thermal equilibrium state. In contrast, in celestial holography the celestial sphere is codimension two, and the bulk theory is Lorentzian while the CCFT is Euclidean. As a result, a thermal equilibrium state of one theory need not be dual to a thermal equilibrium state of the other theory. In what follows, we will use a CHM map in each AdS (and dS) patch of $(d+2)$-dimensional Minkowski spacetime, which will act as usual in the CCFT, giving us a thermal state on $\mathbb{R} \times \mathbb{H}^{d-1}$.[13] However, in the bulk theory we will obtain, not a thermal state, but simply the spacetime with the appropriate conical singularity. In short, for us the CHM map is simply a convenient way to obtain the cosmic brane spacetime we want (which is possible because a spherical entangling region is a special, highly symmetric, case).

In $(d+2)$-dimensional Minkowski spacetime, our AdS patches will actually be Poincaré patches of AdS, so let us convert the spacetime with a cosmic brane conical singularity back to Poincaré patch coordinates. In other words, let us reverse the coordinate transformation in eq. (4.15). We start with the metric in eq. (4.16) with $f(\zeta)$ in eq. (4.19) and $\zeta_0$ in eq. (4.21). Converting back to Poincare patch coordinates, $(y, t, \rho)$, gives a complicated metric, but we will need only its asymptotic form, near the AdS boundary,

$$
\begin{aligned}
\mathrm{d}s^2 = {} & \frac{1}{y^2} \left( \mathrm{d}y^2 - \mathrm{d}t^2 + \mathrm{d}\rho^2 + \rho^2 \mathrm{d}s^2_{\mathrm{S}^{d-2}} \right) \\
& + \frac{Cy^{d-2}}{\omega(t,\rho)^{(d+2)/2}} \left\{ \omega(t,\rho)\,\mathrm{d}y^2 + \left[ (t^2 + \rho^2 - \rho_0^2)\mathrm{d}t - 2t\rho\,\mathrm{d}\rho \right]^2 \right\} + \mathcal{O}(y^d),
\end{aligned} \qquad (4.22)
$$

where we have defined

$$
C \equiv 2^d \rho_0^d \left( \frac{1 + \sqrt{1 + n^2 d(d-2)}}{nd} \right)^d \left( 1 - \frac{n^2 d^2}{1 + \sqrt{1 + n^2 d(d-2)}} \right),
$$

$$
\omega(t,\rho) \equiv (t + \rho + \rho_0)(t + \rho - \rho_0)(t - \rho + \rho_0)(t - \rho - \rho_0). \qquad (4.23)
$$

In these coordinates, the conical singularity corresponding to the cosmic brane is not manifest, but is present since the metric in eqs. (4.16) and (4.19) is related to that in eqs. (4.22) and (4.23) by a coordinate transformation.

The metric in eqs. (4.22) and (4.23) has the desired property that if we use $y^2$ as our defining function, then the metric at the conformal boundary is the Minkowski metric. We thus identify eqs. (4.22) and (4.23) as the bulk dual to the CFT on a single sheet of the replica manifold, with the cosmic brane as the dual of the twist fields. In what follows, for each AdS patch of $(d+2)$-dimensional Minkowski spacetime, we will perform a bulk coordinate change that produces this cosmic brane, allowing us to identify the near-$\mathcal{I}^{\pm}$

---

[13]Strictly speaking, whether a CCFT obeying the Kubo-Martin-Schwinger conditions on $\mathbb{R} \times \mathbb{H}^{d-1}$ describes a thermal state is an open question: given the CCFT's unusual properties, many subtleties could potentially arise. Nevertheless, we will use the terminology "thermal state" for lack of an alternative.

cutoffs that we need (and analogously for each dS patch). This bulk coordinate change will also implement the CHM map in the CCFT.

## 4.2 EREs in higher-dimensional CCFT

We will now translate the CHM method of sec. 4.1 from AdS/CFT to celestial holography, to compute EREs of a ball-shaped subregion in a CCFT of dimension $d \geq 2$. Consider $(d + 2)$-dimensional Minkowski spacetime, with metric

$$\mathrm{d}s^2 = -\mathrm{d}t^2 + \mathrm{d}r^2 + r^2 \left[\mathrm{d}\theta^2 + \sin^2\theta \left(\mathrm{d}\phi^2 + \sin^2\phi \, \mathrm{d}s_{\mathrm{S}^{d-2}}^2\right)\right], \tag{4.24}$$

where $\theta \in [0, \pi]$ and $\phi \in [0, \pi]$. We treat $\phi$ as the CCFT's Euclidean "time," and choose our entangling region to be the ball $\theta \leq \theta_0$ with constant $\theta_0$, at constant "time" $\phi = \pi/2$.

To apply the CHM method, we will foliate Minkowski spacetime with leaves of constant non-zero curvature, leading to the aforementioned AdS and dS patches. We thus divide Minkowski spacetime into three regions, which we label as follows:

1. AdS$_+$, the region inside the future light cone, $t^2 - r^2 > 0$, $t > 0$.

2. AdS$_-$, the region inside the past light cone, $t^2 - r^2 > 0$, $t < 0$.

3. dS, the region outside the light cone, $t^2 - r^2 < 0$.

The AdS$_\pm$ regions may be foliated with $(d + 1)$-dimensional Euclidean slices of constant negative curvature, while the dS region may be foliated with $(d+1)$-dimensional Lorentzian slices of constant positive curvature, hence their names [41, 42, 118]. Fig. 5 depicts these patches in a Penrose diagram of $(d + 2)$-dimensional Minkowski spacetime.

We will compute the EREs using eq. (3.25), reproduced here for convenience:

$$\partial_n \left(\frac{n-1}{n} S_n\right) = -i\partial_n \hat{S}_{\mathrm{grav}}^\star(n), \tag{4.25}$$

where $\hat{S}_{\mathrm{grav}}^\star(n)$ is the on-shell action of the gravitational theory, evaluated on a spacetime with conical singularity of total angle $2\pi/n$ that approaches the boundary of the entangling region $\theta = \theta_0$ near $\mathcal{I}^\pm$. We will generate the appropriate spacetime using the CHM map on the AdS$_{d+1}$ slices in the AdS$_\pm$ patches, and a closely related transformation on the dS$_{d+1}$ slices of the dS patches. We will then compute $\partial_n \hat{S}_{\mathrm{grav}}^\star(n)$ using eq. (3.27), plug the result into eq. (4.25), and integrate in $n$ to obtain the EREs.

In what follows, in both the AdS and dS patches a stereographic projection on the S$^{d-2}$ will be convenient. Defining

$$p \equiv \tan(\theta/2)\cos\phi, \qquad q \equiv \tan(\theta/2)\sin\phi, \tag{4.26}$$

the metric in eq. (4.24) becomes

$$\mathrm{d}s^2 = -\mathrm{d}t^2 + \mathrm{d}r^2 + \frac{4r^2}{(1 + p^2 + q^2)^2} \left(\mathrm{d}p^2 + \mathrm{d}q^2 + q^2\mathrm{d}s_{\mathrm{S}^{d-2}}^2\right), \tag{4.27}$$

In this coordinate system, the entangling region is the ball $q \leq \tan(\theta_0/2)$ at $p = 0$.

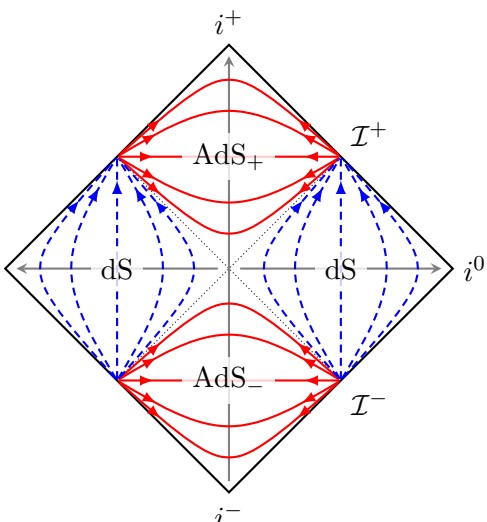

Figure 5: A Penrose diagram of $(d+2)$-dimensional Minkowski space. The dotted lines are the lightcone of the origin. The two regions labelled $\mathrm{AdS}_{\pm}$ are foliated by Euclidean $\mathrm{AdS}_{d+1}$ slices, denoted by the solid red lines in the figure. The red arrows parallel to the slices denotes the direction of increasing radial coordinate $y$. The grey arrow perpendicular to the slices denotes the direction of increasing Milne time $\tau$. The region labelled dS (appearing twice in the diagram) is foliated by $d+1$-dimensional Lorentzian dS slices, denoted by blue dashed lines. The blue arrows parallel to the slices denote the direction of increasing dS time $T$, while the grey arrows perpendicular to the slices denote the direction of increasing Milne time $\tilde{\tau}$. We labeled past and future null infinity as $\mathcal{I}^-$ and $\mathcal{I}^+$, respectively, past and future infinity as $i^-$ and $i^+$, respectively, and spatial infinity as $i^0$.

### 4.2.1   AdS$_{\pm}$ patch contributions

In the AdS$_{\pm}$ patch, we obtain a spacetime with the appropriate conical singularity in four steps, as follows. First, we transform to Milne time, with AdS slices in Poincaré form, analogous to eq. (4.13). Second, we perform the CHM map to a metric analogous to eqs. (4.16) and (4.17). Third, we generalise to $n \neq 1$, analogous to eqs. (4.16), (4.19), and (4.21). Fourth, we transform back to a metric of Poincaré form, but now with the appropriate conical singularity, analogous to eqs. (4.22) and (4.23). Once we have the appropriate spacetime, we introduce near-brane and near-$\mathcal{I}^{\pm}$ cutoffs, and map the latter to the CCFT near twist field cutoff $\epsilon$, similar to what we did in sec. 3.3. The two main results of this section are eq. (4.40) for the AdS$_{\pm}$ contribution to $\partial_n \hat{S}^{\star}_{\mathrm{grav}}(n)$ and eq. (4.44) for the map from the near-brane and near-$\mathcal{I}^{\pm}$ cutoffs to $\epsilon$.

For the first step, in the AdS$_{\pm}$ patches we define Milne coordinates $(\tau, y, x, \rho)$ via

$$\tau \equiv \sqrt{t^2 - r^2}, \qquad\qquad y \equiv \frac{(1 + p^2 + q^2)\sqrt{t^2 - r^2}}{t + r + (t - r)(p^2 + q^2)},$$

$$x \equiv \frac{2rp}{t + r + (t - r)(p^2 + q^2)}, \qquad \rho \equiv \frac{2rq}{t + r + (t - r)(p^2 + q^2)}, \qquad (4.28)$$

where the Milne time $\tau \in [0, \infty)$, and $y \in [0, \infty)$, $x \in (-\infty, \infty)$, and $\rho \in [0, \infty)$, so these coordinates cover the whole of the AdS$_{\pm}$ patch. After the coordinate transformation in eq. (4.28), the Minkowski metric in eq. (4.27) becomes

$$\mathrm{d}s^2 = -\mathrm{d}\tau^2 + \frac{\tau^2}{y^2}\left(\mathrm{d}y^2 + \mathrm{d}x^2 + \mathrm{d}\rho^2 + \rho^2\,\mathrm{d}s^2_{\mathrm{S}^{d-2}}\right). \qquad (4.29)$$

On each slice of constant $\tau$, the induced metric is that of the Poincaré patch of Euclidean AdS$_{d+1}$, or in other words $\mathbb{H}^{d+1}$, with curvature radius $\tau$. The conformal boundary of each $\mathbb{H}^{d+1}$ slice is at $y = 0$. Sending $y \to 0$ with all other coordinates held fixed sends $r \to \infty$ and $u = t - r \to 0$, so we approach the point where $\mathcal{I}^\pm$ intersects the lightcone of the origin, which in fig. 5 is on the black line midway between $i^0$ and $i^\pm$, respectively. To reach points on $\mathcal{I}^\pm$ with $u > 0$, we should send both $y \to 0$ and $\tau \to \infty$ with $\tau y$ held fixed, which in fig. 5 is on the black line between $\mathcal{I}^\pm$ and $i^\pm$.

For the second step, we perform the CHM map by defining coordinates $(\zeta, \sigma, \psi)$ via

$$
y \equiv \frac{\tan(\theta_0/2)}{\zeta \cosh \sigma + \sqrt{\zeta^2 - 1} \cos \psi}, \qquad x \equiv y\sqrt{\zeta^2 - 1} \sin \psi, \qquad \rho \equiv y\zeta \sinh \sigma, \tag{4.30}
$$

which is a Wick rotation of the transformation in eq. (4.15). After the coordinate transformation in eq. (4.30), the metric in eq. (4.29) becomes

$$
\mathrm{d}s^2 = -\mathrm{d}\tau^2 + \tau^2 \left[ \frac{\mathrm{d}\zeta^2}{f(\zeta)} + f(\zeta)\mathrm{d}\psi^2 + \zeta^2 \mathrm{d}\sigma^2 + \zeta^2 \sinh^2 \sigma \, \mathrm{d}s_{\mathrm{S}^{d-2}}^2 \right], \tag{4.31}
$$

$$
f(\zeta) \equiv \zeta^2 - 1. \tag{4.32}
$$

The metric on surfaces of constant $(\tau, \zeta)$ is that of $\mathrm{S}^1 \times \mathbb{H}^{d-1}$. We can reach the boundary of each $\mathbb{H}^{d+1}$ slice, at $y = 0$, in two ways. The first way is to send $\zeta \to \infty$ with all other coordinates held fixed, in which case

$$
u = \frac{\tau \sin \theta_0}{2\zeta(\cos\theta_0 \cos\psi + \cosh\sigma)} + \mathcal{O}(\zeta^{-3}), \tag{4.33a}
$$

$$
r = \frac{\tau\zeta}{\sin\theta_0}(\cos\theta_0 \cos\psi + \cosh\sigma) + \mathcal{O}(\zeta^{-1}), \tag{4.33b}
$$

$$
p = \frac{\sin\psi \tan(\theta_0/2)}{\cos\psi + \cosh\sigma} + \mathcal{O}(\zeta^{-2}), \tag{4.33c}
$$

$$
q = \frac{\sinh\sigma \tan(\theta_0/2)}{\cos\psi + \cosh\sigma} + \mathcal{O}(\zeta^{-2}). \tag{4.33d}
$$

The second way is to send $\sigma \to \infty$ with all other coordinates held fixed, in which case

$$
u = \frac{\tau \sin\theta_0}{\zeta}e^{-\sigma} + \mathcal{O}(e^{-2\sigma}), \tag{4.34a}
$$

$$
r = \frac{\tau\zeta}{2\sin\theta_0}e^{\sigma} + \mathcal{O}(1), \tag{4.34b}
$$

$$
p = \frac{2\tan(\theta_0/2)\sqrt{\zeta^2 - 1}\sin\psi}{\zeta}e^{-\sigma} + \mathcal{O}(e^{-2\sigma}), \tag{4.34c}
$$

$$
q = \tan(\theta_0/2) - \frac{2\tan(\theta_0/2)\sqrt{\zeta^2 - 1}\cos\psi}{\zeta}e^{-\sigma} + \mathcal{O}(e^{-2\sigma}), \tag{4.34d}
$$

which thus takes us to the boundary of the entangling region, $q = \tan(\theta_0/2)$ at $p = 0$.

For the third step, we need to generalise to $n \neq 1$. In the coordinates of eq. (4.31), a circle wound by the angle $\psi$ degenerates at $\zeta = 1$. Eq. (4.31) remains a local solution to the vacuum Einstein equations if we replace $f(\zeta)$ in eq. (4.32) with the function in eq. (4.19), reproduced here for convenience,

$$
f(\zeta) = \zeta^2 - 1 + \frac{\zeta_0^{d-2} - \zeta_0^d}{\zeta^{d-2}}, \tag{4.35}
$$

so that the $\psi$ "circle" now degenerates at $\zeta = \zeta_0$. To maintain the periodicity $\psi \sim \psi + 2\pi$ but obtain a conical singularity at $\zeta = \zeta_0$ of total angle $2\pi/n$, we choose the $\zeta_0$ in eq. (4.21),

$$\zeta_0 = \frac{1 + \sqrt{1 + n^2 d(d-2)}}{nd}. \tag{4.36}$$

For the fourth and final step, we reverse the coordinate transformations in eqs. (4.28) and (4.30), and also switch from $t$ to retarded time $u = t - r$. The metric then becomes similar to that in eqs. (4.22) and (4.23): at leading order near $\mathcal{I}^\pm$,

$$ds^2 = -du^2 - 2du\,dr + \frac{4r^2}{(1 + p^2 + q^2)^2}\left(dp^2 + dq^2 + q^2 ds_{S^{d-2}}^2\right) \tag{4.37}$$

$$- \frac{C}{\omega(p,q)^{(d+2)/2}}(1 + p^2 + q^2)^{d-2}u^{d/2}r^{(4-d)/2}\left[(p^2 - q^2 + q_0^2)dp + 2pq\,dq\right]^2 + \dots,$$

where the dots represent terms that vanish in the limit $r \to \infty$, and

$$C \equiv 2^{(d+4)/2}(\tan\theta_0)^d \zeta_0^{d-2}(\zeta_0^2 - 1),$$
$$\omega(p,q) \equiv \left[p^2 + (q - \tan(\theta_0/2))^2\right]\left[p^2 + (q + \tan(\theta_0/2))^2\right]. \tag{4.38}$$

We will use eqs. (4.37) and (4.38) to compute the near-$\mathcal{I}^\pm$ cutoff, as we will see shortly.

We now want to use eq. (4.25) to calculate EREs, and in particular we want to calculate $\partial_n \hat{S}_{\text{grav}}^\star(n)$. To do so, we will use the metric in eq. (4.31). However, in those coordinates the boundaries of the integration over spacetime in $\partial_n \hat{S}_{\text{grav}}^\star(n)$ will depend on $n$. To avoid that, we define a rescaled radial coordinate, $\xi \equiv \zeta/\zeta_0$, so the metric in eq. (4.31) becomes

$$ds^2 = -d\tau^2 + \zeta_0^2\tau^2\left[\frac{d\xi^2}{F(\xi)} + \frac{F(\xi)}{\zeta_0^2}d\psi^2 + \xi^2 d\sigma^2 + \xi^2 \sinh^2\sigma\,ds_{S^{d-2}}^2\right], \tag{4.39a}$$

$$F(\xi) = \zeta_0^2\xi^2 - 1 + (1 - \zeta_0^2)\xi^{2-d}. \tag{4.39b}$$

We then find that the contribution of the AdS$_\pm$ patch to $\partial_n \hat{S}_{\text{grav}}^\star(n)$ is

$$\partial_n \hat{S}_{\text{AdS}_\pm}^\star(n) \equiv -\frac{\text{vol}(S^{d-2})}{4G_N}\frac{\zeta_0^{d-1}}{n^2}\int_0^L d\tau\,\tau^{d-1}\int_0^{\sigma_c} d\sigma\,\sinh^{d-2}\sigma, \tag{4.40}$$

where $\text{vol}(S^{d-2})$ is the volume of a unit-radius $S^{d-2}$ and $\zeta_0$ is the function of $n$ in eq. (4.36).

In eq. (4.40), the integrals over $\tau$ and $\sigma$ each diverge, so we introduced cutoffs at their upper endpoints. The cutoff on the $\tau$ integration is $L$, which is in fact identical to that in the $d = 2$ case of sec. 3.3, specifically eq. (3.33), $UV = uv = t^2 - r^2 \leq L^2$.

The cutoff on the $\sigma$ integration is $\sigma_c$, which comes from the intersection of a near-brane cutoff with a near-$\mathcal{I}^\pm$ cutoff, similar to what we did in sec. 3.3. In eq. (4.34) at large $\sigma$ the angle $\psi$ parameterises a small circle in the $(p,q)$ plane that winds around the boundary of the entangling region at $p = 0$ and $q = \tan(\theta_0/2)$. To define a cutoff near the boundary of the entangling region, we define

$$q_0 \equiv \tan(\theta_0/2), \tag{4.41}$$

and then move slightly away from $q_0$: using the metric in eq. (4.37) to define a metric on the celestial sphere, we move from $q_0$ to $q_0 + \delta q$ where

$$\delta q \equiv 2\tan(\theta_0/2)\sqrt{1 - \zeta_0^{-2}}\,e^{-\sigma_c}. \tag{4.42}$$

If we regularise the conical singularity at the cosmic brane by cutting out the tube $\xi < 1 + \delta\xi$ for some $\delta\xi \ll 1$, or equivalently $\zeta/\zeta_0 \ll 1$, then this near-brane cutoff meets the large $\sigma$ cutoff at a circle of radius

$$\int_{q_0}^{q_0+\delta q} \frac{2\,\mathrm{d}q}{1+q^2} \approx 2\sin\theta_0 \sqrt{1-\zeta_0^{-2}}\, e^{-\sigma_c}, \tag{4.43}$$

which we identify as the CCFT's near twist field cutoff, $\epsilon$. Actually, to simplify unphysical, scheme-dependent terms in the EREs we will identify eq. (4.43) with $\epsilon\sqrt{1-\zeta_0^{-2}}$. Doing so leaves the physical contributions to the EREs unchanged. We thus obtain the precise relationship between our cutoff $\sigma_c$ and the CCFT near twist field cutoff $\epsilon$, namely

$$\sigma_c = \log\left(\frac{2}{\epsilon}\sin\theta_0\right). \tag{4.44}$$

### 4.2.2   dS contribution

In the dS patch, we obtain a spacetime with the appropriate conical singularity in three steps, as follows. First, we transform to Milne time with Lorentzian dS slices. Second, we perform a CHM map on the dS slice. Third, we generalise to $n \neq 1$. We then have the spacetime with the appropriate conical singularity, in which we introduce near-brane and near-$\mathcal{I}^\pm$ cutoffs, and map the latter to the CCFT near twist field cutoff $\epsilon$. The main result of this section is eq. (4.55) for the dS contribution to $\partial_n \hat{S}^\star_{\mathrm{grav}}(n)$.

For the first step, in the dS patch we define coordinates $\tilde{\tau}$ and $T$ via

$$t \equiv \tilde{\tau}\sinh T, \qquad r \equiv \tilde{\tau}\cosh T, \tag{4.45}$$

or equivalently via

$$T \equiv \frac{1}{2}\log\left(\frac{r+t}{r-t}\right), \qquad \tilde{\tau} \equiv \sqrt{r^2-t^2}, \tag{4.46}$$

where the Milne time $\tilde{\tau} \in [0, \infty)$ and the dS slice time $T \in (-\infty, \infty)$. After the coordinate transformation in eq. (4.46), the Minkowski metric in eq. (4.27) becomes

$$\mathrm{d}s^2 = \mathrm{d}\tilde{\tau}^2 + \tilde{\tau}^2\left[-\mathrm{d}T^2 + \frac{4\cosh^2 T}{(1+p^2+q^2)^2}\left(\mathrm{d}p^2 + \mathrm{d}q^2 + q^2\mathrm{d}s^2_{\mathrm{S}^{d-2}}\right)\right], \tag{4.47}$$

where the factor in brackets is the metric of unit-radius $\mathrm{dS}_{d+1}$ in global coordinates. For convenience, we make a further coordinate transformation to put $\mathrm{dS}_{d+1}$ in flat slicing. However, since the flat slicing covers only half of dS [119], we will need to subdivide the dS patch into two coordinate patches, which we label as $\mathrm{dS}_\pm$. We define $\mathrm{dS}_+$ as the patch where $e^{2T} \geq p^2 + q^2$, while $\mathrm{dS}_-$ is the patch where $e^{2T} \leq p^2 + q^2$. In the $\mathrm{dS}_\pm$ patches we define coordinates $(\tilde{y}, x, \rho)$ via

$$\tilde{y} \equiv \pm\frac{(1+p^2+q^2)\sqrt{r^2-t^2}}{r+t-(r-t)(p^2+q^2)} = \pm\frac{1+p^2+q^2}{e^T - e^{-T}(p^2+q^2)}, \tag{4.48a}$$

$$x \equiv \pm\frac{2rp}{r+t-(r-t)(p^2+q^2)} = \pm\frac{2p\cosh T}{e^T - e^{-T}(p^2+q^2)}, \tag{4.48b}$$

$$\rho \equiv \pm\frac{2rq}{r+t-(r-t)(p^2+q^2)} = \pm\frac{2q\cosh T}{e^T - e^{-T}(p^2+q^2)}, \tag{4.48c}$$

where the $\pm$ sign is the same as that in $dS_\pm$. In each patch the time coordinate $\tilde{y} \in [0, \infty)$. After the coordinate transformation in eq. (4.48), the metric in eq. (4.47) becomes

$$ds^2 = d\tilde{\tau}^2 + \frac{\tilde{\tau}^2}{\tilde{y}^2}\left(-d\tilde{y}^2 + dx^2 + d\rho^2 + \rho^2 ds_{S^{d-2}}^2\right). \tag{4.49}$$

The $dS_+$ patch includes all of $\mathcal{I}^+$, located at $T \to \infty$ with $\tilde{\tau}e^{-T}$ held fixed. Similarly, the $dS_-$ patch includes all of $\mathcal{I}^-$, located at $T \to -\infty$ with $\tilde{\tau}e^T$ held fixed.

For the second step, in either of $dS_\pm$ we perform a CHM map by defining coordinates $(\tilde{\zeta}, \tilde{\sigma}, \tilde{\psi})$ via

$$\tilde{y} \equiv \frac{\tan(\theta_0/2)}{\tilde{\zeta}\sinh\tilde{\sigma} + \sqrt{1 - \tilde{\zeta}^2}\cos\tilde{\psi}}, \qquad x \equiv \tilde{y}\sqrt{1 - \tilde{\zeta}^2}\sin\tilde{\psi}, \qquad \rho \equiv \tilde{y}\tilde{\zeta}\cosh\tilde{\sigma}. \tag{4.50}$$

Eq. (4.50) is a continuation of the CHM transformation in eq. (4.30), obtained by setting $\tilde{\sigma} = \sigma - i\pi/2$, $\tilde{\zeta} = \zeta$, and $\tilde{\psi} = \psi$. After the coordinate transformation in eq. (4.50), the metric in eq. (4.49) becomes

$$ds^2 = d\tilde{\rho}^2 + \tilde{\rho}^2\left[\frac{d\tilde{\zeta}^2}{\tilde{f}(\tilde{\zeta})} + \tilde{f}(\tilde{\zeta})d\tilde{\psi}^2 - \tilde{\zeta}^2 d\tilde{\sigma}^2 + \tilde{\zeta}^2\cosh^2\tilde{\sigma}\, ds_{S^{d-2}}^2\right], \tag{4.51}$$

$$\tilde{f}(\tilde{\zeta}) \equiv 1 - \tilde{\zeta}^2, \tag{4.52}$$

The metric on surfaces of constant $(\tilde{\rho}, \tilde{\zeta})$ is that of $S^1 \times dS_{d-1}$. Although the coordinates $(\tilde{\zeta}, \tilde{\sigma}, \tilde{\psi})$ do not cover all of $dS_\pm$, they include the cosmic brane, located at $\tilde{\zeta} = 1$, where the $\tilde{\psi}$ circle degenerates.

For the third step, we generalise to $n \neq 1$ by replacing the $\tilde{f}(\tilde{\zeta})$ in eq. (4.52) with

$$\tilde{f}(\tilde{\zeta}) = 1 - \tilde{\zeta}^2 - \frac{\zeta_0^{d-2} - \zeta_0^d}{\tilde{\zeta}^{d-2}}, \tag{4.53}$$

so that the $\tilde{\psi}$ circle now degenerates at $\tilde{\zeta} = \zeta_0$. To maintain the periodicity $\tilde{\psi} \sim \tilde{\psi} + 2\pi$ but obtain a conical singularity at $\tilde{\zeta} = \zeta_0$ of total angle $2\pi/n$, we choose the $\zeta_0$ in eq. (4.36).

We now want to calculate $\partial_n \hat{S}_{\text{grav}}^\star(n)$ in the $dS_\pm$ patches. However, with the coordinates $(\tilde{\zeta}, \tilde{\sigma}, \tilde{\psi})$ the boundaries of the integration region depend on $n$. To avoid that, we define a rescaled radial coordinate, $\tilde{\xi} \equiv \tilde{\zeta}/\zeta_0$. The metric in eq. (4.51) then becomes

$$ds^2 = d\tilde{\tau}^2 + \zeta_0^2\tilde{\tau}^2\left[\frac{d\tilde{\xi}^2}{\tilde{F}(\tilde{\xi})} + \frac{\tilde{F}(\tilde{\xi})}{\zeta_0^2}d\tilde{\psi}^2 - \tilde{\xi}^2 d\tilde{\sigma}^2 + \tilde{\xi}^2\cosh^2\tilde{\sigma}\, ds_{S^{d-2}}^2\right]. \tag{4.54}$$

We then find that the contribution of the $dS_\pm$ patch to $\partial_n \hat{S}_{\text{grav}}^\star(n)$ is

$$\partial_n \hat{S}_{dS_\pm}^\star(n) \equiv -\frac{\text{vol}(S^{d-2})}{4G_N}\frac{\zeta_0^{d-1}}{n^2}\int_0^L d\tilde{\tau}\,\tilde{\tau}^{d-1}\int_0^{\sigma_c} d\tilde{\sigma}\,\cosh^{d-2}\tilde{\sigma}. \tag{4.55}$$

In eq. (4.55), the integrals over $\tilde{\tau}$ and $\tilde{\sigma}$ each diverge, so we introduced cutoffs on each. The cutoff on the $\tilde{\tau}$ integration, $L$, is identical to that in eq. (3.33), $UV = uv = t^2 - r^2 \leq L^2$. Arguments identical to those of the previous section give us the cutoff on the $\tilde{\sigma}$ integration, namely that in eq. (4.44), $\sigma_c = \log\left(\frac{2}{\epsilon}\sin\theta_0\right)$.

### 4.2.3   Results for EREs

We now want to compute $\partial_n \hat{S}^{\star}(n)$, which is the sum of contributions from the $\text{AdS}_{\pm}$ patches, eq. (4.40), and the $\text{dS}_{\pm}$ patches, eq. (4.55):

$$\partial_n S_{\text{grav}}^{\star}(n) = \partial_n \hat{S}_{\text{AdS}_+}^{\star}(n) + \partial_n \hat{S}_{\text{AdS}_-}^{\star}(n) + \partial_n \hat{S}_{\text{dS}_+}^{\star}(n) + \partial_n \hat{S}_{\text{dS}_-}^{\star}(n). \tag{4.56}$$

Re-labelling $\tilde{\tau} \to \tau$ and $\tilde{\sigma} \to \sigma$ in the $\text{dS}_{\pm}$ patches allows us to write

$$\partial_n S_{\text{grav}}^{\star}(n) = -\frac{\text{vol}(S^{d-2})}{2G_{\text{N}}} \frac{\zeta_0^{d-1}}{n^2} \int_0^L d\tau\, \tau^{d-1} \int_0^{\sigma_c} d\sigma \left( \sinh^{d-2}\sigma + \cosh^{d-2}\sigma \right), \tag{4.57}$$

with $\sigma_c = \log\left(\frac{2}{\epsilon} \sin\theta_0\right)$. Performing the trivial integration over $\tau$ then gives

$$\partial_n S_{\text{grav}}^{\star}(n) = -\frac{\text{vol}(S^{d-2})}{2G_{\text{N}}} \frac{\zeta_0^{d-1}}{n^2} \frac{L^d}{d} \int_0^{\sigma_c} d\sigma \left( \sinh^{d-2}\sigma + \cosh^{d-2}\sigma \right). \tag{4.58}$$

If we define

$$\mathfrak{t} \equiv \tanh\sigma_c, \tag{4.59}$$

then the integral over $\sigma$ gives

$$\partial_n S_{\text{grav}}^{\star}(n) = -\frac{\text{vol}(S^{d-2})}{2G_{\text{N}}} \frac{\zeta_0^{d-1}}{n^2} \frac{L^d}{d}$$
$$\times \left[ \mathfrak{t}\, {}_2F_1\left(\frac{1}{2}, \frac{d}{2}; \frac{3}{2}; \mathfrak{t}^2\right) + \frac{\mathfrak{t}^{d-1}}{d-1}\, {}_2F_1\left(\frac{d-1}{2}, \frac{d}{2}; \frac{d+1}{2}; \mathfrak{t}^2\right) \right]. \tag{4.60}$$

We now need to plug eq. (4.60) into eq. (3.25) and perform the integration over $n$, subject to the boundary condition that $S_{\rho_0, n}$ does not diverge as a function of $n$ as $n \to 1$. In eq. (4.60), the only dependence on $n$ appears in the factor $\zeta_0^{d-1}/n^2$, with $\zeta_0$ in eq. (4.36). The integral we need is thus, with dummy integration variable $m$,

$$\mathcal{F}_d(n) \equiv \frac{n}{n-1} \int_1^n \frac{dm}{m^2} \left( \frac{1 + \sqrt{1 + m^2 d(d-2)}}{md} \right)^{d-1}. \tag{4.61}$$

Performing the integration over $m$ gives

$$\mathcal{F}_d(n) = \frac{n}{n-1} + \frac{nd}{(n-1)(d-2)^2} \left( \frac{1 + \sqrt{1 + n^2 d(d-2)}}{d} \right)^{d+1} \tag{4.62}$$
$$- \frac{2 + n^2(d-1)(d-2)}{n(n-1)(d-2)^2} \left( \frac{1 + \sqrt{1 + n^2 d(d-2)}}{nd} \right)^d.$$

Relevant for the EE, for all $d$ we have $\lim_{n \to 1} \mathcal{F}_d(n) = 1$. Our final result for the EREs, which is the main result of this paper, is thus

$$S_{\rho_0, n} = i \frac{\text{vol}(S^{d-2})}{2G_{\text{N}}} \frac{L^d}{d} \mathcal{F}_d(n) \left[ \mathfrak{t}\, {}_2F_1\left(\frac{1}{2}, \frac{d}{2}; \frac{3}{2}; \mathfrak{t}^2\right) + \frac{\mathfrak{t}^{d-1}}{d-1}\, {}_2F_1\left(\frac{d-1}{2}, \frac{d}{2}; \frac{d+1}{2}; \mathfrak{t}^2\right) \right]. \tag{4.63}$$

We will compare our eq. (4.63) to the general form expected for EREs of a spherical subregion of a CFT vacuum, namely

$$S_{\rho_0, n} = a_{d-2}(n) \frac{A_{\rho_0}}{\epsilon^{d-2}} + \frac{a_{d-4}(n)}{\epsilon^{d-4}} + \ldots + a_{\log}(n) \log\left(\frac{2}{\epsilon} \sin\theta_0\right) + a_0(n) + \mathcal{O}(\epsilon), \tag{4.64}$$

with dimensionless coefficients $a_{d-2}(n)$, $a_{d-4}(n)$, ... that can depend on $n$, as indicated, where $a_{\log}(n)$ can appear only when $d$ is even, and $A_{\rho_0} = \text{vol}(S^{d-2})\,(\sin\theta_0)^{d-2}$ is the surface area of the spherical entangling region. In other words, the leading divergence exhibits an "area law" [120,121]. The physical, cutoff-independent contributions to $S_{\rho_0,n}$ are either $a_{\log}(n)$ for even $d$, or $a_0(n)$ for odd $d$. For a spherical subregion, $\lim_{n\to 1} a_{\log}(n) \propto (-1)^{\frac{d}{2}-1}a_d$, where $a_d$ is the coefficient of the Euler density term in the CFT's trace anomaly. Analogously, $\lim_{n\to 1} a_0(n) \propto (-1)^{\frac{d}{2}-2}Z(S^d)$, where $Z(S^d)$ is the Euclidean CFT's partition function on $S^d$. In a reflection positive/unitary CFT in $d \leq 4$, proofs exist that these quantities obey $c$-theorems under renormalization group flow [93–97], suggesting that these quantities count the number of dynamical degrees of freedom.

Our eq. (4.63) has the form of eq. (4.64), where all coefficients $a_{d-2}(n)$, $a_{d-4}(n)$, ... are proportional to the same function of $n$, namely $\mathcal{F}_d(n)$ in eq. (4.62). Eq. (4.63) also gives rise to an area law, because $\epsilon \to 0$ means $\sigma_c \to \infty$, so that in eq. (4.58) when $d > 2$,

$$\partial_n S_{\text{grav}}^\star(n) = -\frac{\text{vol}(S^{d-2})}{2G_N}\frac{\zeta_0^{d-1}}{n^2}\frac{L^d}{d}\frac{2\,e^{(d-2)\sigma_c}}{(d-2)} + \ldots, \tag{4.65}$$

and using $\sigma_c = \log\left(\frac{2}{\epsilon}\sin\theta_0\right)$ and $A_{\rho_0} = \text{vol}(S^{d-2})\,(\sin\theta_0)^{d-2}$ then gives

$$\partial_n S_{\text{grav}}^\star(n) = -\frac{1}{2G_N}\frac{\zeta_0^{d-1}}{n^2}\frac{L^d}{d}\frac{2^{d-1}}{d-2}\frac{A_{\rho_0}}{\epsilon^{d-2}} + \ldots, \tag{4.66}$$

so that the leading divergence in eq. (4.63) will indeed have the form $A_{\rho_0}/\epsilon^{d-2}$. To be explicit, expanding eq. (4.63) for small $\epsilon$, or equivalently for $1 - \mathfrak{t} \ll 1$, and neglecting terms that vanish as $\epsilon \to 0$, for $d \leq 6$ we find

$$d = 2: \qquad S_{\ell,n} = \frac{iL^2}{2G_N}\left(1 + \frac{1}{n}\right)\log\left[\frac{2}{\epsilon}\sin\left(\frac{\ell}{2}\right)\right], \tag{4.67a}$$

$$d = 3: \qquad S_{\rho_0,n} = \frac{iL^3}{3G_N}\mathcal{F}_3(n)\left(\frac{A_{\rho_0}}{\epsilon} - \pi\right), \tag{4.67b}$$

$$d = 4: \qquad S_{\rho_0,n} = \frac{iL^4}{8G_N}\mathcal{F}_4(n)\frac{A_{\rho_0}}{\epsilon^2}, \tag{4.67c}$$

$$d = 5: \qquad S_{\rho_0,n} = \frac{iL^5}{15G_N}\mathcal{F}_5(n)\left(\frac{A_{\rho_0}}{\epsilon^3} + 2\pi^2\right), \tag{4.67d}$$

$$d = 6: \qquad S_{\rho_0,n} = \frac{iL^6}{24G_N}\mathcal{F}_6(n)\left[\frac{A_{\rho_0}}{\epsilon^4} + 4\pi^2\log\left(\frac{2}{\epsilon}\sin\theta_0\right)\right]. \tag{4.67e}$$

For any $d$, we may straightforwardly extract the EE using $\lim_{n\to 1}\mathcal{F}_d(n) = 1$.

For the $d = 2$ case in eq. (4.67a), we substituted the explicit result for $\mathcal{F}_2(n)$ and the interval's length $\ell = 2\theta_0$. The resulting expression agrees exactly with our result in sec. 3.3, namely eq. (3.46). In fact, the agreement extends to the non-universal terms, although this was not guaranteed, since we have used different regularisation prescriptions in the two calculations. Furthermore, in the appendix we derive the $d = 2$ result in a third way, using methods similar to those of refs. [73], providing another check.

In eq. (4.67), a logarithmic term is noticeably absent when $d = 4$, suggesting that $a_4 = 0$ in a $d = 4$ CCFT. Ref. [73] found the same result, which occurs because the contributions to the ERE from the $\text{AdS}_\pm$ and $\text{dS}_\pm$ regions cancel each other, when using

the same cutoff in these regions, as we have done. In contrast, in eq. (4.67) a logarithmic term appears when $d = 2$ and $6$, suggesting that $a_d \neq 0$ in those cases. More generally, from eq. (4.63) we find that $a_d = 0$ whenever $d$ is a multiple of 4, whereas $a_d \neq 0$ for all other even $d$, at least up to the largest value we checked, $d = 20$. This raises the tantalizing question of whether CCFTs with $d$ a multiple of 4 have dynamical degrees of freedom at all, as defined via $c$-theorems—a question we leave for future research.

# 5   Summary and Outlook

In this paper, we computed the EREs of CCFTs holographically, using the bulk theory of Einstein–Hilbert gravity with minimally-coupled matter fields. We started with $d = 2$ CCFT and single-interval EREs. In sec. 3.1 we identified the superrotation dual to the uniformisation map, which allowed us to identify the bulk duals of the replica manifold and twist fields. A key result was that twist fields are dual to the endpoints of a particular type of cosmic string, with zero charge under any bulk gauge group and with the tension in eq. (3.19). In sec. 3.2, we made two key assumptions, namely that we can identify the bulk and CCFT partition functions (eq. (3.21)) and that we can make a leading saddle point approximation to the bulk partition function (eq. (3.22)). In sec. 3.3 we thus computed the EREs, with the result in eq. (3.46) of the form required by $d = 2$ conformal symmetry, which enabled us to identify the $c$ in eq. (3.47). For $d \geq 2$, in sec. 4 we made the same assumptions, and argued that the bulk dual of twist operators are cosmic branes. In sec. 4.2 we adapted the CHM map to compute EREs of spherical sub-regions of $d \geq 2$ CCFTs, with the result in eq. (4.63), which is the main result of this paper. The result has the form expected from conformal symmetry, and enabled us to identify central charges in any dimension. A curious result was that the central charge defined as the coefficient of the Ricci scalar in the CCFT trace anomaly appears to vanish in $d = 4 \mod 4$.

Our results raise many questions for future research. What follows are questions that we consider the most important and/or tractable. To be concrete, we will restrict to $d = 2$ CCFT, and hence a 4-dimensional bulk, although the same questions apply when $d > 2$.

- **Interpretation as entanglement:** As mentioned in sec. 1.3, the CCFT is defined on $\mathcal{CS}^2$, so strictly speaking we computed correlators of twist fields in a Euclidean signature theory. Interpreting these in terms of entanglement requires an analytic continuation to Lorentzian signature, which is not unique. Since we had a single interval, and hence two twist fields that defined a great circle of $\mathcal{CS}^2$, our preferred approach is to perform stereographic projection to the complex plane, which maps that great circle to a line with the two twist fields, and then to Wick rotate the direction normal to that line, which is thus our Cauchy surface. Our calculation then maps onto Cardy and Calabrese's relatively straightforwardly. However, in general other bulk solutions will not be dual to EREs. For example, suppose we have multiple strings whose endpoints reach $\mathcal{CS}^2$, or a scattering process in the bulk in the presence of cosmic strings. These can lead to additional operator insertions on $\mathcal{CS}^2$, and not all of these must be on the same great circle. If we nevertheless choose a great circle, stereographically project to the plane, and then Wick rotate as described above, then some operator insertions will not be on our Cauchy surface, and so will instead define two *different* states on either side of our Cauchy surface. In other words, we will obtain, not a density matrix of a single state, but a transition matrix between two different states. For details, see for example ref. [19]. The analogue of EE for a transition matrix is called *pseudo*-entropy, whose properties are very different from EE [122–124]. For example, even in a unitary field theory,

a transition matrix is in general non-Hermitian, and as a result, pseudo-entropy is complex-valued and need not obey (strong) sub-additivity or other entanglement inequalities. We believe such issues will arise with *any* choice of Wick rotation of the CCFT: in general, bulk calculations involving cosmic strings dual to twist fields will compute pseudo-entropy, not EREs or EEs. In short, a fundamental conceptual question for follow-up research is: for a generic bulk solution involving cosmic strings, exactly what is the dual CCFT quantity? Is it some kind of pseudo-entropy?

- **Parallel cosmic strings:** As mentioned in 1.4, many cosmic string solutions exist in the bulk Einstein–Hilbert theory: see for example ref. [125] and references therein. Our results open the door to interpreting these as probes of CCFT ERE or pseudo-entropy. For instance, a relatively simple generalization of the single cosmic string solution is to multiple parallel cosmic strings [126]. By definition, parallel strings all approach the same points on $\mathcal{CS}^2$, so such a solution should be dual to a correlation function of multiple twist fields inserted at coincident points. However, since the parallel strings are translated with respect to one-another, the correlation function should also contain insertions of the appropriate (super)translation operator. For example, we might consider constructing a configuration of two parallel strings by first performing a superrotation of the form in eq. (3.12) to insert a string with endpoints at $z_1$ and $z_2$, followed by a translation to move the string away from the origin, followed by another superrotation to insert another string, also with endpoints at $z_1$ and $z_2$, and possibly with different tension. We expect this bulk solution to be dual to a correlation function schematically of the form

$$\langle \Phi_{n'}(z_1)\Phi_{-n'}(z_2)\tilde{\Phi}_n(z_1)\tilde{\Phi}_{-n}(z_2)\rangle, \qquad \tilde{\Phi}_n(z) \equiv U^\dagger \Phi_n(z)U, \qquad (5.1)$$

where $U$ is the appropriate (super)translation operator and $n$ and $n'$ are the indices related to the tensions of the two strings through eq. (3.19). Analysis of this correlation function in the CCFT will require careful treatment of the coincident limit of twist fields, perhaps along the lines of ref. [127].

- **Time-dependent cosmic string processes:** Other well-known classes of bulk solutions describe a single cosmic string that snaps into two pieces, the time-reversed process in which two string pieces meet and connect to form a single cosmic string, or a process in which a cosmic string of finite length nucleates and grows to infinite length [106, 107, 128–130]. In these cases the cosmic string endpoints travel out to infinity at the speed of light, accompanied by a gravitational shockwave. A closely related family of solutions are the $C$-metrics, which describe black holes created in pairs and pulled to infinity by cosmic strings [131]. The $C$-metrics approach the previous class of solutions in the limit that the black holes are massless, and so lose their horizons and become point particles moving at the speed of light. What are all of these dual to in the CCFT? A natural conjecture is that they are dual to a correlation function involving two twist field insertions, corresponding to the endpoints of the cosmic string on $\mathcal{CS}^2$, along with insertions of some other operators dual to the moving endpoints of the string/the gravitational shockwave.

- **EREs of multiple intervals:** A direct generalisation of our work is to CCFT EREs of multiple intervals. These are proportional to a theory-dependent function of the cross ratios of the interval endpoints, and can thus provide more detailed CFT information than just $c$ [101]. How can we calculate CCFT multiple-interval EREs holographically? Clearly we must introduce more than two cosmic string endpoints on $\mathcal{CS}^2$, which in our preferred Wick rotation must all lie on the same great

circle. The challenge then is to determine the bulk solution with minimum action, subject to that boundary condition. Such a calculation will almost certainly be more complicated than our single-interval case. For example, multiple cosmic strings can snap and re-connect to one another in various ways. Even if the cosmic strings remain static, different configurations may minimise the bulk action, depending on the relative positions of their endpoints–possibly leading to phase transitions in the ERE, as occurs in AdS/CFT [132, 133]. AdS/CFT may provide a guide, however, for example in applying Schottky uniformisation [134, 135] or modifying the cosmic brane to produce correct ERE behaviour when $n < 1$ [136].

- **Other CCFT states:** In the bulk, we considered only empty Minkowski spacetime, holographically dual to the CCFT's conformal vacuum. What about other asymptotically flat spacetimes, presumably dual to more complicated CCFT states? Obvious candidates include black holes, or more generally any solution that breaks bulk translational symmetry. Indeed, translational symmetry over-constrains the CCFT, for example forcing low-point correlation functions to be distributional, unless a shadow transformation is performed [22]. Celestial amplitudes have been computed in many backgrounds without bulk translational symmetry: for a sampling, see refs. [43, 72, 137–144]. Can these be extended to include cosmic strings dual to twist fields, and if so, then what can we learn about CCFT EREs or pseudo-entropy?

- **Other approaches to flat holography:** How do our results relate to other approaches to holography in asymptotically flat spacetime? In particular, how can we calculate EREs holographically in these other approaches? For example, Carrollian holography proposes that a theory with Carrollian conformal symmetry is holographically dual to Einstein–Hilbert gravity in asymptotically flat spacetime. The conjectured Carrollian theory "lives" on all of $\mathcal{I}^{\pm}$, and hence is codimension one relative to the bulk, and has Carrollian conformal symmetry, which roughly speaking is the limit of conformal symmetry in which the speed of light goes to infinity. Holographic calculations of Carrollian EE appear in refs. [145–147]. Are these related to our calculations for CCFT? What about bulk theories besides Einstein–Hilbert gravity? For example, how do we calculate CCFT EREs in the top-down proposal for celestial holography of refs. [13, 39] (based on string theory constructions in which D1-branes break bulk translational symmetry)? Can we calculate EREs independently in the bulk theory and in the CCFT, and compare? More generally, we hope that our results open the door to many new approaches to EREs in flat holography.

# Acknowledgements

We would like to thank Christoph Uhlemann for collaboration at an early stage in this project, and Tadashi Takayanagi for reading and commenting on the draft. We would also like to thank Martin Ammon, Arjun Bagchi, Nele Callebaut, Laura Donnay, Sabine Harribey, Nabil Iqbal, Victoria Martin, Prahar Mitra, Robert Myers, Gerben Oling, Andrea Puhm, Ana-Maria Raclariu, Atul Sharma, István M. Szécsényi, and Julio Virrueta for useful discussions.

**Funding information**     F.C. was supported by the Deutsche Forschungsgemeinschaft (DFG) under Grant No. 406116891 within the Research Training Group RTG 2522/1. A. O'B. gratefully acknowledges support from the Simons Center for Geometry and

Physics, Stony Brook University at which some of the research for this paper was performed. Nordita is supported in part by Nordforsk. The work of R.R. was supported by the European Union's Horizon Europe research and innovation program under the Marie Sklodowska-Curie Grant Agreement No. 101104286. The work of S.T. was supported by Basic Science Research Program through the National Research Foundation of Korea (NRF) funded by the Ministry of Education through the Center for Quantum Spacetime (CQUeST) of Sogang University (NRF-2020R1A6A1A03047877)

# A  $d = 2$ CCFT EREs from boundary terms

In this appendix we present an alternative derivation of the results in sec. 3.3 for the EREs of $d = 2$ CCFT by adapting the AdS/CFT approach of ref. [98] to Minkowski spacetime. From a celestial holography perspective, our method extends that of ref. [73] to $n \neq 1$.

## A.1  A boundary integral for the on-shell action

In order to compute the EREs we will follow a method of calculation suggested in the context of AdS/CFT in ref. [98]. Rather than orbifolding the spacetime and computing the $n^{\text{th}}$-derivative of the on-shell action through a boundary term on the regulated conical singularity, as we did in sec. 3.3, we instead compute the on-shell action of empty Minkowski space with a non-trivial cutoff near the boundary of the hyperbolic slices.

Our starting point is Minkowski space in flat slicing as in eq. (3.6), which we reproduce here for convenience

$$\mathrm{d}s^2 = -\mathrm{d}U\mathrm{d}V + V^2\mathrm{d}w\,\mathrm{d}\bar{w}. \tag{A.1}$$

We adopt hyperbolic slicing by dividing the spacetime into different regions, similar to what we did in spherical coordinates in sec. 4.2:

- AdS$_+$: $U, V > 0$;

- AdS$_-$: $U, V < 0$;

- dS$_+$: $U < 0$, $V > 0$; and

- dS$_-$: $U > 0$, $V < 0$.

In the AdS$_\pm$ regions, we adopt hyperbolic slicing by defining new coordinates $\tau$ and $y$ via

$$U \equiv \tau y, \qquad V \equiv \frac{\tau}{y}. \tag{A.2}$$

These coordinates take values in the ranges $\tau \in (-\infty, \infty)$ and $y \in [0, \infty)$, with positive and negative $\tau$ corresponding to AdS$_+$ and AdS$_-$, respectively. In these coordinates the metric becomes

$$\mathrm{d}s^2 = -\mathrm{d}\tau^2 + \frac{\tau^2}{y^2}\left(\mathrm{d}y^2 + \mathrm{d}w\,\mathrm{d}\bar{w}\right). \tag{A.3}$$

In the dS$_\pm$ regions we define new coordinates $\tilde{\tau}$ and $\tilde{\psi}$ via

$$U \equiv -\tilde{\tau}\tilde{y}, \qquad V \equiv \frac{\tilde{\tau}}{\tilde{y}}. \tag{A.4}$$

These take values in the ranges $\tilde{\tau} \in (-\infty, \infty)$ and $\tilde{y} \in [0, \infty)$, with positive and negative $\tilde{\tau}$ corresponding to dS$_+$ and dS$_-$, respectively. In these coordinates the metric becomes

$$\mathrm{d}s^2 = \mathrm{d}\tilde{\tau}^2 + \frac{\tilde{\tau}^2}{\tilde{y}^2}\left(-\mathrm{d}\tilde{y}^2 + \mathrm{d}w\,\mathrm{d}\bar{w}\right). \tag{A.5}$$

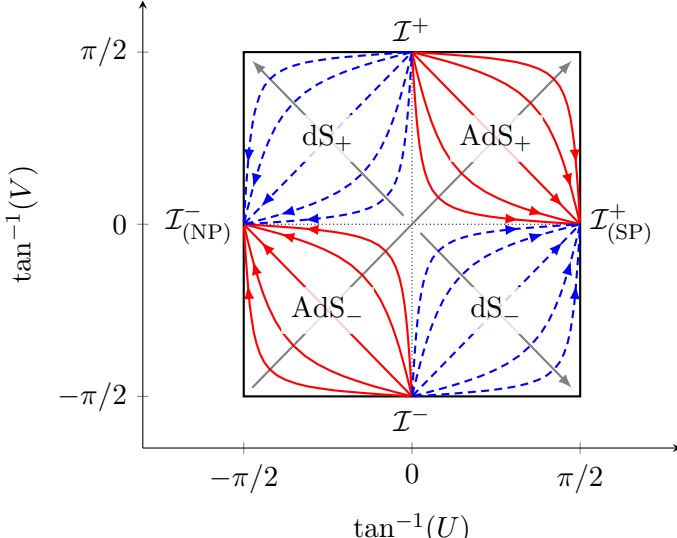

Figure 6: Hyperbolic slicing of Minkowski spacetime, depicted in the plane of $\tan^{-1}(V)$ versus $\tan^{-1}(U)$, with $U$ and $V$ the flat slicing coordinates of eqs. (3.5) and (A.1). The dotted lines are the lightcone of the origin. The two regions labelled $\mathrm{AdS}_{\pm}$ are foliated by Euclidean $\mathrm{AdS}_{d+1}$ slices, denoted by the solid red lines in the figure, with the metric in eq. (A.3). The red arrows parallel to the slices denotes the direction of increasing radial coordinate $y$. The grey arrow perpendicular to the slices denotes the direction of increasing Milne time $\tau$. The two regions labelled $\mathrm{dS}_{\pm}$ are foliated by Lorentzian $\mathrm{dS}_{d+1}$ slices, denoted by blue dashed lines, with the metric in eq. (A.5). The blue arrows parallel to the slices denote the direction of increasing dS time $\tilde{y}$, while the grey arrows perpendicular to the slices denote the direction of increasing Milne time $\tilde{\tau}$. We labeled past and future null infinity as $\mathcal{I}^-$ and $\mathcal{I}^+$, respectively, and the north and south poles of these as $\mathcal{I}^-_{(\mathrm{NP})}$ and $\mathcal{I}^+_{(\mathrm{SP})}$, respectively.

The slicings in the $\mathrm{AdS}_{\pm}$ and $\mathrm{dS}_{\pm}$ regions are sketched in fig. 6.

As in sec. 3.3 we will regularise the volume of this spacetime by imposing cutoffs on the coordinates. We again impose $|UV| \leq L^2$ for some large distance $L$, which corresponds to imposing $|\tau| \leq L$ and $\tilde{\tau} \leq L$ in the $\mathrm{AdS}_{\pm}$ and $\mathrm{dS}_{\pm}$ regions, respectively. However, instead of imposing a cutoff on large $V$ as in eq. (3.36), in the $\mathrm{AdS}_{\pm}$ regions we will impose a Fefferman–Graham cutoff on the hyperbolic slices,

$$y \geq y_c\, e^{-\chi(w,\bar{w})}, \qquad e^{\chi(w,\bar{w})} = \frac{n|z_2 - z_1||w|^{n-1}}{|w^n - 1|^2 + |z_2 w^n - z_1|^2}, \tag{A.6}$$

for some small $y_c$. With this cutoff, the induced metric on this cutoff surface is $y_c^{-2}$ times the metric of the uniformised sphere appearing on the right-hand side of eq. (2.13),

$$\mathrm{d}s_{y_c}^2 = y_c^{-2}\, \frac{4n^2|z_2 - z_1|^2|w|^{2(n-1)}}{\left(|w^n - 1|^2 + |z_2 w^n - z_1|^2\right)^2}\mathrm{d}w\,\mathrm{d}\bar{w} + \dots\,, \tag{A.7}$$

where the dots denote terms which are subleading at small $y_c$. Using the natural defining function on the hyperbolic slices, we would thus read off the "boundary metric" as precisely that of the uniformised sphere. Equivalently, we can motivate eq. (A.6) by first imposing a flat Fefferman–Graham cutoff on the hyperbolic slicing of the metric in eq. (3.20), $u/v \geq y_c^2$

for $u, v > 0$ and $v/u \geq y_c^2$ for $u, v < 0$, where $v = u + 2r$. At leading order in small $y_c$, the finite superrotation in eq. (3.12) maps this to the cutoff in eq. (A.6). By the same logic, in the $\mathrm{dS}_\pm$ regions we impose

$$\tilde{y} \geq \tilde{y}_c \, e^{-\chi(w,\bar{w})}, \tag{A.8}$$

for some small $\tilde{y}_c$.

We wish to evaluate the on-shell action $S_{\mathrm{grav}}^\star(n)$ appearing in the ERE formula, eq. (3.23). Since the spacetime is just empty Minkowski space, the bulk contribution to this on-shell action vanishes. However, we will find non-zero contributions to the on-shell action from the cutoff surfaces.

We begin by considering the contribution to the on-shell action from the $\mathrm{AdS}_+$ region. This region has two boundary components: the infrared cutoff surface at $\tau = L$, and the Fefferman–Graham cutoff surface at $y = y_c$. The Gibbons–Hawking–York boundary term evaluated on the Fefferman–Graham cutoff surface is

$$
\begin{aligned}
S_{y_c}^\star &= \frac{1}{8\pi G_{\mathrm{N}}} \int_{y=y_c} \mathrm{d}^3 x \, \sqrt{|\gamma|} \, K_\gamma \\
&= \frac{1}{8\pi G_{\mathrm{N}}} \int \mathrm{d}w \, \mathrm{d}\bar{w} \int_0^L \mathrm{d}\tau \, \tau \left( \frac{e^{2\chi}}{2y_c^2} + \partial\chi \, \bar{\partial}\chi - \partial\bar{\partial}\chi \right) + O(y_c^2) \\
&= \frac{L^2}{16\pi G_{\mathrm{N}}} \int \mathrm{d}w \, \mathrm{d}\bar{w} \left( \frac{e^{2\chi}}{y_c^2} + 2\partial\chi \, \bar{\partial}\chi - 2\partial\bar{\partial}\chi \right) + O(y_c^2),
\end{aligned}
\tag{A.9}
$$

where $\gamma$ is the induced metric on the cutoff surface, and $K_\gamma$ is its mean curvature. Meanwhile, on the $\tau = L$ cutoff surface we impose Neumann boundary conditions through the boundary term [73]

$$
\begin{aligned}
S_L^\star &= -\frac{1}{8\pi G_{\mathrm{N}}} \int_{\tau=L} \mathrm{d}^3 x \, \sqrt{|\gamma|} \left( K_\gamma - \frac{2}{L} \right) \\
&= -\frac{L^2}{16\pi G_{\mathrm{N}}} \int \mathrm{d}w \, \mathrm{d}\bar{w} \int_{y_c e^{-\chi}}^\infty \mathrm{d}y \, \frac{1}{y^3} \\
&= -\frac{L^2}{32\pi G_{\mathrm{N}}} \int \mathrm{d}w \, \mathrm{d}\bar{w} \, \frac{e^{2\chi}}{y_c^2}.
\end{aligned}
\tag{A.10}
$$

In principle, a corner term at the joint between the two cutoff surfaces could also appear [148]. However, in this case the corner term vanishes because the normals to the two surfaces are orthogonal to each other.

Summing the action contributions in eqs. (A.9) and (A.10) yields an answer that diverges when we remove the cutoff $y_c \to 0$. We can remedy this by adding to the action a volume counterterm on the $y = y_c$ surface,

$$
\begin{aligned}
S_{\mathrm{c.t.}} &= -\frac{1}{8\pi G_{\mathrm{N}}} \int_{y=y_c} \mathrm{d}^3 x \sqrt{|\gamma|} \frac{1}{\tau} \\
&= -\frac{L^2}{32\pi G_{\mathrm{N}}} \int \mathrm{d}w \, \mathrm{d}\bar{w} \left( \frac{e^{2\chi}}{y_c^2} + 2\partial\chi \, \bar{\partial}\chi \right) + O(y_c^2).
\end{aligned}
\tag{A.11}
$$

With this addition, we find that the total contribution of the $\mathrm{AdS}_+$ region to the on-shell action is

$$
S_{\mathrm{AdS}_+}^\star(n) = \frac{L^2}{16\pi G_{\mathrm{N}}} \int \mathrm{d}w \, \mathrm{d}\bar{w} \left( \partial\chi \, \bar{\partial}\chi - 2\partial\bar{\partial}\chi \right). \tag{A.12}
$$

The calculation of the contribution of the $\mathrm{AdS}_-$ region to the on-shell action proceeds identically, and yields a contribution to the on-shell action equal to that from the $\mathrm{AdS}_+$ region. Similarly, the calculations in the $\mathrm{dS}_\pm$ regions proceed almost identically, but with

appropriate minus signs due to the change in signature of the slices. For example, for the $dS_+$ region, the Gibbons–Hawking–York term on the small-$\tilde{y}_c$ cutoff is

$$
\begin{aligned}
S^\star_{\tilde{y}_c} &= -\frac{1}{8\pi G_N} \int_{\tilde{y}=\tilde{y}_c} \mathrm{d}^3 x \, \sqrt{|\gamma|} \, K_\gamma \\
&= -\frac{L^2}{16\pi G_N} \int \mathrm{d}w \, \mathrm{d}\bar{w} \left( \frac{e^{2\chi}}{\tilde{y}_c^2} - 2\partial\chi \, \bar{\partial}\chi + 2\partial\bar{\partial}\chi \right).
\end{aligned}
\tag{A.13}
$$

The contribution from the IR cutoff surface at large $\tilde{\tau} = L$ is

$$
S^\star_L = \frac{1}{8\pi G_N} \int_{\tilde{\tau}=L} \mathrm{d}^3 x \, \sqrt{|\gamma|} \left( K_\gamma - \frac{2}{L} \right) = \frac{L^2}{32\pi G_N} \int \mathrm{d}w \, \mathrm{d}\bar{w} \, \frac{e^{2\chi}}{\tilde{y}_c^2}.
\tag{A.14}
$$

Finally, we remove the term which diverges when $\tilde{y}_c \to 0$ using a volume counterterm,

$$
\begin{aligned}
S_{\text{c.t.}} &= \frac{1}{8\pi G_N} \int_{\tilde{y}=\tilde{y}_c} \mathrm{d}^3 x \sqrt{|\gamma|} \frac{1}{\tilde{\tau}} \\
&= \frac{L^2}{32\pi G_N} \int \mathrm{d}w \, \mathrm{d}\bar{w} \left( \frac{e^{2\chi}}{\tilde{y}_c^2} - 2\partial\chi \, \bar{\partial}\chi \right).
\end{aligned}
\tag{A.15}
$$

Summing these contributions, we obtain a contribution to the on-shell action from the $dS_+$ region that is equal to that from the $AdS_+$ region given in eq. (A.12). The computation for the $dS_-$ region is identical.

The total on-shell action, obtained from the sum of the contributions from the $AdS_\pm$ and $dS_\pm$ regions, is thus four times the result in eq. (A.12),

$$
S^\star_{\text{grav}}(n) = \frac{L^2}{4\pi G_N} \int \mathrm{d}w \, \mathrm{d}\bar{w} \left( \partial\chi \, \bar{\partial}\chi - 2 \, \partial\bar{\partial}\chi \right),
\tag{A.16a}
$$

$$
e^{\chi(w,\bar{w})} = \frac{n|z_2 - z_1||w|^{n-1}}{|w^n - 1|^2 + |z_2 w^n - z_1|^2}.
\tag{A.16b}
$$

Eq. (A.16a) reproduces the expected $d = 2$ CFT Weyl anomaly reviewed in eq. (2.6). The integrand is singular where $\chi$ is singular, namely at $w = 0$ and $w \to \infty$. We will regularise the integral by limiting the integration region to $W_{\min} \leq |w| \leq W_{\max}$, where $W_{\min}$ and $W_{\max}$ are chosen such that the circumference of these cutoff surfaces, measured using the metric (2.13), is equal to $2\pi n\epsilon$, for some small $\epsilon$,

$$
W_{\min} = \left( \frac{1 + |z_1|^2}{2|z_2 - z_1|} \epsilon \right)^{1/n}, \qquad W_{\max} = \left( \frac{1 + |z_2|^2}{2|z_2 - z_1|} \epsilon \right)^{-1/n}.
\tag{A.17}
$$

By construction, these limits on the integration over $W$ match those in eq. (3.41). We now proceed to evaluate the integral in eq. (A.16a) over this integration region.

## A.2   Evaluation of the integral

We write eq. (A.16a) as

$$
S^\star_{\text{grav}}(n) = \frac{L^2}{4G_N} I,
\tag{A.18}
$$

where we have defined the integral

$$
\begin{aligned}
I &\equiv \frac{1}{\pi} \int \mathrm{d}w \, \mathrm{d}\bar{w} \left( \partial\chi \, \bar{\partial}\chi - 2\partial\bar{\partial}\chi \right) \\
&= \frac{1}{2\pi} \int \mathrm{d}^2 w \, \sqrt{|q|} \left( q^{AB} \nabla_A \chi \nabla_B \chi - 2 q^{AB} \nabla_A \nabla_B \chi \right),
\end{aligned}
\tag{A.19}
$$

where $q_{AB}$ is the metric of the complex plane, $q_{AB}dw^A dw^B = dw\,d\bar{w}$ with $w \equiv w^1 + iw^2$. In eq. (A.19), we integrate the first term by parts, and the second term is a total derivative that integrates to zero. Defining polar coordinates via $w \equiv We^{i\theta}$ we thus find

$$I = \frac{1}{2\pi}\int_0^{2\pi} d\theta \Big[W(\chi-2)\partial_W\chi\Big]_{W=W_{\min}}^{W=W_{\max}} - \frac{1}{2\pi}\int d^2w\,\sqrt{|q|}q^{AB}\chi\nabla_A\nabla_B\chi. \tag{A.20}$$

The boundary contribution in eq. (A.20), which we call $I_1$, is straightforward to evaluate,

$$I_1 \equiv \int_0^{2\pi} d\theta\Big[W(\chi-2)\partial_W\chi\Big]_{W=W_{\min}}^{W=W_{\max}}$$
$$= 4n + (n+1)\log\left(\frac{1+|z_2|^2}{n|z_2-z_1|}W_{\max}^{n+1}\right) + (n-1)\log\left(\frac{1+|z_1|^2}{n|z_2-z_1|}W_{\min}^{1-n}\right). \tag{A.21}$$

Substituting the limits on $W$ in eq. (A.17), we then find

$$I_1 = 2\left(n+\frac{1}{n}\right)\log\left(\frac{2}{\epsilon}\right) - 2n\log n + 4n$$
$$+ \left(1+\frac{1}{n}\right)\log\left(\frac{|z_2-z_1|}{1+|z_2|^2}\right) - \left(1-\frac{1}{n}\right)\log\left(\frac{|z_2-z_1|}{1+|z_1|^2}\right). \tag{A.22}$$

The bulk contribution in eq. (A.20), which we call $I_2$, is

$$I_2 \equiv -\frac{1}{2\pi}\int d^2w\,\sqrt{|q|}q^{AB}\chi\nabla_A\nabla_B\chi$$
$$= \frac{2n^2|z_2-z_1|^2}{\pi}\int_0^\infty dW\int_0^{2\pi}d\theta\,\frac{W^{2n-1}}{F(W,\theta)^2}\log\left[\frac{n|z_2-z_1|W^{n-1}}{F(W,\theta)}\right], \tag{A.23}$$

where we have defined

$$F(W,\theta) \equiv 1 + |z_1|^2 - 2W^n\,\mathrm{Re}\left[(1+\bar{z}_1z_2)e^{in\theta}\right] + W^{2n}\left(1+|z_2|^2\right). \tag{A.24}$$

We simplify the integrand in eq. (A.23) by defining new variables $s \equiv \sqrt{a/b}\,W^n$ and $\phi \equiv n\theta + \beta$, where $\beta$ is the complex argument of $1 + \bar{z}_1z_2$ and

$$a \equiv \frac{|z_2-z_1|}{1+|z_1|^2}, \qquad b \equiv \frac{|z_2-z_1|}{1+|z_2|^2}. \tag{A.25}$$

Eq. (2.15) gives $ab = \sin^2(\ell/2)$, and hence $0 < ab < 1$. In terms of these new variables,

$$I_2 = \big[2n\log n - (n-1)\log a - (n+1)\log b\big]J_1(ab) - 2nJ_2(ab) + 2(n-1)J_3(ab), \tag{A.26}$$

where we have defined the three integrals,

$$J_1(\lambda) \equiv \frac{\lambda}{\pi}\int_0^\infty ds\int_0^{2\pi}d\phi\,\frac{s}{\left(1+s^2-2\sqrt{1-\lambda}\,s\cos\phi\right)^2}, \tag{A.27a}$$

$$J_2(\lambda) \equiv \frac{\lambda}{\pi}\int_0^\infty ds\int_0^{2\pi}d\phi\,\frac{s\left[\log\left(1+s^2-2\sqrt{1-\lambda}\,s\cos\phi\right)^2 - \log\lambda\right]}{\left(1+s^2-2\sqrt{1-\lambda}\,s\cos\phi\right)^2}, \tag{A.27b}$$

$$J_3(\lambda) \equiv \frac{\lambda}{\pi}\int_0^\infty ds\int_0^{2\pi}d\phi\,\frac{s\log s}{\left(1+s^2-2\sqrt{1-\lambda}\,s\cos\phi\right)^2}, \tag{A.27c}$$

to be evaluated for $0 < \lambda < 1$. The integrals $J_1(\lambda)$ and $J_2(\lambda)$ are best handled by the change of variables[14]

$$s^2 \equiv 1 + \lambda(\sigma - 1) + 2\sqrt{\lambda(1-\lambda)}\,\sqrt{\sigma}\cos\varphi, \qquad \tan\phi \equiv \frac{\sqrt{\lambda\sigma}\sin\varphi}{\sqrt{1-\lambda}+\sqrt{\lambda\sigma}\cos\varphi}, \qquad \text{(A.28)}$$

with $\sigma \in [0, \infty)$ and $\varphi \in [0, 2\pi)$. This transformation removes all dependence of the integrands on the angular coordinate $\varphi$, such that the integral over $\varphi$ becomes trivial, giving a multiplicative factor of $2\pi$. We then find

$$J_1(\lambda) = \int_0^\infty \mathrm{d}\sigma \frac{1}{(1+\sigma)^2} = 1, \qquad \text{(A.29a)}$$

$$J_2(\lambda) = \int_0^\infty \mathrm{d}\sigma \frac{\log(1+\sigma)}{(1+\sigma)^2} = 1, \qquad \text{(A.29b)}$$

where the integral in the second line is straightforwardly evaluated by parts.

To evaluate $J_3(\lambda)$, we make the same transformation as in eq. (A.28), except in place of $\sigma$ we use $\rho \equiv \sqrt{\sigma}$, which gives

$$J_3(\lambda) = \frac{1}{2\pi}\int_0^\infty \mathrm{d}\rho \int_0^{2\pi} \mathrm{d}\varphi \frac{\rho}{(1+\rho^2)^2}\log\left[1 - \lambda + \lambda\rho^2 + 2\sqrt{\lambda(1-\lambda)}\,\rho\cos\varphi\right]. \qquad \text{(A.30)}$$

Integrating by parts to remove the logarithm, we obtain

$$J_3(\lambda) = \log(1-\lambda) + \frac{1}{\pi}\int_0^\infty \mathrm{d}\rho \int_0^{2\pi} \mathrm{d}\varphi \frac{\lambda\rho + \sqrt{\lambda(1-\lambda)}\cos\varphi}{(1+\rho^2)\left[1 - \lambda + \lambda\rho^2 + 2\sqrt{\lambda(1-\lambda)}\,\rho\cos\varphi\right]} \qquad \text{(A.31a)}$$

$$= \log(1-\lambda) + \int_0^\infty \mathrm{d}\rho \int_{-\infty}^\infty \mathrm{d}\alpha\, \mathcal{J}(\rho, \alpha; \lambda), \qquad \text{(A.31b)}$$

where in the second line we have performed the substitution $\alpha \equiv \tan(\varphi/2)$ and defined

$$\mathcal{J}(\rho, \alpha; \lambda) \equiv \frac{2}{\pi} \frac{\lambda\rho(1+\alpha^2) + \sqrt{\lambda(1-\lambda)}(1-\alpha^2)}{(1+\rho^2)(1+\alpha^2)\left[(1-\lambda+\lambda\rho^2)(1+\alpha^2) + 2\sqrt{\lambda(1-\lambda)}\,\rho(1-\alpha^2)\right]}. \qquad \text{(A.32)}$$

The integrand $\mathcal{J}(\rho, \alpha; \lambda)$ has four simple poles on the imaginary $\alpha$ axis, two at $\alpha = \pm i$ and two at $\alpha = \pm i\alpha_*$, where

$$\alpha_* \equiv \frac{|\lambda\rho^2 - 1 + \lambda|}{(\sqrt{1-\lambda}-\sqrt{\lambda}\rho)^2}. \qquad \text{(A.33)}$$

We denote the corresponding residues as $R(\pm i)$ and $R(\pm i\alpha_*)$, respectively, where

$$2\pi i R(\pm i) = \pm\frac{1}{\rho(1+\rho^2)}, \qquad 2\pi i R(\pm i\alpha_*) = \pm\frac{\mathrm{sign}(\lambda\rho^2 - 1 + \lambda)}{\rho(1+\rho^2)}. \qquad \text{(A.34)}$$

Furthermore, $\mathcal{J}(r, \alpha; \lambda)$ vanishes as $\alpha^{-2}$ as $\alpha \to \infty$. We can therefore close the contour of integration over $\alpha$ into a large semicircle in the upper-half complex $\alpha$-plane, picking up contributions only from the poles at $\alpha = i$ and $\alpha = i\alpha_*$. The result is

$$J_3(\lambda) = \log(1-\lambda) + \int_0^\infty \mathrm{d}\rho \frac{1 + \mathrm{sign}(\lambda\rho^2 - 1 + \lambda)}{\rho(1+\rho^2)} = 0. \qquad \text{(A.35)}$$

---

[14]This transformation arises from a shift in the origin of the polar coordinates, with $\varphi$ the angular coordinate around the new origin and $\sigma$ proportional to the square of the radial distance to the new origin.

Substituting the results for $J_1$, $J_2$, and $J_3$ from eqs (A.29) and (A.35), respectively, and $a$ and $b$ from eq. (A.25), into eq. (A.26) then gives

$$I_2 = 2n \log n - 2n - (n-1) \log \left( \frac{|z_2 - z_1|}{1 + |z_1|^2} \right) - (n+1) \log \left( \frac{|z_2 - z_1|}{1 + |z_2|^2} \right). \tag{A.36}$$

Substituting $I = I_1 + I_2$ into eq. (A.18), with the result for $I_1$ in eq. (A.22) and the result for $I_2$ in eq. (A.36), we obtain the on-shell gravitational action at replica index $n$,

$$S_{\text{grav}}^\star(n) = \frac{L^2}{4G_{\text{N}}} \left[ 2 \left( n + \frac{1}{n} \right) \log \left( \frac{2}{\epsilon} \right) - \left( n - \frac{1}{n} \right) \log \left( \frac{|z_2 - z_1|}{1 + |z_2|^2} \right) \right.$$
$$\left. - \left( n - \frac{1}{n} \right) \log \left( \frac{|z_2 - z_1|}{1 + |z_1|^2} \right) + 2n \right]. \tag{A.37}$$

Finally, we substitute this result into eq. (3.23) to obtain the single-interval EREs,

$$S_{\ell,\text{n}} = \left( 1 + \frac{1}{n} \right) \frac{iL^2}{2G_{\text{N}}} \log \left( \frac{2|z_2 - z_1|}{\epsilon \sqrt{(1 + |z_1|^2)(1 + |z_2|^2)}} \right). \tag{A.38}$$

Using eq. (2.15) to replace $z_1$ and $z_2$ with the great circle distance $\ell$, we thus find

$$S_{\ell,\text{n}} = \left( 1 + \frac{1}{n} \right) \frac{iL^2}{2G_{\text{N}}} \log \left[ \frac{2}{\epsilon} \sin \left( \frac{\ell}{2} \right) \right], \tag{A.39}$$

which agrees with our result in sec. 3.3, namely eq. (3.46).

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
