# Peer review of "Entanglement Rényi entropies in celestial holography"

_SciPost Physics, doi:SciPost Phys. 19, 042 (2025)_

## Round 1 · Referee Report · Hong Zhe Chen (Referee 1) · 2025-3-6

Strengths

  1. This paper fills a gap in our understanding of the celestial holographic dictionary: namely, the explicit classical bulk dual of CCFT Renyi entropy.

  2. This paper presents in celestial holography new and creative applications of methods for calculating Renyi entropies originally established in other contexts: uniformization conformal transformations in two-dimensional CFTs, and the Casini-Huerta-Myers transformation in higher dimensions.

  3. The explanation of the calculations were clear and should be understandable by an appropriately broad audience (though some suggestions for improvement on finer details can be found in the "Requested changes" section).

Weaknesses

  1. One might argue that it's not clear why CCFT Renyi entropies are an interesting set of quantities to consider. In particular, it is not obvious yet what new lessons we will be able to learn about the bulk or boundary theories from this entry in the celestial holographic dictionary.

  2. There are some minor issues with wording and typos, laid out in the "Requested changes" section.

Report

This paper studies the Renyi entanglement entropies of celestial CFTs (CCFTs) hypothetically dual to gravitational theories in asymptotically flat spacetime. In particular, it finds creative uses in this context for existing tools in calculating Renyi entropies in two-dimensional CFTs and AdS/CFT in higher dimensions. The main results include the construction of explicit classical bulk solutions dual to CCFTs on replica manifolds and expressions for CCFT Renyi entropies calculated using these bulk solutions. There are some suggestions for minor improvements and typo corrections found in the "Requested changes" section of this referee report, but these do not significantly detract from the impact of this paper.

In general, I believe this paper meets all of SciPost Physics' general acceptance criteria and the particular journal expectations indicated by the authors.

Requested changes

** Section 1: Introduction and summary

*** Section 1.1: Motivation: celestial holography

  1. On Page 2, in the second paragraph, it is stated that "Translations act as a constant shift of the advanced or retarded time coordinate". However, I believe this is only true for time translations. Spatial translations result in a non-constant shift of the advanced or retarded time. See penultimate sentence in paragraph around (5.2.3) in arXiv:1703.05448.

  2. In the same paragraph, "Lorentz transformations acts" should be grammatically corrected to "Lorentz transformations act".

  3. In the last paragraph of page 2, it is stated that "Any duality is ultimately an isomorphism between gauge-invariant states of the Hilbert spaces of two theories." For dS/CFT and celestial holography, I suspect this is not true if one adopts a conventional notion of "states" in the boundary CFT. I will now explain what I mean and why it might be worthwhile to discuss this point particularly in this paper.

A state of an asymptotically flat or dS bulk describes the configuration on a slice (say future infinity) which spans all d directions of the boundary. However, a state of a d-dimensional CFT (or more generally a QFT) describes the configuration on a (d-1)-dimensional slice --- for the sake of disambiguation, I will refer to this as the "conventional" notion of states in the CFT. It therefore seems implausible for bulk states to be related isometrically to conventional states in the boundary theory. (The second paragraph on page 3 touches on this point somewhat. This point is also emphasized in the referenced paper arXiv:2105.00331.)

For completeness (though this is somewhat beside the point) I'll mention that other authors sometimes advocate for a generalized notion of a state in the CFT, which, by definition, are isometric to states in the bulk. (For example, I believe this is the viewpoint of arXiv:2501.00462 --- see section 3.3 therein.) However, this generalized notion of states in the boundary CFT should not be confused with the conventional states of the CFT. For example, celestial CFTs are not unitary with respect to conventional states, but are unitary with respect to this generalized notion of states, since the bulk theory is unitary (presumably).

Why this point might be worth discussing further in this paper: I believe the Renyi entropies considered here are the entropies for conventional states of the CFT, in particular describing configurations on (d-1)-dimensional slices where the twist operators are inserted. Because these conventional states are not isomorphic to bulk states, the aforementioned entropies do not describe the entropies of bulk states (contrary to what a reader with a background in AdS/CFT might expect). There is a hint of this discussion in the second paragraph on page 31, but I feel this fact that CCFT entropy does not equal bulk entropy is a conceptually important point that is worth stating explicitly in the introduction.

  1. On page 3, in the fourth paragraph, it may be worthwhile to remark that bulk in- and out-states do not correspond respectively to the two hemispheres, so the relation between the bulk and boundary transition matrices is not the naive equality that readers might imagine. (Each hemisphere can contain operator insertions for both in- and out-states of the bulk. This is closely related to point 3 above.)

*** Section 1.3: The replica trick in celestial holography

  1. Page 7, second paragraph: Usually in holography, at least in AdS/CFT, the UV of the boundary theory corresponds to the IR of the bulk. (Admittedly, it is not obvious to me that this is true also in celestial holography.) With this intuition in mind, it seems peculiar to attribute the UV divergence of the CFT to a bulk UV divergence. In particular, I would not say that $S_{grav}^*$ diverges due to effects at short distances near the string deep in the bulk. Indeed, if the bulk did not have infinite IR extent, then (3.28) would be finite, as expressed in (3.29). It may very well be that $\delta$ has a subtle effect on the final answer for the entropy, but I am suggesting that this comes from the part of the regulator tube near the spacetime boundary. I would therefore have attributed all divergences of $S_{grav}^*$ to effects occurring in the IR of the bulk, near the asymptotic boundary. However, the choice of wording is subjective to some degree and it is not obvious to me that a UV-IR relation between the boundary and bulk generally holds in celestial holography anyway, so I leave it to the authors whether or not to rephrase the wording.

  2. Page 7, third paragraph: It is stated that "we could Wick rotate the direction normal to the great circle defined by the two twist fields, thus producing a compact time direction." Don't we simply end up with two-dimensional Lorentzian de Sitter spacetime? We might get global de Sitter or just static patches depending on the choice of time coordinate that is Wick rotated, but in both cases the Lorentzian time direction is not compact.

(A parenthetical comment that the authors can feel free to ignore: The states one obtains from either these immediate Wick rotations from the sphere or the Wick rotation from $\mathbb{R}^2$ mentioned at the end of this paragraph are just conformal to each other. It's a choice of language, but I would have called these states the "same" in a CFT.)

*** Section 1.4: Summary of results

  1. On page 8, it is stated that the universal term in the entropy for general CFTs in any odd dimension is $\propto (-1)^{\frac{d}{2}-2} Z(CS^d)$. What is the reference for this formula? Is the exponent really correct? (If so, the $-2$ seems superfluous.)

** Section 3: Holographic calculation of CCFT EREs: $d=2$

*** Section 3.1: Holographic duals of uniformisation and twist fields

  1. Above (3.1): Should $\theta_s$ be ranged in $[0,\pi]$ instead?

  2. Below (3.5): Should $U$ be ranged in $(-\infty,\infty)$ instead?

*** Section 3.2: Setup of holographic calculation

  1. Below (3.27), in "$\mu=1,2,\ldots,d$", is $d$ really $2$ in this section? I would have thought that we want the Greek indices $\mu,\nu,\rho,\ldots$ to be bulk $(d+2)=4$-dimensional tensor indices. Since we are working with a fixed dimension in this section, it would at any rate be good to specify the range of the indices explicitly or remind the reader that we have $d=2$ in this section.

  2. Below (3.29), I think it's inaccurate to say that (3.29) necessarily gives a flat entanglement spectrum (i.e. that $S_{\ell,n}$ is independent of $n$). This is because $A_{string}$ is generically $n$-dependent.

    This should be evident from the calculations in section 4 starting from (4.14), but just in case, let me briefly describe two ways to see that $A_{string}$ is generically $n$-dependent. Firstly, from the perspective before the orbifold by the replica symmetry (so the bulk is everywhere smooth and has $\mathcal{M}_n$ as its celestial boundary), the bulk geometry generically depends on $n$ and the bulk codimension-two fixed surface under the replica symmetry can have an area $A_{string}$ dependent on $n$. Secondly, from the perspective after the orbifold (so the bulk is conically singular and has $\mathcal{M}_n/\mathbb{Z}_n=\mathcal{M}_1$ as its celestial boundary), the backreaction of the string on the spacetime geometry should depend on the string tension. Consequently, $A_{string}$ should generically depend on $n$. The fact that $A_{string}$ depends on $n$ is also the reason we find a non-flat entanglement spectrum in AdS/CFT, except when we fix the value of $A_{string}$ by hand (i.e. when we consider fixed-area states). I gather that we are not considering fixed-area states in this paper.

    Perhaps a better set of words is that, in order to obtain a finite regularized expression for $A_{string}$ which displays its $n$-dependence, we need to regulate its divergence near the asymptotic boundary. This involves the part of the regulator tube near the asymptotic boundary, so we will keep $\delta$ around in case it plays a subtle role here.

*** Section 3.3: Holographic calculation

  1. Top of page 20 and Figure 2: I believe it's inaccurate to say that the geometry depicted in the bottom right of Figure 20 is conically singular. In particular, in the top and middle rows (which are coordinate transformations of each other), the celestial geometry $\mathcal{M}_n$ has an opening angle of $2\pi n$. The phase of $w$ in the middle panel gives a coordinate opening angle $2\pi$, but the actual metric opening angle is $2\pi n$ if we account for the singular conformal factor. If we take an orbifold by the replica symmetry, we should then get a smooth geometry $\mathcal{M}_n/\mathbb{Z}_n=\mathcal{M}_1$ with metric opening angle $2\pi$ in the bottom row. (Again, $2\pi/n$ is just a coordinate opening angle for the phase of $w$.)

  2. Eqns. (3.36) and (3.37) and Figure 3: Eqns. (3.36) and (3.37) use (3.15a) and assume that the corrections to (3.15a) are negligible. Can the profile of the red cutoff surface in Figure 3 really be trusted everywhere it is drawn? I suspect that, for any large but fixed value of $r_c$, the approximations (3.15a), (3.36), and (3.37) (as well as the profile of the red surface in Figure 3) cannot be trusted for $z$ sufficiently close to $z_1$ and $z_2$. (E.g. one might worry that the corrections to (3.15a) can become comparable to the displayed term as $z\to z_1,z_2$.) If this is the case, then to take increasingly smaller $\delta$, one must choose an increasingly large $r_c$ in order for the calculations in this section to be accurate.

    If the above is true, then I suggest firstly explaining this explicitly, and secondly omitting any parts of the red cutoff surface in Figure 3 that might not be accurate.

  3. Page 25, third paragraph: Again, can we really trust Figure 3b in the limit where we take $\delta\to 0$ while keeping $r_c$ finite? (See preceding point.)

** Section 4: Holographic calculation of CCFT EREs: $d\ge 2$

*** Section 4.1: Review: Casini-Huerta-Myers in AdS/CFT

**** Section 4.1.2: Casini-Huerta-Myers in AdS

  1. Somewhere around (4.19), one should mention that this geometry is an AdS hyperbolic (a.k.a. topological) black hole and give appropriate references. (At least one reference that I know of which studies entanglement in AdS/CFT using hyperbolic black holes of various temperatures is arXiv:2312.06803. One might look for older references cited therein.)

  2. At the bottom of page 31, it is stated that "we will perform a bulk coordinate change that produces this cosmic brane" --- I think it's inaccurate to describe the procedure as a coordinate change in $d>2$. In particular, the intrinsic geometry of each AdS slice changes, even away from the brane, in the step where we replace (4.17) with (4.19). For example, with (4.17), the geometry is locally pure AdS and is locally maximally symmetric, as can be seen if one calculates the Riemann tensor. However, I believe the hyperbolic black hole with (4.19) is not locally pure AdS even away from the brane and has a different Riemann tensor.

    A comment about why $d>2$ is different from $d=2$: In $3$ dimensions, solutions to vacuum Einstein's equation with a cosmological constant are all locally pure (A)dS. This explains why we were successful in section 3 in describing everything in terms of changes in coordinates, without affecting the local geometry away from the string. The same statement is not true in $d+1>3$ dimensions.

*** Section 4.2: EREs in higher-dimensional CCFT

  1. Around (4.24), it might be helpful, for sake of the reader, to state that the $t$ here and below is different from the $t$ introduced in section 4.1.

  2. Figure 5: The caption states that the vertical grey arrow indicates the direction of increasing Milne time $\tau$. However, from the definition (4.28) of $\tau$, I believe that the arrow should be reversed in AdS$_-$.

**** Section 4.2.1: AdS$_\pm$ patch contributions

  1. Around (4.28), one should mention that the $\tau$ here and below is different from the $\tau$ introduced in section 4.1.

  2. Above (4.35), it is stated that "Eq. (4.31) remains a local solution to the vacuum Einstein equations if we replace $f(\zeta)$ in eq. (4.32) with the function in eq. (4.19)". I encourage the authors to explain why this is true.

    As remarked in point 16 above, replacing (4.32) by (4.19) leads to a different local geometry on each slice of fixed $\tau$. In particular, the metric in the square brackets of (4.31) becomes a Euclidean AdS hyperbolic black hole at different temperatures set by $\zeta_0$ --- I understand that this is a solution to the $(d+1)$-dimensional Einstein equations with a cosmological constant (with the possible exception of the conical singularity at the bifurcation surface). However, it is not immediately obvious to me that this must uplift to a solution of the $(d+2)$-dimensional Einstein equations.

    I do believe the claim, because I briefly checked it computationally (out of laziness) for several examples of $d\ge 3$. Nevertheless I think it would be beneficial to have a general argument presented. One might say that at least half of section 4 is devoted to deriving this solution, so it would be worthwhile to explain why it is indeed a solution.

**** Section 4.2.3: Results for EREs

  1. Top of page 39: Same comment about the formula $(-1)^{\frac{d}{2}-2}Z(S^d)$ as point 7 above.

** Section 5: Summary and outlook

  1. In the last paragraph of the main text, it is stated that "Carrollian conformal symmetry [...] roughly speaking is the limit of conformal symmetry in which the speed of light goes to infinity." I am no expert on this, but I thought that limit gives a Galilean theory and the opposite limit of zero speed of light gives the Carrollian theory.

Recommendation

Publish (easily meets expectations and criteria for this Journal; among top 50%)

  • validity: high
  • significance: good
  • originality: high
  • clarity: high
  • formatting: perfect
  • grammar: excellent

Author:  Ronnie Rodgers  on 2025-06-10  [id 5558]

(in reply to Report 1 by Hong Zhe Chen on 2025-03-06)

We thank Referee #1 for their detailed reading and positive evaluation of our paper. We especially thank Referee #1 for recognizing the paper's strengths.

In the "Weaknesses" section of their report, Referee #1 lists two items. The first item is a question: why are CCFT entanglement Rényi entropies (EREs) interesting? What can we learn from them, about either the bulk theory or the CCFT? The second item is a detailed list of changes throughout the paper.

With regard to the first item: in the draft we submitted, we attempted to address these questions in the last two paragraphs of section 1.1. In AdS/CFT, entanglement has been studied for almost 20 years now, providing a long list of things to learn from holographic entanglement, and ways to use holographic entanglement. Perhaps the biggest lesson is that holography itself may come from CFT entanglement: see our references 50 to 55. Holographic entanglement can also be used for many other things: proving c-theorems and proposing new ones, finding order parameters for confinement and other phase transitions, quantifying the speed at which information spreads after a quench, and so on. In general, the Rényi entropies and other measures of entanglement, such as mutual information, complexity, symmetry resolved entanglement entropy, and others, are a class of (in general non-local) operators that can provide detailed information about a quantum field theory's operator spectrum, excitation spectrum, dynamics, symmetries, and more. Indeed, in AdS/CFT these quantities are relatively easy to calculate, compared to other methods (like lattice simulation or perturbation theory), and indeed AdS/CFT motivated a renaissance of sorts in the study of entanglement in continuum quantum field theories. However, that was only possible once the basic dictionary was established by Ryu and Takayanagi. Our belief is that, in the long-term, all of the above will apply also to the CCFT. Is the dual CFT's mutual information monogomous, as in AdS/CFT? How does entanglement spread in the CCFT after a quench? The CCFT is not unitary, and so does not obey the Zomolodchikov's c-theorem, but could some measure of entanglement provide a different type of c-theorem (i.e. count degrees of freedom along RG flows)? As in AdS/CFT, answering any of these questions first requires establishing the basic dictionary. That was our goal. Our strategy was to start with the simplest case, namely the single-interval Rényi entropies of the CCFT's conformal vacuum, since that is determined entirely by a conformal transformation and the central charge, i.e. by symmetries.

To address the referee's concerns, we added sentences to the final two paragraphs of section 1.1, listing some of the specific questions that holographic calculations of CCFT EREs could answer. We hope this suffices as motivation for our paper.

With regard to Referee #1's second item: the following lists the changes we made.

Section 1.1:

  1. We corrected that sentence. We now refer only to a shift, rather than a constant shift.

  2. We corrected that sentence.

  3. Referee #1 claims that in dS/CFT and celestial holography bulk states cannot be isomorphic to ''conventional'' states of the dual CFT, defined as a configuration on a Cauchy slice of the CFT. However, the literature firmly establishes precisely that for dS/CFT, and the same has been proposed for celestial holography.

In AdS/CFT, dS/CFT, and celestial holography, the isomorphism always comes from the fact that both bulk states and CFT states are representations of the same symmetry group, namely the CFT's conformal group. However, the isomorphism between these can be highly non-trivial. Indeed, a crucial point is that ''isomorphism'' does not mean ''identity map.''

In AdS/CFT the isomorphism is more or less an identity map, because we identify the asymptotic AdS time coordinate with the CFT time coordinate. As a result, bulk unitarity maps to CFT unitarity, bulk entanglement maps to CFT entanglement, thermal states map to thermal states, and so on.

dS/CFT and celestial holography are fundamentally different from AdS/CFT, because in their cases the bulk theory is Lorentzian while the dual CFT is Euclidean (a point emphasised repeatedly in 2105.00331). That does not forbid an isomorphism between the bulk Hilbert space and the dual CFT conventional Hilbert space, but does mean the isomorphism cannot be the identity map. As a result, although bulk unitarity will not necessarily map to unitarity of conventional CFT states, it must map to some kind of constraint on the conventional CFT states. Indeed, in some sense that is what 2501.00462 is doing: determining what bulk unitarity maps to in the CCFT.

What is the isomorphism in dS/CFT? Strominger described the isomorphism in his original proposal for dS/CFT: see section 7 of hep-th/0106113. The bulk wave function defines a state on the past boundary. Bulk diffeomorphisms map that to itself, and in particular, the states form representations of those diffeomorphisms. As mentioned above, these are representations of the CFT's conformal group. Strominger identifies those with dual CFT states defined on a codimension-one sphere of the past boundary, themselves defined by the CFT wave function. He then invokes radial quantization, but crucially for us, that is isomorphic to conventional quantization. The holographic dictionary is thus, ultimately, an ismorphism between bulk states and conventional CFT states, but is clearly not an identity map, i.e. time evolution does not map to time evolution. Building on that, many people have pointed out that the dual CFT is not unitary, even if the bulk theory is: see for example Maldacena astro-ph/0210603 section 5.2.

Celestial holography is less well developed, but the fundamental arguments still apply. In fact, in all cases (AdS/CFT, dS/CFT, and celestial holography) the fundamental argument is just that the states of both theories, bulk gravity and dual CFT, form representations of the same symmetry group, namely the CFT's conformal group, hence they must be isomorphic. We used 2105.00331 as the most explicit example of the isomorphism that we could find for celestial holography, though for the sake of space we omitted the full details, which are complicated (see section 4 of 2105.00331). Indeed, all we really need is that an isomorphism exists, and in principle could be computed.

We received the key message, though: these points did not come across clearly in the paper. We thus made the following changes:

  • On page 2, in the paragraph starting ''Any duality is ultimately...'' we added three sentences providing more detail about the AdS/CFT dictionary, to set up subsequent statements about the dS/CFT and celestial holography dictionaries.

  • On page 3, in the paragraph starting ''The dS/CFT holographic dictionary...'' we added two sentences about unitarity.

  • On page 3, in the paragraph that originally started ''The bulk theory's S-matrix...'' we added several sentences clarifying our current understanding of the celestial holography dictionary.

  • On page 4, in the paragraph starting ''However, celestial holography still...'' we modified and added some text.

  • On page 7, in the paragraph starting ''We can thus calculate...'' we added a sentence about the isomorphism of the CCFT Hilbert spaces obtained via different Wick rotations.

  • Below eq. 1.5 we modified the text that refers to our earlier discussion of the lack of CFT unitarity in dS/CFT.

  • We clarified this in our changes on page 3, in the paragraph that originally started ''The bulk theory's S-matrix...'' (as mentioned in the previous point).

Section 1.3

  1. We acknowledge Referee #1's concern about identifying UV divergences of the bulk theory and the CCFT. In that paragraph, we therefore removed some references to UV and IR, and re-arranged text to eliminate any suggestion of such an identification.
  2. We corrected this sentence, and correspondingly modified the subsequent sentence.

Section 1.4

  1. In that sentence we provided the reference and updated the expression.

Section 3.1

  1. We corrected that typo.
  2. We corrected that typo.

Section 3.2

  1. We corrected that typo.
  2. We agree with Referee #1 on this point. To clarify the need for finite $\delta$ to obtain the correct $n$-dependence, we added text on pages 25-26, in a new paragraph around eq. 3.50. Back on page 19, below eq. 3.29 we added a future-reference to the new text on pages 25-26.

Section 3.3

  1. We agree with Referee #1, and have provided a more careful wording on page 20, in the paragraph starting ''We have introduced many different coordinate systems...'', and in figure 2, in the final row of the figure and in the caption.

  2. and 3. We agree with Referee #1 that corrections to the asymptotic form of the superrotation can become important close to $z_1$ and $z_2$. We have re-worded the discussion of the near-$\cal{I}^{\pm}$ cutoff to account for this: see the changes in the text on page 22, around equations 3.35 to 3.39, including a new paragraph after eq. 3.39. The cutoff is now defined first in terms of $V$, such that away from $z_1$ and $z_2$ it matches $r<r_c$. Close to $z_1$ and $z_2$ the cutoff on $r$ deviates from this, in order to ensure the metric on the celestial sphere remains that of a round sphere. With this updated definition, the cutoffs drawn in the figures are accurate.

Section 4.1.2

  1. Below eq. 4.18 we added a reference to Casini, Huerta, and Myers and to Emparan's original paper on the AdS hyperbolic/topological black hole.

  2. We agree with Referee #1 and changed the language about coordinate changes to language about CHM maps. See also the related changes below in section 4.2.1 around eq. 4.35.

Section 4.2

  1. Mathematically, $t$ in section 4.1 and in equation 4.24 are the same: they are both the time coordinate of Minkowski spacetime in spherical coordinates. However, we understand Referee #1's point, that $t$ in section 4.1 is in the CFT, while the $t$ in eq. 4.24 is in the bulk Einstein-Hilbert theory---and we need to identify a ''time'' coordinate in the dual CCFT, which is Euclidean. To clarify this for readers, we added text below eq. 4.24.

  2. We agree with Referee #1 and corrected figure 5 and its caption accordingly.

Section 4.2.1

  1. On page 34 we added footnote 14 to highlight the difference in the two $\tau$'s defined in eq. 4.28 versus that defined in section 4.1.2.

  2. Referee #1 asked us to prove that the more general $f(\zeta)$ solves the vacuum Einstein equations. To do so, we added appendix B, which derives the Ricci tensor, and added equations and text on page 36 around eqs. 4.35 and 4.36 proving that the more general $f(\zeta)$ makes the Ricci tensor vanish i.e. solves of the vacuum Einstein equations.

Section 4.2.3

  1. We updated the expression below eq. 4.65, consistent with our change in section 1.4.

Section 5

  1. We corrected that typo.

---

## Round 1 · Referee Report · Anonymous (Referee 2) · 2025-3-20

Report

The manuscript aims to calculate the entanglement R\'enyi entropy (ERE) in celestial holography, a conjecture that maps a $(d+2)$-dimensional theory in asymptotically flat spacetime to a d-dimensional celestial sphere. The authors argue that the superrotated Minkowski spacetime serves as the holographic dual of the celestial conformal field theory (CCFT) on the replica manifold (one should extend the angular coordinates warping around the cosmic string to remove the conical singularity in the bulk). They then compute the holographic ERE following the spirit of AdS/CFT as in [109]. The single-interval ERE matches the results of [73] in the limit $n\to 1$. The authors further extend their analysis to higher dimensions using a CHM-like coordinate transformation and identify universal coefficients from the Weyl anomaly. While the work is technically sound, I have several suggestions for improvement:

1) Figure/Table for section 4: Figure 2 provides a helpful summary of coordinates used in Section 3. However, Section 4 introduces additional coordinates related to the CHM map. To enhance clarity, the authors should include a table or figure summarizing the key coordinates and solutions specific to this section.

2)Field theoretic interpretation of ERE: While the holographic computation of ERE is thorough, the manuscript lacks explicit field-theoretic calculations in CCFT. Since the equivalence between holographic ERE and field-theoretic ERE (defined via the reduced density matrix and twist operator insertions) is non-trivial, the authors should clarify how the reduced density matrix is constructed in CCFT or discuss potential subtleties in interpreting their results as genuine ERE.

3)Clarification on the central charge: The discussion of the imaginary central charge (previously noted in [73]) and its dependence on the IR cutoff $L$ requires refinement. The authors should explicitly reconcile two seemingly conflicting claims:

i)The assertion that the central charge is unphysical and vanishes after holographic renormalization.

ii)The argument (via AdS/CFT and dS/CFT analogies) that the central charge diverges in the $L\to\infty$ limit.

A clearer stance on whether the central charge approaches zero or infinity and the physical implications of this divergence is essential.

The authors should clarify these points in the revised version.

Recommendation

Ask for major revision

  • validity: -
  • significance: -
  • originality: -
  • clarity: -
  • formatting: -
  • grammar: -

Author:  Ronnie Rodgers  on 2025-06-10  [id 5559]

(in reply to Report 2 on 2025-03-20)

We thank Referee #2 for their overall positive comments on the technical strength of our work and for their questions about the open conceptual issues. We address each of the three points of their report below.

  1. To explain the relationship among the various coordinate systems introduced in section 4, we have added figure 6 on page 35 and a sentence below eq. 4.32.

  2. Here Referee #2 makes two requests that we believe are related but distinct.

First is a request for an explicit field-theoretic calculation in CCFT. At the moment that is impossible. In some special cases the CCFT is known explicitly, i.e. is defined independently of the bulk theory. For example, 2208.14233 and 2306.00940 propose a string theory construction in which a certain bulk theory of topological gravity is dual to a certain topological CCFT. The latter can be defined independently of the bulk theory, allowing for independent calculations in the bulk theory and CCFT.

However, when the bulk theory is Einstein-Hilbert gravity, no independent definition of the CCFT is currently known: everything we know about it comes from bulk calculations. Indeed, as stated in the introduction to 2105.003311: ''It is an important open challenge to find an intrinsic construction of any CCFT (i.e. other than as a transformation of a bulk theory) starting with a microscopic theory such as string theory. However, it is implausible that such a construction will be possible for the real world any time soon as it would amount to a complete knowledge of all the laws of physics.''

We mentioned this problem on our page 4, in the paragraph starting ''However, celestial holography still faces many challenges.'' In response to Referee #2's request, we added a sentence to that paragraph, emphasising more explicitly that we can only perform calculations in the bulk theory, i.e. that we cannot perform calculations in the CCFT itself.

Second is a request to clarify how the reduced density matrix is constructed in the CCFT and how our results may be interpreted as EREs. In our original version, we explicitly addressed this issue in two places.

First is the final paragraph of section 3.3. There we explain our preferred choice of Wick rotation, which leads to our definition of the CCFT Hilbert space and of the reduced density matrix, namely the following. The two cosmic string endpoints on the celestial sphere define a great circle. If we perform a stereographic projection from the celestial sphere to the complex plane, then that great circle is a line, and the cosmic string endpoints define an interval on that line. We then Wick rotate the direction normal to that line and define a Hilbert space on the resulting Cauchy surface. Our reduced density matrix comes from tracing out states outside the interval defined by the cosmic string endpoints. However, that paragraph also emphasises that other Wick rotations are possible, and we mention some explicitly.

Second is in Section 5, in the bullet point ''Interpretation as entanglement''. There our point was that only cosmic strings with endpoints on the same great circle will compute EREs. If cosmic string endpoints are present away from that great circle, then in our interpretation they will change the states of the CCFT, and indeed they will modify the CCFT density matrix to a transition matrix, and therefore modify EREs to pseudo-entropies. That will be the case for any isomorphism between bulk and CCFT Hilbert space, using any Wick rotation. Exploring those pseudo-entropies is crucial for celestial holography, and we hope that our paper provides some key tools to do so.

Referee #2's second request is directly related to Referee #1's request in their point 3 in section 1.1, to clarify the isomorphism between bulk and CCFT Hilbert spaces. We thus crafted our changes to address both Referee #1 and #2. In particular, our changes on page 3 and the final paragraph of section 1.3 address the isomorphism between Hilbert spaces, how we define the CCFT Hilbert space, and how we define our single interval and reduced density matrix for EREs.

3 What is the value of the $d=2$ CCFT's central charge, $c$? We agree with Referee #2 that this question is essential for celestial holography. However, we believe that currently no one can answer this question, at least when the bulk theory is Einstein-Hilbert gravity.

As stated in our paper, we believe one consistent way to answer this question definitively is via holographic renormalisation, by which we mean adding covariant counterterms at the long-distance cutoff surface $L$ that cancel $L \to \infty$ divergences, render the bulk theory's phase space well-defined and its variational principle well-posed, and guarantee that all observables are diffeomorphism invariant and finite. However, in celestial holography holographic renormalization remains prohibitively difficult. This is why on page 7 we say ''holographic renormalization remains a (perhaps the) major open question in celestial holography''.

As we explain below eq. 1.5, dimensional analysis suggests that $c \propto L^2/G_N$. The key question is: what is the constant of proportionality, and in particular, after holographic renormalization is it zero or not? Since no one has holographically renormalized celestial holography, this question remains unanswered.

In our original draft we could therefore only summarize the state of the art, namely the two proposals for $c$ in the literature, and explain where our result fits in. Calculations of the CCFT stress tensor operator product expansion (OPE) and symmetry algebra suggest $c=0$ (hinting that the CCFT may be a logarithmic CFT). Calculations of the bulk on-shell action suggest $c = i 3 L^2/G_N$. Our result is essentially a calculation of the bulk on-shell action (with a cosmic string), and reassuringly, our result for $c$ agrees with the previous calculations in that class. We believe that, given the current state of the art, that is the best anyone can do.

(As a side comment (too long to include in the paper), other examples of holography do not provide clear guidance. In most examples of AdS/CFT and dS/CFT, the CFT is typically in a large-$N$ limit, with correspondingly large central charge. The main examples are Einstein-Hilbert gravity in (A)dS (matrix large-$N$), Vasiliev higher-spin gravity in (A)dS (vector large-$N$), and SYK models (melonic large-$N$). However, some examples cases have $c=0$, such as new massive gravity in $(2+1)$-dimensional AdS at the critical mass, dual to a logarithmic CFT. Even in celestial holography, some examples have large, non-zero $c$, but for bulk theories that are not Einstein-Hilbert gravity. For example, Yang-Mills theory (coupled to a bulk dilaton) (2209.02724, 2308.09741, 2403.18896) or Mabuchi gravity coupled to $d=4$ Wess-Zumino Witten theory on Burns space (2208.14233, 2306.00940) are both conjectured to be dual to chiral algebras on the celestial sphere in a large $N$ limit, with correspondingly large, non-zero $c$. However, these cases have not been holographically renormalized either, so these values of $c$ are just as tentative as those computed using Einstein-Hilbert gravity.)

In short, the state of the art remains incomplete, but promising. Our key contribution, as emphasized in the paper, is to add an entry to the celestial holographic dictionary: ERE twist fields are dual to a certain type of cosmic string. Crucially, we are highly confident that this identification will survive holographic renormalization, regardless of the value of $c$. In $d=2$ this identification is determined by conformal symmetry, which survives holographic renormalization in all known cases, and the $d>2$ cases are straightforward extrapolations, consistent also with Lewkowycz and Maldacena's definition of generalized entropy (1304.4926), which also survives holographic renormalization in all known cases. In short, our results do not settle the questions about $c$, but are consistent with comparable calculations, and they provide a key ingredient for further work to determine $c$, among many other things. (For example, our results should enable holographic calculations of CCFT mutual information, which is independent of renormalisation, and can provide $c$.)

To address Referee #2's concern, we added several sentences to the paragraph that starts ''However, in $d=2$ CCFT the value of $c$...'' at the bottom of page 8. These sentences clarify how our result fits with earlier attempts to compute $c$ holographically, and how our entry in the celestial holographic dictionary should survive holographic renormalization, regardless of the value of $c$.

---

## Round 2 · Referee Report · Hong Zhe Chen (Referee 1) · 2025-6-11

Report

Warnings issued while processing user-supplied markup:

  • Inconsistency: plain/Markdown and reStructuredText syntaxes are mixed. Markdown will be used.
    Add "#coerce:reST" or "#coerce:plain" as the first line of your text to force reStructuredText or no markup.
    You may also contact the helpdesk if the formatting is incorrect and you are unable to edit your text.

Echoing my previous report, I believe this paper meets SciPost Physics' general acceptance criteria and the journal expectations indicated by the authors. Moreover, it seems that the authors have addressed the suggestions made in my last report. I therefore recommend the publication of this manuscript.

As I will elaborate below, I am still not convinced that the bulk Hilbert space is isomorphic to the CFT Hilbert space (according to the conventional notion of "states" in the CFT) in dS/CFT and celestial holography. However, I do not think this is a point on which the decision to publish should critically hinge. (So, if the authors still feel confident about this claim, then I would not be strongly opposed to the authors keeping this stance in the paper.) Let me now elaborate on my confusion, so that at least I might learn something from the authors' reply or vice versa.

The authors remind us in their reply that both the bulk Hilbert space and the CFT Hilbert space give representations for the same conformal symmetry group. Firstly, I agree with this statement, but I don't think that's enough to conclude that the Hilbert spaces are isomorphic to each other.

Secondly, to establish a common language, let us consider section 4 of arXiv:2105.00331, which the authors mention in their reply. The bulk Hilbert space, I would say in this setup, is $\mathcal{H}_{in}$ or equivalently $\mathcal{H}_{out}$ --- the two are isomorphic, being related by unitary evolution. The (conventional) CFT Hilbert space, I would say, is $\mathcal{H}_N$ or equivalently $\mathcal{H}_S$ --- again, I think the two are isomorphic. Let me emphasize I really do mean "or" and not "and" in these sentences, meaning, e.g. the CFT Hilbert space is $\mathcal{H}_N$ or equivalently it is $\mathcal{H}_S$, but it is not some union, sum, or product of the two at the same time. In CFT for example, by radial quantization, one should be able to prepare all, e.g. ket, states using operator insertions in say, the southern hemisphere, or even just at the south pole --- these states form the full Hilbert space. Bra states can be prepared on the opposing hemisphere and the path integral over the whole sphere computes inner products. (The restriction of operator insertions to a hemisphere in state preparation is related to the radial-ordering mentioned further below.)

The point is, I don't think $\mathcal{H}_{in}$ (or equivalently $\mathcal{H}_{out}$) is isomorphic to $\mathcal{H}_N$ (or equivalently $\mathcal{H}_S$). I also don't think section 4 in arXiv:2105.00331 claims this isomorphism exists. (And, if I am reading the text on page 3 of v2 of the current manuscript correctly, it seems the authors here are also not claiming this isomorphism.) But what other meaning could there be to having an isomorphism between the bulk Hilbert space and CCFT Hilbert space? (As mentioned in my previous report, if one instead considers a less conventional notion "states" in the CFT as I think some papers do, then one might be able to claim an isomorphism with the bulk Hilbert space almost by definition.)

Thirdly, to further illustrate this point, let us try to write down the hypothetical isomorphism between bulk states and CCFT states (in the conventional sense). Consider the bulk states

$$ |\psi\rangle=a_{in}(w_1)^\dagger \cdots a_{in}(w_n)^\dagger |0\rangle_{bulk}\in\mathcal{H}_{in} $$
$$ |\psi'\rangle=a_{out}(w_1')^\dagger \cdots a_{out}(w_m')^\dagger |0\rangle_{bulk}\in\mathcal{H}_{out} $$
where each $a(w)^\dagger$ is a creation operator associated with a conformal primary wavefunction (or possibly a shadow) for a bulk field, with $w$ denoting a position on the celestial sphere. Suppose $w_1,\ldots,w_n$ includes points distributed throughout the sphere (specifically, not restricted to a hemisphere) and the same is true of $w_1',\ldots,w_m'$. The question I would like to ask is: what CCFT states would the above bulk states correspond to under the hypothetical isomorphism?

An isomorphism of Hilbert spaces must be an isometry, preserving inner products, so let's consider the inner product of the above bulk states. Keeping implicit the isomorphism between $\mathcal{H}_{in}$ and $\mathcal{H}_{out}$ (i.e. the unitary evolution that involves the $S$-matrix),

$$ \langle \psi'|\psi\rangle =\langle 0| a_{out}(w_1') \cdots a_{out}(w_m') a_{in}(w_1)^\dagger \cdots a_{in}(w_n)^\dagger |0\rangle_{bulk} $$
$$ =\langle O_-(w_1') \cdots O_-(w_m') O_+(w_1) \cdots O_+(w_n) \rangle_{CCFT} $$
From what little I know about celestial holography (so please correct me if I'm wrong), this is equated to a CCFT correlator, as expressed in the second line. This notation for the correlator means I imagine performing the Euclidean CCFT path integral with the indicated operator insertions. Now, crucially, this means that if I want to write it as a CCFT vacuum expectation value of operators acting on states in, say radial quantization on celestial space, then radial ordering $\mathcal{R}$ must be imposed:
$$\langle O_-(w_1') \cdots O_-(w_m') O_+(w_1) \cdots O_+(w_n) \rangle_{CCFT} \ = \langle 0| \mathcal{R}\left[ O_-(w_1') \cdots O_-(w_m') O_+(w_1) \cdots O_+(w_n) \right] |0\rangle_{CCFT} $$
I find it difficult to believe that there is an isomorphic map of $|\psi\rangle$ and $|\psi'\rangle$ to the CCFT Hilbert space, such that the inner product of the resulting CCFT states coincides with the above. Among other things, because $\mathcal{R}$ reorders the operator insertions, the naive guess, that $|\psi\rangle$ maps isomorphically to $O_+(w_1) \cdots O_+(w_n) |0\rangle_{CCFT}$ and similarly for $|\psi'\rangle$, does not seem to work.

Recommendation

Publish (easily meets expectations and criteria for this Journal; among top 50%)

---

## Round 2 · Referee Report · Anonymous (Referee 2) · 2025-7-10

Report

The authors have addressed my question in detail. Entanglement entropy in flat holography is a topic that deserves further study, and while some issues may remain unresolved, the results presented here are consistent with prior works and provide an interpretation of a certain cosmic string as the holographic dual of a twist operator. Therefore, I recommend acceptance for publication.

Recommendation

Publish (meets expectations and criteria for this Journal)

---

## Round 2 · Author Response

We thank both referees for their time and effort reading and commenting on the paper. In response to the referees, we have made various changes to the paper, which provide detail on many points but do not change any of our results or conclusions. We believe that these changes greatly clarify various points for readers, and that they fully address all concerns of both referees.

---

## Round 2 · List of Changes

In the resubmitted version of the paper, all changes appear as blue text. Details of the changes and their motivations are given in the replies to the individual referee reports.

---

## Editorial Decision

published